# Why Lottery Ticket Wins? A Theoretical Perspective of Sample Complexity on Pruned Neural Networks

**Shuai Zhang**
Rensselaer Polytechnic Institute
Troy, NY, USA 12180
zhangs21@rpi.edu

**Meng Wang**
Rensselaer Polytechnic Institute
Troy, NY, USA 12180
wangm7@rpi.edu

**Sijia Liu**
Michigan State University
East Lansing, MI, USA 48824
MIT-IBM Watson AI Lab, IBM Research
liusiji5@msu.edu

**Pin-Yu Chen**
IBM Research
Yorktown Heights, NY, USA 10562
Pin-Yu.Chen@ibm.com

**Jinjun Xiong**
University at Buffalo
Buffalo NY, USA 14260
jinjun@buffalo.edu

## Abstract

The *lottery ticket hypothesis* (LTH) [20] states that learning on a properly pruned network (the *winning ticket*) improves test accuracy over the original unpruned network. Although LTH has been justified empirically in a broad range of deep neural network (DNN) involved applications like computer vision and natural language processing, the theoretical validation of the improved generalization of a winning ticket remains elusive. To the best of our knowledge, our work, for the first time, characterizes the performance of training a pruned neural network by analyzing the geometric structure of the objective function and the sample complexity to achieve zero generalization error. We show that the convex region near a desirable model with guaranteed generalization enlarges as the neural network model is pruned, indicating the structural importance of a winning ticket. Moreover, when the algorithm for training a pruned neural network is specified as an (accelerated) stochastic gradient descent algorithm, we theoretically show that the number of samples required for achieving zero generalization error is proportional to the number of the non-pruned weights in the hidden layer. With a fixed number of samples, training a pruned neural network enjoys a faster convergence rate to the desired model than training the original unpruned one, providing a formal justification of the improved generalization of the winning ticket. Our theoretical results are acquired from learning a pruned neural network of one hidden layer, while experimental results are further provided to justify the implications in pruning multi-layer neural networks.

## 1 Introduction

Neural network pruning can reduce the computational cost of model training and inference significantly and potentially lessen the chance of overfitting [33, 26, 15, 25, 28, 51, 58, 41]. The recent *Lottery Ticket Hypothesis* (LTH) [20] claims that a randomly initialized dense neural network al-

35th Conference on Neural Information Processing Systems (NeurIPS 2021).

ways contains a so-called "winning ticket," which is a sub-network bundled with the corresponding initialization, such that when trained in isolation, this winning ticket can achieve at least the same testing accuracy as that of the original network by running at most the same amount of training time. This so-called "improved generalization of winning tickets" is verified empirically in [20]. LTH has attracted a significant amount of recent research interests [45, 70, 39]. Despite the empirical success [19, 63, 55, 11], the theoretical justification of winning tickets remains elusive except for a few recent works. [39] provides the first theoretical evidence that within a randomly initialized neural network, there exists a good sub-network that can achieve the same test performance as the original network. Meanwhile, recent work [42] trains neural network by adding the $\ell_1$ regularization term to obtain a relatively sparse neural network, which has a better performance numerically.

However, the theoretical foundation of network pruning is limited. The existing theoretical works usually focus on finding a sub-network that achieves a tolerable loss in either expressive power or training accuracy, compared with the original dense network [2, 71, 61, 43, 4, 3, 35, 5, 59]. To the best of our knowledge, there exists no theoretical support for the *improved* generalization achieved by winning tickets, i.e., pruned networks with faster convergence and better test accuracy.

**Contributions**: This paper provides the *first* systematic analysis of learning pruned neural networks with a finite number of training samples in the oracle-learner setup, where the training data are generated by a unknown neural network, the *oracle*, and another network, the *learner*, is trained on the dataset. Our analytical results also provide a justification of the LTH from the perspective of the sample complexity. In particular, we provide the *first* theoretical justification of the improved generalization of winning tickets. Specific contributions include:

1. **Pruned neural network learning via accelerated gradient descent (AGD)**: We propose an AGD algorithm with tensor initialization to learn the pruned model from training samples. Our algorithm converges to the oracle model linearly, which has guaranteed generalization.

2. **First sample complexity analysis for pruned networks**: We characterize the required number of samples for successful convergence, termed as the *sample complexity*. Our sample complexity bound depends linearly on the number of the non-pruned weights and is a significant reduction from directly applying conventional complexity bounds in [69, 66, 67].

3. **Characterization of the benign optimization landscape of pruned networks**: We show analytically that the empirical risk function has an *enlarged* convex region for a pruned network, justifying the importance of a good sub-network (i.e., the winning ticket).

4. **Characterization of the improved generalization of winning tickets**: We show that gradient-descent methods converge faster to the oracle model when the neural network is properly pruned, or equivalently, learning on a pruned network returns a model closer to the oracle model with the same number of iterations, indicating the improved generalization of winning tickets.

**Notations.** Vectors are bold lowercase, matrices and tensors are bold uppercase. Scalars are in normal font, and sets are in calligraphy and blackboard bold font. $\boldsymbol{I}$ denote the identity matrix. $\mathbb{N}$ and $\mathbb{R}$ denote the sets of nature number and real number, respectively. $\|\boldsymbol{z}\|$ denotes the $\ell_2$-norm of a vector $\boldsymbol{z}$, and $\|\boldsymbol{Z}\|_2$, $\|\boldsymbol{Z}\|_F$ and $\|\boldsymbol{Z}\|_\infty$ denote the spectral norm, Frobenius norm and the maximum value of matrix $\boldsymbol{Z}$, respectively. $[Z]$ stands for the set of $\{1, 2, \cdots, Z\}$ for any number $Z \in \mathbb{N}$. In addition, $f(r) = \mathcal{O}(g(r))$ ( or $f(r) = \Omega(g(r))$ ) if $f \leq C \cdot g$ ( or $f \geq C \cdot g$ ) for some constant $C > 0$ when $r$ is large enough. $f(r) = \Theta(g(r))$ if both $f(r) = \mathcal{O}(g(r))$ and $f(r) = \Omega(g(r))$ holds, where $c \cdot g \leq f \leq C \cdot g$ for some constant $0 \leq c \leq C$ when $r$ is large enough.

## 1.1 Related Work

**Network pruning**. Network pruning methods seek a compressed model while maintaining the expressive power. Numerical experiments have shown that over 90% of the parameters can be pruned without a significant performance loss [10]. Examples of pruning methods include irregular weight pruning [25], structured weight pruning [57], neuron-based pruning [28], and projecting the weights to a low-rank subspace [13].

**Winning tickets**. [20] employs an *Iterative Magnitude Pruning* (IMP) algorithm to obtain the proper sub-network and initialization. IMP and its variations [22, 46] succeed in deeper networks like Residual Networks (Resnet)-50 and Bidirectional Encoder Representations from Transformers (BERT) network [11]. [21] shows that IMP succeeds in finding the "winning ticket" if the ticket is

stable to stochastic gradient descent noise. In parallel, [36] shows numerically that the "winning ticket" initialization does not improve over a random initialization once the correct sub-networks are found, suggesting that the benefit of "winning ticket" mainly comes from the sub-network structures. [18] analyzes the sample complexity of IMP from the perspective of recovering a sparse vector in a linear model rather than learning neural networks.

**Feature sparsity**. High-dimensional data often contains redundant features, and only a subset of the features is used in training [6, 14, 27, 60, 68]. Conventional approaches like wrapper and filter methods score the importance of each feature in a certain way and select the ones with highest scores [24]. Optimization-based methods add variants of the $\ell_0$ norm as a regularization to promote feature sparsity [68]. Different from network pruning where the feature dimension still remains high during training, the feature dimension is significantly reduced in training when promoting feature sparsity.

**Over-parameterized model.** When the number of weights in a neural network is much larger than the number of training samples, the landscape of the objective function of the learning problem has no spurious local minima, and first-order algorithms converge to one of the global optima [37, 44, 64, 50, 9, 49, 38]. However, the global optima is not guaranteed to generalize well on testing data [62, 64].

**Generalization analyses.** The existing generalization analyses mostly fall within three categories. One line of research employs the Mean Field approach to model the training process by a differential equation assuming infinite network width and infinitesimal training step size [12, 40, 56]. Another approach is the neural tangent kernel (NTK) [30], which requires strong and probably unpractical over-parameterization such that the nonlinear neural network model behaves as its linearization around the initialization [1, 17, 72, 73]. The third line of works follow the oracle-learner setup, where the data are generated by an unknown oracle model, and the learning objective is to estimate the oracle model, which has a generalization guarantee on testing data. However, the objective function has intractably many spurious local minima even for one-hidden-layer neural networks [48, 47, 64]. Assuming an infinite number of training samples, [8, 16, 52] develop learning methods to estimate the oracle model. [23, 69, 66, 67] extend to the practical case of a finite number of samples and characterize the sample complexity for recovering the oracle model. Because the analysis complexity explodes when the number of hidden layers increases, all the analytical results about estimating the oracle model are limited to one-hidden-layer neural networks, and the input distribution is often assumed to be the standard Gaussian distribution.

## 2   Problem Formulation

In an oracle-learner model, given any input $\boldsymbol{x} \in \mathbb{R}^d$, the corresponding output $y$ is generated by a pruned one-hidden-layer neural network, called ***oracle***, as shown in Figure 1. The oracle network is equipped with $K$ neurons where the $j$-th neuron is connected to any arbitrary $r_j^*$ ($r_j^* \leq d$) input features. Let $\boldsymbol{W}^* = [\boldsymbol{w}_1^*, \cdots, \boldsymbol{w}_K^*] \in \mathbb{R}^{d \times K}$ denotes all the weights (pruned ones are represented by zero). The number of non-zero entries in $\boldsymbol{w}_j^*$ is at most $r_j^*$. The oracle network is not unique because permuting neurons together with the corresponding weights does not change the output. Therefore, the output label $y$ obtained by the oracle network satisfies [1]

$$y = \frac{1}{K} \sum_{j=1}^{K} \phi(\boldsymbol{w}_j^{*T} \boldsymbol{x}) + \xi := g(\boldsymbol{x}; \boldsymbol{W}^*) + \xi = g(\boldsymbol{x}; \boldsymbol{W}^* \boldsymbol{P}) + \xi, \tag{1}$$

where $\xi$ is arbitrary unknown additive noise bounded by some constant $|\xi|$, $\phi$ is the rectified linear unit (ReLU) activation function with $\phi(z) = \max\{z, 0\}$, and $\boldsymbol{P} \in \{0, 1\}^{K \times K}$ is any permutation matrix. $\boldsymbol{M}^*$ is a ***mask matrix*** for the oracle network, such that $M_{j,i}^*$ equals to 1 if the weight $\boldsymbol{w}_{j,i}^*$ is not pruned, and 0 otherwise. Then, $\boldsymbol{M}^*$ is an indicator matrix for the non-zero entries of $\boldsymbol{W}^*$ with $\boldsymbol{M}^* \odot \boldsymbol{W}^* = \boldsymbol{W}^*$, where $\odot$ is entry-wise multiplication.

Based on $N$ pairs of training samples $\mathcal{D} = \{\boldsymbol{x}_n, y_n\}_{n=1}^N$ generated by the oracle, we train on a ***learner*** network equipped with the same number of neurons in the oracle network. However, the $j$-th neuron in the learner network is connected to $r_j$ input features rather than $r_j^*$. Let $r_{\min}, r_{\max}$, and $r_{\text{ave}}$ denote the minimum, maximum, and average value of $\{r_j\}_{j=1}^K$, respectively. Let $\boldsymbol{M}$ denote the

---

[1]It is extendable to binary classification, and the output is generated by $\text{Prob}(y_n = 1|\boldsymbol{x}_n) = g(\boldsymbol{x}_n; \boldsymbol{W}^*)$.

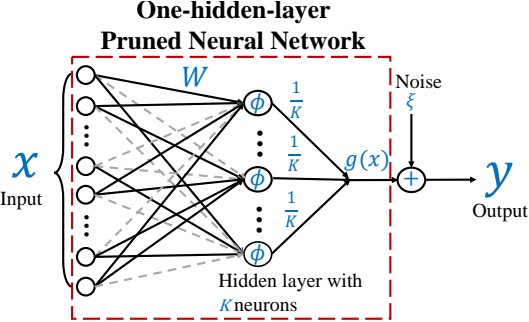

Figure 1: Illustration of the model

mask matrix with respect to the learner network, and $\boldsymbol{w}_j$ is the $j$-th column of $\boldsymbol{W}$. The empirical risk function is defined as

$$\hat{f}_{\mathcal{D}}(\boldsymbol{W}) = \frac{1}{2N} \sum_{n=1}^{N} \Big(\frac{1}{K} \sum_{j=1}^{K} \phi(\boldsymbol{w}_j^T \boldsymbol{x}_n) - y_n\Big)^2. \tag{2}$$

When the mask $\boldsymbol{M}$ is given, the learning objective is to estimate a proper weight matrix $\boldsymbol{W}$ for the *learner* network from the training samples $\mathcal{D}$ via solving

$$\min_{\boldsymbol{W} \in \mathbb{R}^{d \times K}} \quad \hat{f}_{\mathcal{D}}(\boldsymbol{W}) \qquad \text{s.t.} \quad \boldsymbol{M} \odot \boldsymbol{W} = \boldsymbol{W}. \tag{3}$$

$\boldsymbol{M}$ is called an ***accurate mask*** if the support of $\boldsymbol{M}$ covers the support of a permutation of $\boldsymbol{M}^*$, i.e., there exists a permutation matrix $\boldsymbol{P}$ such that $(\boldsymbol{M}^*\boldsymbol{P}) \odot \boldsymbol{M} = \boldsymbol{M}^*$. When $\boldsymbol{M}$ is accurate, and $\xi = 0$, there exists a permutation matrix $\boldsymbol{P}$ such that $\boldsymbol{W}^*\boldsymbol{P}$ is a global optimizer to (3). Hence, if $\boldsymbol{W}^*\boldsymbol{P}$ can be estimated by solving (3), one can learn the oracle network accurately, which has guaranteed generalization performance on the testing data.

We assume $\boldsymbol{x}_n$ is independent and identically distributed from the standard Gaussian distribution $\mathcal{N}(\boldsymbol{0}, \boldsymbol{I}_{d \times d})$. The Gaussian assumption is motivated by the data whitening [34] and batch normalization techniques [29] that are commonly used in practice to improve learning performance. Moreover, training one-hidden-layer neural network with multiple neurons has intractable many fake minima [47] without any input distribution assumption. In addition, the theoretical results in Section 3 assume an accurate mask, and inaccurate mask is evaluated empirically in Section 4.

The questions that this paper addresses include: 1. **what algorithm** to solve (3)? 2. what is the **sample complexity** for the accurate estimate of the weights in the oracle network? 3. what is the **impact of the network pruning** on the difficulty of the learning problem and the performance of the learned model?

# 3 Algorithm and Theoretical Results

Section 3.1 studies the geometric structure of (3), and the main results are in Section 3.2. Section 3.3 briefly introduces the proof sketch and technical novelty, and the limitations are in Section 3.4.

## 3.1 Local Geometric Structure

Theorem 1 characterizes the local convexity of $\hat{f}_{\mathcal{D}}$ in (3). It has two important implications.

1. **Strictly locally convex near ground truth**: $\hat{f}_{\mathcal{D}}$ is strictly convex near $\boldsymbol{W}^*\boldsymbol{P}$ for some permutation matrix $\boldsymbol{P}$, and the radius of the convex ball is negatively correlated with $\sqrt{\tilde{r}}$, where $\tilde{r}$ is in the order of $r_{\text{ave}}$. Thus, the convex ball enlarges as any $r_j$ decreases.

2. **Importance of the winning ticket architecture**: Compared with training on the dense network directly, training on a properly pruned sub-network has a larger local convex region near $\boldsymbol{W}^*\boldsymbol{P}$, which may lead to easier estimation of $\boldsymbol{W}^*\boldsymbol{P}$. To some extent, this result can be viewed as a theoretical validation of the importance of the winning architecture (a good sub-network) in [20]. Formally, we have

**Theorem 1** (Local Convexity). *Assume the mask $M$ of the learner network is accurate. Suppose constants $\varepsilon_0$, $\varepsilon_1 \in (0, 1)$ and the number of samples satisfies*

$$N = \Omega\big(\varepsilon_1^{-2} K^4 \widetilde{r} \log q\big), \tag{4}$$

*for some large constant $q > 0$, where*

$$\widetilde{r} = \frac{1}{8K^4}\Big(\textstyle\sum_{k=1}^{K}\sum_{j=1}^{K}(1 + \delta_{j,k})(r_j + r_k)^{\frac{1}{2}}\Big)^2, \tag{5}$$

*$\delta_{j,k}$ is 1 if the indices of non-pruned weights in the $j$-th and $k$-th neurons overlap and 0 otherwise. Then, there exists a permutation matrix $P$ such that for any $W$ that satisfies*

$$\|W - W^*P\|_F = \mathcal{O}\big(\tfrac{\varepsilon_0}{K^2}\big), \quad \text{and } M \odot W = W, \tag{6}$$

*its Hessian of $\hat{f}_{\mathcal{D}}$, with probability at least $1 - K \cdot q^{-r_{\min}}$, is bounded as:*

$$\Theta\Big(\frac{1 - \varepsilon_0 - \varepsilon_1}{K^2}\Big)I \preceq \nabla^2 \hat{f}_{\mathcal{D}}(W) \preceq \Theta\Big(\frac{1}{K}\Big)I. \tag{7}$$

**Remark 1.1 (Parameter $\widetilde{r}$):** Clearly $\widetilde{r}$ is a monotonically increasing function of any $r_j$ from (5). Moreover, one can check that $\frac{1}{8} r_{\text{ave}} \leq \widetilde{r} \leq r_{\text{ave}}$. Hence, $\widetilde{r}$ is in the order of $r_{\text{ave}}$.

**Remark 1.2 (Local landscape):** Theorem 1 shows that with enough samples as shown in (4), in a local region of $W^*P$ as shown in (6), all the eigenvalues of the Hessian matrix of the empirical risk function are lower and upper bounded by two positive constants. This property is useful in designing efficient algorithms to recover $W^*P$, as shown in Section 3.2.

**Remark 1.3 (Size of the convex region):** When the number of samples $N$ is fixed and $r$ changes, $\varepsilon_1$ can be $\Theta(\sqrt{\widetilde{r}/N})$ while (4) is still met. $\varepsilon_0$ in (7) can be arbitrarily close to but smaller than $1 - \varepsilon_1$ so that the Hessian matrix is still positive definite. Then from (6), the radius of the convex ball is $\Theta(1) - \Theta(\sqrt{\widetilde{r}/N})$, indicating an enlarged region when $\widetilde{r}$ decreases. The enlarged convex region serves as an important component in proving the faster convergence rate, summarized in Theorem 2. Besides this, as Figure 1 shown in [20], the authors claim that the learning is stable if the linear interpolation of the learned models with SGD noises still remain similar in performance, which is summarized as the concept "linearly connected region." Intuitively, we conjecture that the winning ticket shows a better performance in the stability analysis because it has a larger convex region. In the other words, a larger convex region indicates that the learning is more likely to be stable in the linearly connected region.

### 3.2 Convergence Analysis with Accelerated Gradient Descent

We propose to solve the non-convex problem (3) via the accelerated gradient descent (AGD) algorithm, summarized in Algorithm 1. Compared with the vanilla gradient descent (GD) algorithm, AGD has an additional momentum term, denoted by $\beta(W^{(t)} - W^{(t-1)})$, in each iteration. AGD enjoys a faster convergence rate than vanilla GD in solving optimization problems, including learning neural networks [65]. Vanilla GD can be viewed as a special case of AGD by letting $\beta = 0$. The initial point $W^{(0)}$ can be obtained through a tensor method, and the details are provided in Appendix B.

---

**Algorithm 1** Accelerated Gradient Descent (AGD) Algorithm

---

1: **Input:** training data $\mathcal{D} = \{(x_n, y_n)\}_{n=1}^{N}$, gradient step size $\eta$, momentum parameter $\beta$, and an initialization $W^{(0)}$ by the tensor initialization method;
2: Partition $\mathcal{D}$ into $T = \log(1/\varepsilon)$ disjoint subsets, denoted as $\{\mathcal{D}_t\}_{t=1}^{T}$;
3: **for** $t = 1, 2, \cdots, T$ **do**
4:     $W^{(t+1)} = W^{(t)} - \eta \cdot M \odot \nabla \hat{f}_{\mathcal{D}_t}(W^{(t)}) + \beta(W^{(t)} - W^{(t-1)})$
5: **end for**
6: **Return:** $W^{(T)}$

---

The theoretical analyses of our algorithm are summarized in Theorem 2 (convergence) and Lemma 1 (Initialization). The significance of these results can be interpreted from the following aspects.

1. **Linear convergence to the oracle model**: Theorem 2 implies that if initialized in the local convex region, the iterates generated by AGD converge linearly to $\boldsymbol{W}^*\boldsymbol{P}$ for some $\boldsymbol{P}$ when noiseless. When there is noise, they converge to a point $\boldsymbol{W}^{(T)}$. The distance between $\boldsymbol{W}^{(T)}$ and $\boldsymbol{W}^*\boldsymbol{P}$ is proportional to the noise level and scales in terms of $\mathcal{O}(\sqrt{\widetilde{r}/N})$. Moreover, when $N$ is fixed, the convergence rate of AGD is $\Theta(\sqrt{\widetilde{r}/K})$. Recall that Algorithm 1 reduces to the vanilla GD by setting $\beta = 0$. The rate for the vanilla GD algorithm here is $\Theta(\sqrt{\widetilde{r}}/K)$ by setting $\beta = 0$ by Theorem 2, indicating a slower convergence than AGD. Lemma 1 shows the tensor initialization method indeed returns an initial point in the convex region.

2. **Sample complexity for accurate estimation**: We show that the required number of samples for successful estimation of the oracle model is $\Theta\big(\widetilde{r}\log q \log(1/\varepsilon)\big)$ for some large constant $q$ and estimation accuracy $\varepsilon$. Our sample complexity is much less than the conventional bound of $\Theta(d\log q \log(1/\varepsilon))$ for one-hidden-layer networks [69, 66, 67]. This is the first theoretical characterization of learning a pruned network from the perspective of sample complexity.

3. **Improved generalization of winning tickets**: We prove that with a fixed number of training samples, training on a properly pruned sub-network converges faster to $\boldsymbol{W}^*\boldsymbol{P}$ than training on the original dense network. Our theoretical analysis justifies that training on the winning ticket can meet or exceed the same test accuracy within the same number of iterations. To the best of our knowledge, our result here provides the first theoretical justification for this intriguing empirical finding of "improved generalization of winning tickets" by [20].

**Theorem 2** (Convergence). *Assume the mask $\boldsymbol{M}$ of the learner network is accurate. Suppose $\boldsymbol{W}^{(0)}$ satisfies* (6) *and the number of samples satisfies*

$$N = \Omega\big(\varepsilon_0^{-2}K^6\widetilde{r}\log q \log(1/\varepsilon)\big) \tag{8}$$

*for some $\varepsilon_0 \in (0, 1/2)$. Let $\eta = K/14$ in Algorithm 1. Then, the iterates $\{\boldsymbol{W}^{(t)}\}_{t=1}^T$ returned by Algorithm 1 converges linearly to $\boldsymbol{W}^*$ up to the noise level with probability at least $1 - K^2T \cdot q^{-r_{\min}}$*

$$\|\boldsymbol{W}^{(t)} - \boldsymbol{W}^*\boldsymbol{P}\|_F \leq \nu(\beta)^t \|\boldsymbol{W}^{(0)} - \boldsymbol{W}^*\boldsymbol{P}\|_F + \mathcal{O}\Big(\sum_j \sqrt{\frac{r_j \log q}{N}}\Big) \cdot |\xi|, \tag{9}$$

$$\text{and} \quad \|\boldsymbol{W}^{(T)} - \boldsymbol{W}^*\boldsymbol{P}\|_F \leq \varepsilon\|\boldsymbol{W}^*\|_F + \mathcal{O}\Big(\sum_j \sqrt{\frac{r_j \log q}{N}}\Big) \cdot |\xi|, \tag{10}$$

*for a fixed permutation matrix $\boldsymbol{P}$, where $\nu(\beta)$ is the rate of convergence that depends on $\beta$ with $\nu(\beta^*) = 1 - \Theta\big(\frac{1-\varepsilon_0}{\sqrt{K}}\big)$ for some non-zero $\beta^*$ and $\nu(0) = 1 - \Theta\big(\frac{1-\varepsilon_0}{K}\big)$.*

**Lemma 1** (Initialization). *Assume the noise $|\xi| \leq \|\boldsymbol{W}^*\|_2$ and the number of samples $N = \Omega\big(\varepsilon_0^{-2}K^5 r_{\max}\log q\big)$ for $\varepsilon_0 > 0$ and large constant $q$, the tensor initialization method outputs $\boldsymbol{W}^{(0)}$ such that (6) holds, i.e., $\|\boldsymbol{W}^{(0)} - \boldsymbol{W}^*\|_F = \mathcal{O}\big(\frac{\varepsilon_0\sigma_K}{K^2}\big)$ with probability at least $1 - q^{-r_{\max}}$.*

**Remark 2.1 (Faster convergence on pruned network)**: With a fixed number of samples, when $\widetilde{r}$ decreases, $\varepsilon_0$ can increase as $\Theta(\sqrt{\widetilde{r}})$ while (8) is still met. Then $\nu(0) = \Theta(\sqrt{\widetilde{r}}/K)$ and $\nu(\beta^*) = \Theta(\sqrt{\widetilde{r}/K})$. Therefore, when $\widetilde{r}$ decreases, both the stochastic and accelerated gradient descent converge faster. Note that as long as $\boldsymbol{W}^{(0)}$ is initialized in the local convex region, not necessarily by the tensor method, Theorem 2 guarantees the accurate recovery. [66, 67] analyze AGD on convolutional neural networks, while this paper focuses on network pruning.

**Remark 2.2 (Sample complexity of initialization)**: From Lemma 1, the required number of samples for a proper initialization is $\Omega\big(\varepsilon_0^{-2}K^5 r_{\max}\log q\big)$. Because $r_{\max} \leq K r_{\text{ave}}$ and $\widetilde{r} = \Omega(r_{\text{ave}})$, this number is no greater than the sample complexity in (8). Thus, provided that (8) is met, Algorithm 1 can estimate the oracle network model accurately.

**Remark 2.3 (Inaccurate mask)**: The above analyses are based on the assumption that the mask of the learner network is accurate. In practice, a mask can be obtained by an iterative pruning method such as [20] or a one-shot pruning method such as [55]. In Appendix E, we prove that the magnitude pruning method can obtain an accurate mask with enough training samples. Moreover, empirical experiments in Section 4.2 and 4.3 suggest that even if the mask is not accurate, the three properties (linear convergence, sample complexity with respect to the network size, and improved generalization of winning tickets) can still hold. Therefore, our theoretical results provide some insight into the empirical success of network pruning.

### 3.3 The Sketch of Proofs and Technical Novelty

Our proof outline is inspired by [69] on fully connected neural networks, however, major technical changes are made in this paper to generalize the analysis to an arbitrarily pruned network. To characterize the local convex region of $\hat{f}_{\mathcal{D}}$ (Theorem 1), the idea is to bound the Hessian matrix of the population risk function, which is the expectation of the empirical risk function, locally and then characterize the distance between the empirical and population risk functions through the concentration bounds. Then, the convergence of AGD (Theorem 2) is established based on the desired local curvature, which in turn determines the sample complexity. Finally, to initialize in the local convex region (Lemma 1), we construct tensors that contain the weights information and apply a decomposition method to estimate the weights.

Our technical novelties are as follows. First, a direct application of the results in [69] leads to a sample complexity bound that is linear in the feature dimension $d$. We develop new techniques to tighten the sample complexity bound to be linear in $\tilde{r}$, which can be significantly smaller than $d$ for a sufficiently pruned network. Specifically, we develop new concentration bounds (Lemmas 4 and 5 in Appendix) to bound the distance between the population and empirical risk functions rather than using the bound in [69]. Second, instead of restricting the acitivation to be smooth for convergence analysis, we study the case of ReLU function which is non-smooth. Third, new tensors are constructed for pruned networks (see (21)-(23) in Appendix) in computing the initialization, and our new concentration bounds are employed to reduce the required number of samples for a proper initialization. Last, Algorithm 1 employs AGD and is proved to converge faster than the GD algorithm in [69].

### 3.4 Limitations

Like most theoretical works based on the oracle-learner setup, limitations of this work include (1) one hidden layer only; and (2) the input follows the Gaussian distribution. Extension to multi-layer might be possible if the following technical challenges are addressed. First, when characterizing the local convex region, one needs to show that the Hessian matrix is positive definite. In multi-layer networks, the Hessian matrix is more complicated to compute. Second, new concentration bounds need to be developed because the input feature distributions to the second and third layers depend on the weights in previous layers. Third, the initialization approach needs to be revised. The team is also investigating the other input distributions such as Gaussian mixture models.

## 4 Numerical Experiments

The theoretical results are first verified on synthetic data, and we then analyze the pruning performance on both synthetic and real datasets. In Section 4.1, Algorithm 1 is implemented with minor modification, such that, the initial point is randomly selected as $\|\boldsymbol{W}^{(0)} - \boldsymbol{W}^*\|_F/\|\boldsymbol{W}^*\|_F < \lambda$ for some $\lambda > 0$ to reduce the computation. Algorithm 1 terminates when $\|\boldsymbol{W}^{(t+1)} - \boldsymbol{W}^{(t)}\|_F/\|\boldsymbol{W}^{(t)}\|_F$ is smaller than $10^{-8}$ or reaching 10000 iterations. In Sections 4.2 and 4.3, the Gradient Signal Preservation (GraSP) algorithm [55] and IMP algorithm [10, 20][2] are implemented to prune the neural networks. As many works like [11, 10, 20] have already verified the faster convergence and better generalization accuracy of the winning tickets empirically, we only include the results of some representative experiments, such as training MNIST and CIFAR-10 on Lenet-5 [32] and Resnet-50 [27] networks, to verify our theoretical findings.

The synthetic data are generated using a oracle model in Figure 1. The input $\boldsymbol{x}_n$'s are randomly generated from Gaussian distribution $\mathcal{N}(0, \boldsymbol{I}_{d \times d})$ independently, and indices of non-pruned weights of the $j$-th neuron are obtained by randomly selecting $r_j$ numbers without replacement from $[d]$. For the convenience of generating specific $\tilde{r}$, the indices of non-pruned weights are almost overlapped ($\sum_j \sum_k \delta_j \delta_k > 0.95K^2$) except for Figure 5. In Figures 2 and 4, $r_j$ is selected uniformly from $[0.9\tilde{r}, 1.1\tilde{r}]$ for a given $\tilde{r}$, and $r_j$ are the same in value for all $j$ in other figures. Each non-zero entry of $\boldsymbol{W}^*$ is randomly selected from $[-0.5, 0.5]$ independently. The noise $\xi_n$'s are i.i.d. from $\mathcal{N}(0, \sigma^2)$, and the noise level is measured by $\sigma/E_y$, where $E_y$ is the root mean square of the noiseless outputs.

---

[2]The source codes used are downloaded from https://github.com/VITA-Group/CV_LTH_Pre-training.

## 4.1 Evaluation of theoretical findings on synthetic data

**Local convexity near $W^*$.** We set the number of neurons $K = 10$, the dimension of the data $d = 500$ and the sample size $N = 5000$. Figure 2 illustrates the success rate of Algorithm 1 when $\widetilde{r}$ changes. The $y$-axis is the relative distance of the initialization $W^{(0)}$ to the ground-truth. For each pair of $\widetilde{r}$ and the initial distance, 100 trails are constructed with the network weights, training data and the initialization $W^{(0)}$ are all generated independently in each trail. Each trail is called successful if the relative error of the solution $W$ returned by Algorithm 1, measured by $\|W - W^*\|_F / \|W^*\|_F$, is less than $10^{-4}$. A black block means Algorithm 1 fails in estimating $W^*$ in all trails while a white block indicates all success. As Algorithm 1 succeeds if $W^{(0)}$ is in the local convex region near $W^*$, we can see that the radius of convex region is indeed linear in $-\widetilde{r}^{\frac{1}{2}}$, as predicted by Theorem 1.

**Convergence rate.** Figure 3 shows the convergence rate of Algorithm 1 when $\widetilde{r}$ changes. $N = 5000$, $d = 300$, $K = 10$, $\eta = 0.5$, and $\beta = 0.2$. Figure 3(a) shows that the relative error decreases exponentially as the number of iterations increases, indicating the linear convergence of Algorithm 1. As shown in Figure 3(b), the results are averaged over 20 trials with different initial points, and the areas in low transparency represent the standard deviation errors. We can see that the convergence rate is almost linear in $\sqrt{r}$, as predicted by Theorem 2. We also compare with GD by setting $\beta$ as 0. One can see that AGD has a smaller convergence rate than GD, indicating faster convergence.

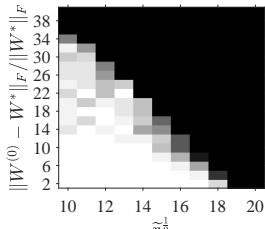

Figure 2: The radius of the local convex region against $\widetilde{r}^{\frac{1}{2}}$

Figure 3: Convergence rate when $\widetilde{r}$ changes

**Sample complexity.** Figures 4 and 5 show the success rate of Algorithm 1 when varying $N$ and $\widetilde{r}$. $d$ is fixed as 100. In Figure 4, we construct 100 independent trails for each pair of $N$ and $\widetilde{r}$, where the ground-truth model and training data are generated independently in each trail. One can see that the required number of samples for successful estimation is linear in $\widetilde{r}$, as predicted by (8). In Figure 5, $r_j$ is fixed as 20 for all neurons, but different network architectures after pruning are considered. One can see that although the number of remaining weights is the same, $\widetilde{r}$ can be different in different architectures, and the sample complexity increases as $\widetilde{r}$ increases, as predicted by (8).

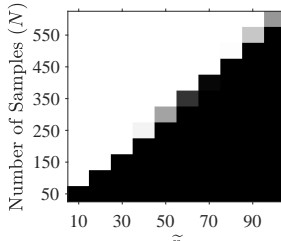
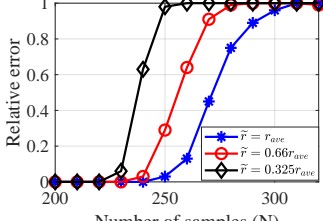
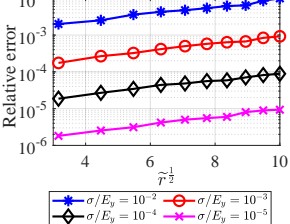

Figure 4: Sample complexity when $\widetilde{r}$ changes

Figure 5: Relative error against $\widetilde{r}$

Figure 6: Relative error against $\widetilde{r}^{\frac{1}{2}}$ at different noise level

**Performance in noisy case.** Figure 6 shows the relative error of the learned model by Algorithm 1 from noisy measurements when $\widetilde{r}$ changes. $N = 1000$, $K = 10$, and $d = 300$. The results are averaged over 100 independent trials, and standard deviation is around $2\%$ to $8\%$ of the corresponding relative errors. The relative error is linear in $\widetilde{r}^{\frac{1}{2}}$, as predicted by (9). Moreover, the relative error is proportional to the noise level $|\xi|$.

## 4.2 Performance with inaccurate mask on synthetic data

The performance of Algorithm 1 is evaluated when the mask $M$ of the learner network is inaccurate. The number of neurons $K$ is 5. The dimension of inputs $d$ is 100. $r_j^*$ of the oracle model is 20 for

all $j \in [K]$. GraSP algorithm [55] is employed to find masks based only on early-trained weights in 20 iterations of AGD. The mask accuracy is measured by $\|\boldsymbol{M}^* \odot \boldsymbol{M}\|_0 / \|\boldsymbol{M}^*\|_0$, where $\boldsymbol{M}^*$ is the mask of the oracle model. The pruning ratio is defined as $(1 - r_{\text{ave}}/d) \times 100\%$. The number of training samples $N$ is 200. The model returned by Algorithm 1 is evaluated on $N_{\text{test}} = 10^5$ samples, and the test error is measured by $\sqrt{\sum_n |y_n - \hat{y}_n|^2 / N_{\text{test}}}$, where $\hat{y}_n$ is the output of the learned model with the input $\boldsymbol{x}_n$, and $(\boldsymbol{x}_n, y_n)$ is the $n$-th testing sample generated by the oracle network.

**Improved generalization by GraSP.** Figure 7 shows the test error with different pruning ratios. For each pruning ratio, we randomly generate 1000 independent trials. Because the mask of the learner network in each trail is generated independently, we compute the average test error of the learned models in all the trails with same mask accuracy. If there are less than 10 trails for certain mask accuracy, the result of that mask accuracy is not reported as it is statistically meaningless. The test error decreases as the mask accuracy increases. More importantly, at fixed mask accuracy, the test error decreases as the pruning ratio increases. That means the generalization performance improves when $\widetilde{r}$ deceases, even if the mask is not accurate.

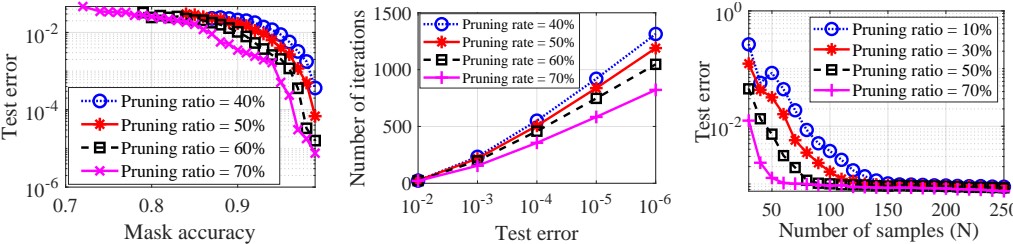

Figure 7: Test error against mask accuracy with different pruning ratios

Figure 8: Convergence rate with mask accuracy in $[0.85, 0.9]$

Figure 9: Test error against the number of samples with mask accuracy in $[0.85, 0.9]$

**Linear convergence.** Figure 8 shows the convergence rate of Algorithm 1 with different pruning ratios. We show the smallest number of iterations required to achieve a certain test error of the learned model, and the results are averaged over the independent trials with mask accuracy between $0.85$ and $0.90$. Even with inaccurate mask, the test error converges linearly. Moreover, as the pruning ratio increases, Algorithm 1 converges faster.

**Sample complexity with respect to the pruning ratio.** Figure 9 shows the test error when the number of training samples $N$ changes. All the other parameters except $N$ remain the same. The results are averaged over the trials with mask accuracy between $0.85$ and $0.90$. We can see the test error decreases when $N$ increases. More importantly, as the pruning ratio increases, the required number of samples to achieve the same test error (no less than $10^{-3}$) decreases dramatically. That means the sample complexity decreases as $\widetilde{r}$ decreases even if the mask is inaccurate.

### 4.3 Performance of IMP on synthetic, MNIST and CIFAR-10 datasets

We implement the IMP algorithm to obtain pruned networks on synthetic, MNIST and CIFAR-10 datasets. Figure 10 shows the test performance of a pruned network on synthetic data with different sample sizes. Here in the oracle network model, $K = 5, d = 100$, and $r_j^* = 20$ for all $j \in [K]$. The noise level $\sigma/E_y = 10^{-3}$. One observation is that for a fixed sample size $N$ greater than 100, the test error decreases as the pruning ratio increases. This verifies that the IMP algorithm indeed prunes the network properly. It also shows that the learned model improves as the pruning progresses, verifying our theoretical result in Theorem 2 that the difference of the learned model from the oracle model decreases as $r_j$ decreases. The second observation is that the test error decreases as $N$ increases for any fixed pruning ratio. This verifies our result in Theorem 2 that the difference of the learned model from the oracle model decreases as the number of training samples increases. When the pruning ratio is too large (greater than 80%), the pruned network cannot explain the data properly, and thus the test error is large for all $N$. When the number of samples is too small, like $N = 100$, the test error is always large, because it does not meet the sample complexity requirement for estimating the oracle model even though the network is properly pruned.

Figures 11 and 12 show the test performance of learned models by implementing the IMP algorithm on MNIST and CIFAR-10 using Lenet-5 [32] and Resnet-50 [27] architecture, respectively. The

experiments follow the standard setup in [10] except for the size of the training sets. To demonstrate the effect of sample complexity, we randomly selected $N$ samples from the original training set without replacement. As we can see, a properly pruned network (i.e., winning ticket) helps reduce the sample complexity required to reach the test accuracy of the original dense model. For example, training on a pruned network returns a model (e.g., $P_1$ and $P_3$ in Figures 11 and 12) that has better testing performance than a dense model (e.g., $P_2$ and $P_4$ in Figures 11 and 12) trained on a larger data set. Given the number of samples, we consistently find the characteristic behavior of winning tickets: That is, the test accuracy could increase when the pruning ratio increases, indicating the effectiveness of pruning. The test accuracy then drops when the network is overly pruned. The results show that our theoretical characterization of sample complexity is well aligned with the empirical performance of pruned neural networks and explains the improved generalization observed in LTH.

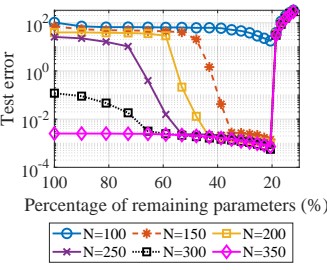

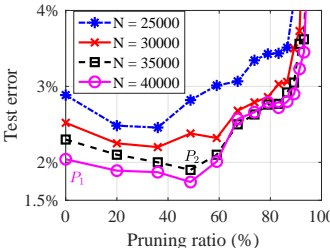

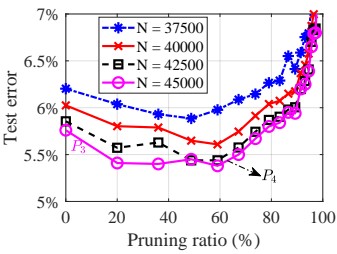

Figure 10: Test error of pruned models on the synthetic dataset

Figure 11: Test accuracy of pruned LeNet-5 on Mnist

Figure 12: Test accuracy of pruned Resnet-50 on Cifar-10

## 5 Conclusions

This paper provides the first theoretical analysis of learning one-hidden-layer pruned neural networks, which offers formal justification of the improved generalization of winning ticket observed from empirical findings in LTH. We characterize analytically the impact of the number of remaining weights in a pruned network on the required number of samples for training, the convergence rate of the learning algorithm, and the accuracy of the learned model. We also provide extensive numerical validations of our theoretical findings.

## Broader impacts

We see no ethical or immediate societal consequence of our work. This paper contributes to the theoretical foundation of both network pruning and generalization guarantee. The former encourages the development of learning method to reduce the computational cost. The latter increases the public trust in incorporating AI technology in critical domains.

## Acknowledgement

This work was supported by AFOSR FA9550-20-1-0122, ARO W911NF-21-1-0255, NSF 1932196 and the Rensselaer-IBM AI Research Collaboration (http://airc.rpi.edu), part of the IBM AI Horizons Network (http://ibm.biz/AIHorizons). We thank Tianlong Chen at University of Texas at Austin, Haolin Xiong at Rensselaer Polytechnic Institute and Yihua Zhang at Michigan State University for the help in formulating numerical experiments. We thank all anonymous reviewers for their constructive comments.

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
