# Supplementary Materials for:

## Why Lottery Ticket Wins? A Theoretical Perspective of Sample Complexity on Pruned Neural Networks

We first provide an overview about techniques used in proving the landscape (Theorem 1), linear convergence to the ground truth (Theorem 2) and tensor initialization (Lemma 1).

1. **Sample complexity scales in** $\{r_j\}_{j=1}^K$: To guarantee the theoretical bounds depend on $\{r_j\}_{j=1}^K$ instead of $d$, we define an equivalent empirical risk function as shown in (12) in Appendix A, from $\mathbb{R}^{\sum_j r_j}$ to $\mathbb{R}$. Existing concentration theorems and landscape analysis built upon (2) can no longer be used here, and thus we revised or updated the corresponding lemmas, which can be found in Appendix G to I. In the initialization methods, for estimating a proper weights that match new empirical risk function, the construction of high-momenta in Appendix B and corresponding proofs in Appendix J are updated accordingly as well;

2. **Local convex region**: In proving Theorem 1 (Appendix C), we first bound the Hessian of the expectation of the new empirical risk function and then obtain the distance of the Hessian of the new empirical risk function to its expectation by concentration theorem. By triangle inequality, the Hessian of the new empirical risk function is characterized in terms of sample size $N$;

3. **Linear Convergence**: In proving Theorem 2 (Appendix D), we first characterize the gradient descent term by *Intermediate Value Theorem* (IVT). However, since the empirical risk function is non-smooth due to the ReLU activation function, IVT is applied in the expectation of the empirical risk function instead, and we later show the gradient generated by finite number of samples is close to its expectation. Therefore, the iterates still converge to the ground truth with enough samples. Further, the linear convergence rate are determined by $\|\boldsymbol{W}^{(t+1)} - \boldsymbol{W}^* \boldsymbol{P}\| / \|\boldsymbol{W}^{(t)} - \boldsymbol{W}^* \boldsymbol{P}\|$, which turns out to be dependent on $\beta$;

4. **Initialization via Tensor Method**: The major challenge for tensor initialization is to construct the proper high dimensional momenta. As we mentioned above, if one directly applies the method in [69], the sample complexity is in $\Theta(d)$. In this paper, we select $\widetilde{\boldsymbol{x}}$ (see (20) in Appendix B), which is the sum of the augmented $\boldsymbol{x}_{\Omega_j}$. In proving Lemma 1, the major idea to bound the estimations of the directions and magnitudes of $\boldsymbol{w}_{j,\Omega_j}$ to the ground values, respectively (see in Appendix F).

## A    Notations

In this section, we first introduce some important notations that will be used in the following proofs, and the notations are summarized in Table 1.

First, for the convenience of proofs, some notations in main contexts, namely, $\Omega_j^*$, $r_j^*$ and $\hat{f}_{\mathcal{D}}$ will be re-defined. We emphasize here that the re-definition of these notations will not affect the presentation of theoretical results in Section 3, and the explanations can be found in the following paragraphs.

Next, given a permutation matrix $\boldsymbol{P}$, we define a group of sets $\{\Omega_j^*\}_{j=1}^K$ with $|\Omega_j^*| = r_j^*$, and $\Omega_j^*$ denotes the indices of non-zero entries in $\boldsymbol{M}^* \boldsymbol{P}$, which is also the non-pruned weights of the $j$-th neuron in the oracle model with respect to ground truth weights $\boldsymbol{M}^* \boldsymbol{P}$, instead of $\boldsymbol{M}^*$. Please note that the sets $\{\Omega_j^*\}_{j=1}^K$ and $\{r_j^*\}_{j=1}^K$ here are just a permutation of these in the main context. Since the permutation of $\{r_j\}_{j=1}^K$ will not change the results in Section 3, we abuse the notations for the convenience of proofs. Correspondingly, for the learner model, the indices of non-pruned weights of the $j$-th neuron is denoted as $\Omega_j$, and $|\Omega_j| = r_j$. Therefore, we have

$$\boldsymbol{w}_j^T \boldsymbol{x} = \boldsymbol{w}_{j,\Omega_j}^T \boldsymbol{x}_{\Omega_j}, \tag{11}$$

where $\boldsymbol{z}_{\Omega_j} \in \mathbb{R}^{r_j}$ is the subvector of $\boldsymbol{z}$ with respect to indices $\Omega_j$ for any vector $\boldsymbol{z} \in \mathbb{R}^d$.

Then, recall the *empirical risk function* defined in (2), it can be re-written as

$$\hat{f}_{\mathcal{D}}(\widetilde{\boldsymbol{w}}) := \frac{1}{2N} \sum_{n=1}^N \left( \frac{1}{K} \sum_{j=1}^K \phi(\boldsymbol{w}_{j,\Omega_j}^T \boldsymbol{x}_{n,\Omega_j}) - y_n \right)^2, \tag{12}$$

Table 1: Table of Notations

| Notation | Description |
|---|---|
| $N$ | The number of training samples; a scalar in $\mathbb{Z}$ |
| $K$ | The number of neurons in the neural network; a scalar in $\mathbb{R}$ |
| $d$ | The dimension of input data; a scalar in $\mathbb{R}$ |
| $\boldsymbol{x}$ | The input data/features; a vector in $\mathbb{R}^d$ |
| $y$ | The output label; a scalar in $\mathbb{R}$ |
| $\hat{f}_{\mathcal{D}}$ | The empirical risk function defined in (12); a mapping from $\mathbb{R}^{\sum_j r_j}$ to $\mathbb{R}$ |
| $f$ | The population risk function defined as $f = \mathbb{E}_{\mathcal{D}}\hat{f}_{\mathcal{D}}$; a mapping from $\mathbb{R}^{\sum_j r_j}$ to $\mathbb{R}$ |
| $\boldsymbol{P}$ | The permutation matrix; a binary matrix in $\{0,1\}^{K \times K}$ |
| $\boldsymbol{W}^*$ | The ground truth weights of oracle network; a matrix in $\mathbb{R}^{d \times K}$ |
| $\boldsymbol{M}^*$ | The mask matrix of the oracle network; a binary matrix in $\{0,1\}^{d \times K}$ |
| $r_j^*$ | The number of non-pruned weights in the $j$-th neuron of oracle network |
| $\boldsymbol{W}$ | The ground truth weights of learner network; a matrix in $\mathbb{R}^{d \times K}$ |
| $\boldsymbol{M}$ | The mask matrix of the learner network; a binary matrix in $\{0,1\}^{d \times K}$ |
| $r_j$ | The number of non-pruned weights in the $j$-th neuron of learner network |
| $r_{\min}$ | The minimal value in $\{r_j\}_{j=1}^K$ |
| $r_{\max}$ | The maximal value in $\{r_j\}_{j=1}^K$ |
| $\Omega_j^*$ | The indices of non-pruned weights in teacher network; a set with size of $r_j^*$ |
| $\Omega_j$ | The indices of non-pruned weights in learner network; a set with size of $r_j$ |
| $\widetilde{\boldsymbol{w}}$ | Contains the non-pruned weights of $\boldsymbol{W}$ and equals to $[\boldsymbol{w}_{1,\Omega_1}^T, \boldsymbol{w}_{2,\Omega_2}^T, \cdots, \boldsymbol{w}_{K,\Omega_K}^T]^T$; a vector in $\mathbb{R}^{\sum_j r_j}$ |
| $\widetilde{\boldsymbol{w}}^*$ | Contains the non-pruned weights of the oracle model; a vector in $\mathbb{R}^{\sum_j r_j}$ |
| $\delta_{i,j}$ | A binary scalar, and the value is 1 if $\Omega_j$ and $\Omega_k$ are overlapped and 0 otherwise |
| $\widetilde{r}$ | The value of $\frac{1}{8K^4}\left(\sum_k \sum_j (1+\delta_{j,k})(r_j+r_k)^{\frac{1}{2}}\right)^2$ |
| $\sigma_i$ | The $i$-th largest singular value of $\boldsymbol{W}^*\boldsymbol{P}$, and the value equals to the $i$-th largest singular value of $\boldsymbol{W}^*$ |
| $\kappa$ | The value of $\sigma_1/\sigma_K$ |
| $\gamma$ | The value of $\prod_{i=1}^K \sigma_i/\sigma_K$ |
| $\rho$ | A fixed positive constant in $\mathbb{R}^+$ |
| $q$ | Some large constant in $\mathbb{R}$ |

where $\widetilde{\boldsymbol{w}} = [\boldsymbol{w}_{1,\Omega_1}^T, \boldsymbol{w}_{2,\Omega_2}^T, \cdots, \boldsymbol{w}_{K,\Omega_K}^T]^T \in \mathbb{R}^{\sum_j r_j}$. Here, we abuse the notation of $\hat{f}_{\mathcal{D}}$ to represent a mapping from $\mathbb{R}^{\sum_j r_j}$, instead of $\mathbb{R}^{K \times d}$ in (2), to $\mathbb{R}$. In fact, under the constrint of $\boldsymbol{W} = \boldsymbol{M} \odot \boldsymbol{W}$, the degree of freedom of $\boldsymbol{W}$ is actually $\sum_j r_j$ instead of $Kd$, and the definition in (2) is a easier way for us to present the following proofs. Therefore, the optimization problem in (3) is equivalent as

$$\min_{\widetilde{\boldsymbol{w}}}: \quad \hat{f}_{\mathcal{D}}(\widetilde{\boldsymbol{w}}). \tag{13}$$

Let us define $\widetilde{\boldsymbol{w}}^* = [\boldsymbol{w}_{1,\Omega_1}^{*T}, \boldsymbol{w}_{2,\Omega_2}^{*T}, \cdots, \boldsymbol{w}_{K,\Omega_K}^{*T}]^T \in \mathbb{R}^{\sum_j r_j}$, where $\boldsymbol{w}_j^{*T}$ is the $j$-th column of $\boldsymbol{W}^*\boldsymbol{P}$. and it is clear that $\widetilde{\boldsymbol{w}}^*$ is the global optimal to (13). Additionally, the population risk function, which is the expectation of the empirical risk function over the data $\mathcal{D}$, is defined as

$$f(\widetilde{\boldsymbol{w}}) = \mathbb{E}_{\mathcal{D}}\hat{f}_{\mathcal{D}}(\widetilde{\boldsymbol{w}}) = \mathbb{E}_{\mathcal{D}}\frac{1}{2N}\sum_{n=1}^N \left(\frac{1}{K}\sum_{j=1}^K \phi(\boldsymbol{w}_{j,\Omega_j}^T \boldsymbol{x}_{n,\Omega_j}) - y_n\right)^2$$

$$= \mathbb{E}_{\boldsymbol{x}}\frac{1}{2}\left(\frac{1}{K}\sum_{j=1}^K \phi(\boldsymbol{w}_{j,\Omega_j}^T \boldsymbol{x}_{\Omega_j}) - y\right)^2, \tag{14}$$

where $\boldsymbol{x} \in \mathbb{R}^d$ belongs to standard Gaussian distribution, and $y = g(\boldsymbol{W}^*\boldsymbol{P}^*; \boldsymbol{x})$.

Moreover, for the convenience of proofs, we use $\sigma_i$ to denote the $i$-th largest singular value of $\boldsymbol{W}^*\boldsymbol{P}$, and it is clear that $\sigma_i(\boldsymbol{W}^*\boldsymbol{P}) = \sigma_i(\boldsymbol{W}^*)$ for all $i$. Then, $\kappa$ is defined as $\sigma_1/\sigma_K$, and

$\gamma = \prod_{i=1}^{K} \sigma_i/\sigma_K$. Factor $\rho$ is defined in Property 3.2 [69] and a fixed constant for the ReLU activation function. In addition, without special descriptions, $\boldsymbol{\alpha} = [\boldsymbol{\alpha}_1^T, \boldsymbol{\alpha}_2^T, \cdots, \boldsymbol{\alpha}_K^T]^T$ stands for any unit vector that in $\mathbb{R}^{\sum_j r_j}$ with $\boldsymbol{\alpha}_j \in \mathbb{R}_j^r$. Therefore, we have

$$\|\nabla^2 \hat{f}_{\mathcal{D}}\|_2 = \max_{\boldsymbol{\alpha}} \|\boldsymbol{\alpha}^T \nabla^2 \hat{f}_{\mathcal{D}} \boldsymbol{\alpha}\|_2 = \max_{\boldsymbol{\alpha}} \Big( \sum_{j=1}^{K} \boldsymbol{\alpha}_j^T \frac{\partial \hat{f}_{\mathcal{D}}}{\partial \boldsymbol{w}_j} \Big)^2. \tag{15}$$

Finally, since we focus on order-wise analysis, some constant number will be ignored in the majority of the steps. In particular, we use $h_1(z) \gtrsim$ ( or $\lesssim, \eqsim)h_2(z)$ to denote there exists some positive constant $C$ such that $h_1(z) \geq$ ( or $\leq, =)C \cdot h_2(z)$ when $z \in \mathbb{R}$ is sufficiently large.

# B   Initialization via tensor method

In this section, we present the revised tensor initialization based on that in [69]. To reduce the dependency of input dimension from $d$ to the order of $r_{\max}$, we need to define $\widetilde{\boldsymbol{x}}$ in (20) instead of directly using $\boldsymbol{x}$ to generate the high order momentum as shown in (21) to (23). In addition, as $\boldsymbol{w}_{j,\Omega_j}$'s are different in dimensions, we need to define the corresponding augmented weights by inserting 0 such that augmented $\boldsymbol{w}_{j,\Omega_j}$ are additive in a sense. The additional notations used in presenting are summarized in Table 2, and one can skip this part if the focus is only on the local convexity analysis (Theorem 1) and convergence analysis (Theorem 2). The intuitive reasons for selecting $\widetilde{\boldsymbol{x}}$ mainly lie in two aspects: first, $\widetilde{\boldsymbol{x}}$ is much lower dimensional vector considering $r_j \ll d$; second, $\widetilde{\boldsymbol{x}}$ belongs to zero mean Gaussian distribution, which is rotational invariant and is correlate with $\phi(\boldsymbol{w}_j^{*T} \boldsymbol{x})$. Therefore, the magnitude and direction information of $\{\boldsymbol{w}_{j,\Omega_j}\}_{j=1}^K$ are separable after tensor decomposition, and the dimension of the tensors are at most in the order of $r_{\max}$.

Table 2: Table of Additional Notations for Tensor method

| Notation | Description |
|---|---|
| $\widetilde{\boldsymbol{x}}_{\Omega_j}^{(j)}$ | The argumented vector in $\mathbb{R}^{r_{\max}}$ of $\boldsymbol{x}_{\Omega_j}$ by inserting 0; defined in (16) |
| $\mathcal{F}_j$ | A linear mapping that generats a augmented vector; defined in (17) |
| $\mathcal{F}_j^{\dagger}$ | The pseudo inverse of $\mathcal{F}_j$; a linear mapping |
| $\widetilde{\boldsymbol{x}}$ | The value of $\frac{1}{\sqrt{K}} \sum_j \widetilde{\boldsymbol{x}}_{\Omega_j}^{(j)}$; |
| $\boldsymbol{u}_j^*$ | The argumented vector in $\mathbb{R}^{r_{\max}}$ of $\boldsymbol{w}_{j,\Omega_j}^*$ by inserting 0; defined in (19) |
| $\overline{\boldsymbol{u}}_j^*$ | The normalized vector of $\boldsymbol{u}_j$ as $\boldsymbol{u}_j^*/\|\boldsymbol{u}_j^*\|_2$ |
| $\widehat{\overline{\boldsymbol{u}}}_j^*$ | The estimation of the normalized vector of $\boldsymbol{u}_j^*$ |
| $\psi_1, \psi_2, \psi_3$ | Some fixed constants depends on the distribution of $\{\boldsymbol{x}_{\Omega_j}\}_{j=1}^K$ |
| $\boldsymbol{M}_1$ | A vector in $\mathbb{R}^{r_{\max}}$ defined in (21) |
| $\widehat{\boldsymbol{M}}_1$ | The estimation of $\boldsymbol{M}_1$ |
| $\boldsymbol{M}_2$ | A matrix in $\mathbb{R}^{r_{\max} \times r_{\max}}$ defined in (22) |
| $\widehat{\boldsymbol{M}}_2$ | The estimation of $\boldsymbol{M}_2$ |
| $\boldsymbol{M}_3$ | A tensor in $\mathbb{R}^{r_{\max} \times r_{\max} \times r_{\max}}$ defined in (23) |
| $\widehat{\boldsymbol{M}}_3$ | The estimation of $\boldsymbol{M}_3$ |
| $\boldsymbol{V}$ | The orthogonal matrix in $\mathbb{R}^{K \times K}$ that span the sub-space of the convex hull of $\{\boldsymbol{u}_j\}_{j=1}^K$ |
| $\widehat{\boldsymbol{V}}$ | The estimation of $\boldsymbol{V}$ |
| $\boldsymbol{M}(\widehat{\boldsymbol{V}}, \widehat{\boldsymbol{V}}, \widehat{\boldsymbol{V}})$ | A tensor in $\mathbb{R}^{K \times K \times K}$ defined in (29) |
| $\widehat{\boldsymbol{M}}(\widehat{\boldsymbol{V}}, \widehat{\boldsymbol{V}}, \widehat{\boldsymbol{V}})$ | The estimation of $\boldsymbol{M}(\widehat{\boldsymbol{V}}, \widehat{\boldsymbol{V}}, \widehat{\boldsymbol{V}})$ |
| $\boldsymbol{s}_j$ | The value of $\boldsymbol{V} \boldsymbol{u}_j^*$; a vector in $\mathbb{R}^K$ |
| $\widehat{\boldsymbol{s}}_j$ | The estimation of $\boldsymbol{s}_j$ |
| $\alpha_j$ | The value of $\|\boldsymbol{u}_j^*\|_2$; a scalar in $\mathbb{R}$ |
| $\widehat{\alpha}_j$ | The estimation of $\alpha_j$ |

First, we define a group of augmented vectors $\{\widetilde{\boldsymbol{x}}_{\widetilde{\Omega}_j}^{(j)}\}_{j=1}^K$ based on $\{\boldsymbol{x}_{\Omega_j}\}_{j=1}^K$ such that $\Omega_j \subseteq \widetilde{\Omega}_j$ with $|\widetilde{\Omega}_j| = r_{\max}$ and

$$\widetilde{x}_i^{(j)} = \begin{cases} x_i & \text{if } i \in \Omega_j \\ 0 & \text{if } i \in \widetilde{\Omega}_j/\Omega_j \end{cases}. \tag{16}$$

For notation convenience, we use $\mathcal{F}_j$ to denote the mapping from $\mathbb{R}^{r_j}$ to $\mathbb{R}^{r_{\max}}$ as

$$\mathcal{F}_j(\boldsymbol{z}) = [\boldsymbol{z}^T, \boldsymbol{0}_{(j)}^T]^T, \tag{17}$$

where $\boldsymbol{0}$ is a zero vector in $\mathbb{R}^{r_{\max}-r_j}$. Obviously, we have

$$\widetilde{\boldsymbol{x}}_{\widetilde{\Omega}_j}^{(j)} = \mathcal{F}_j(\boldsymbol{x}_{\Omega_j}). \tag{18}$$

Correspondingly, the augmented weights $\{\boldsymbol{u}_j^*\}_{j=1}^K$ are defined as

$$\boldsymbol{u}_j^* = \mathcal{F}_j(\boldsymbol{w}_{j,\Omega_j}^*) \tag{19}$$

for $j \in [K]$. The steps above guarantee the augmented weights $\boldsymbol{u}_j$'s are in the same dimension so that the high order momenta are able to characterize the directions of weights simultaneously. Additionally, we define

$$\widetilde{\boldsymbol{x}} = \frac{1}{\sqrt{K}} \sum_{j=1}^K \widetilde{\boldsymbol{x}}_{\widetilde{\Omega}_j}^{(j)}, \tag{20}$$

and corresponding high order momenta are defined in the following way instead:

$$\boldsymbol{M}_1 = \mathbb{E}_{\boldsymbol{x}}\{y\widetilde{\boldsymbol{x}}\} \in \mathbb{R}^{r_{\max}}, \tag{21}$$

$$\boldsymbol{M}_2 = \mathbb{E}_{\boldsymbol{x}}\Big[y\big(\widetilde{\boldsymbol{x}} \otimes \widetilde{\boldsymbol{x}} - \mathbb{E}_{\boldsymbol{x}}\widetilde{\boldsymbol{x}}\widetilde{\boldsymbol{x}}^T\big)\Big] \in \mathbb{R}^{r_{\max} \times r_{\max}}, \tag{22}$$

$$\boldsymbol{M}_3 = \mathbb{E}_{\boldsymbol{x}}\Big[y\big(\widetilde{\boldsymbol{x}}^{\otimes 3} - \widetilde{\boldsymbol{x}}\widetilde{\otimes}\mathbb{E}_{\boldsymbol{x}}\widetilde{\boldsymbol{x}}\widetilde{\boldsymbol{x}}^T\big)\Big] \in \mathbb{R}^{r_{\max} \times r_{\max} \times r_{\max}}, \tag{23}$$

where $\mathbb{E}_{\boldsymbol{x}}$ is the expectation over $\boldsymbol{x}$ and $\boldsymbol{z}^{\otimes 3} := \boldsymbol{z} \otimes \boldsymbol{z} \otimes \boldsymbol{z}$ defined as

$$\boldsymbol{v}\widetilde{\otimes}\boldsymbol{Z} = \sum_{i=1}^{d_2}(\boldsymbol{v} \otimes \boldsymbol{z}_i \otimes \boldsymbol{z}_i + \boldsymbol{z}_i \otimes \boldsymbol{v} \otimes \boldsymbol{z}_i + \boldsymbol{z}_i \otimes \boldsymbol{z}_i \otimes \boldsymbol{v}), \tag{24}$$

for any vector $\boldsymbol{v} \in \mathbb{R}^{d_1}$ and $\boldsymbol{Z} \in \mathbb{R}^{d_1 \times d_2}$.

Following the same calculate formulas in the Claim 5.2 [69], there exist some known constants $\psi_i, i = 1, 2, 3$, such that

$$\boldsymbol{M}_1 = \sum_{j=1}^K \psi_1 \cdot \|\boldsymbol{u}_j^*\|_2 \cdot \overline{\boldsymbol{u}}_j^*, \tag{25}$$

$$\boldsymbol{M}_2 = \sum_{j=1}^K \psi_2 \cdot \|\boldsymbol{u}_j^*\|_2 \cdot \overline{\boldsymbol{u}}_j^*\overline{\boldsymbol{u}}_j^{*T}, \tag{26}$$

$$\boldsymbol{M}_3 = \sum_{j=1}^K \psi_3 \cdot \|\boldsymbol{u}_j^*\|_2 \cdot \overline{\boldsymbol{u}}_j^{*\otimes 3}, \tag{27}$$

where $\overline{\boldsymbol{u}}_j^* = \boldsymbol{u}_j^*/\|\boldsymbol{u}_j^*\|_2$ in (21)-(23) is the normalization of $\boldsymbol{u}_j^*$.

$\boldsymbol{M}_1$, $\boldsymbol{M}_2$ and $\boldsymbol{M}_3$ can be estimated through the samples $\{(\boldsymbol{x}_n, y_n)\}_{n=1}^N$, and let $\widehat{\boldsymbol{M}}_1, \widehat{\boldsymbol{M}}_2, \widehat{\boldsymbol{M}}_3$ denote the corresponding estimates. First, we will decompose the rank-$k$ tensor $\boldsymbol{M}_3$ and obtain the $\{\overline{\boldsymbol{u}}_j^*\}_{j=1}^K$. By applying the tensor decomposition method [31] to $\widehat{\boldsymbol{M}}_3$, the outputs, denoted by $\{\widehat{\overline{\boldsymbol{u}}}_j^*\}_{j=1}^K$, are the estimations of $\{\overline{\boldsymbol{u}}_j^*\}_{j=1}^K$. Next, we will estimate $\|\boldsymbol{u}_j^*\|_2$ through solving the following optimization problem:

$$\widehat{\boldsymbol{\alpha}} = \arg\min_{\boldsymbol{\alpha} \in \mathbb{R}^K} : \quad \Big|\widehat{\boldsymbol{M}}_1 - \sum_{j=1}^K \psi_1\alpha_j\widehat{\overline{\boldsymbol{u}}}_j^*\Big|, \tag{28}$$

---
**Subroutine 1** Tensor Initialization Method
---
1: **Input:** training data $\mathcal{D} = \{(\boldsymbol{x}_n, y_n)\}_{n=1}^N$;
2: Generate augmented inputs and weights through $\mathcal{F}_j$ as shown in (17) and (19);
3: Partition $\mathcal{D}$ into three disjoint subsets $\mathcal{D}_1, \mathcal{D}_2, \mathcal{D}_3$;
4: Calculate $\widehat{\boldsymbol{M}}_1, \widehat{\boldsymbol{M}}_2$ following (21), (22) using $\mathcal{D}_1, \mathcal{D}_2$, respectively;
5: Obtain the estimate subspace $\widehat{\boldsymbol{V}}$ of $\widehat{\boldsymbol{M}}_2$;
6: Calculate $\widehat{\boldsymbol{M}}_3(\widehat{\boldsymbol{V}}, \widehat{\boldsymbol{V}}, \widehat{\boldsymbol{V}})$ through $\mathcal{D}_3$;
7: Obtain $\{\widehat{\boldsymbol{s}}_j\}_{j=1}^K$ via tensor decomposition method [31] on $\widehat{\boldsymbol{M}}_3(\widehat{\boldsymbol{V}}, \widehat{\boldsymbol{V}}, \widehat{\boldsymbol{V}})$;
8: Obtain $\widehat{\boldsymbol{\alpha}}$ by solving optimization problem (28);
9: **Return:** $\boldsymbol{w}_{j,\Omega_j}^{(0)} = \mathcal{F}_j^\dagger \big( |\widehat{\alpha}_j| \widehat{\boldsymbol{V}} \widehat{\boldsymbol{s}}_j \big), j = 1, ..., K$.
---

From (25) and (28), we know that $|\widehat{\alpha}_j|$ is the estimation of $\|\boldsymbol{u}_j^*\|_2$. Thus, $\widehat{\boldsymbol{U}}$ is given as $\big[ |\widehat{\alpha}_1| \widehat{\overline{\boldsymbol{u}}}_1^*, \cdots, |\widehat{\alpha}_j| \widehat{\overline{\boldsymbol{u}}}_j^*, \cdots, |\widehat{\alpha}_K| \widehat{\overline{\boldsymbol{u}}}_K^* \big]$.

To reduce the computational complexity of tensor decomposition, one can project $\widehat{\boldsymbol{M}}_3$ to a lower-dimensional tensor [69]. The idea is to first estimate the subspace spanned by $\{\boldsymbol{w}_j^*\}_{j=1}^K$, and let $\widehat{\boldsymbol{V}}$ denote the estimated subspace.

Moreover, we have

$$\boldsymbol{M}_3(\widehat{\boldsymbol{V}}, \widehat{\boldsymbol{V}}, \widehat{\boldsymbol{V}}) = \mathbb{E}_{\boldsymbol{x}} \Big[ y\big( (\widehat{\boldsymbol{V}}^T \widetilde{\boldsymbol{x}})^{\otimes 3} - (\widehat{\boldsymbol{V}}^T \widetilde{\boldsymbol{x}}) \widetilde{\otimes} \mathbb{E}_{\boldsymbol{x}} (\widehat{\boldsymbol{V}}^T \widetilde{\boldsymbol{x}}) (\widehat{\boldsymbol{V}}^T \widetilde{\boldsymbol{x}})^T \big) \Big] \in \mathbb{R}^{K \times K \times K}, \qquad (29)$$

Then, one can decompose the estimate $\widehat{\boldsymbol{M}}_3(\widehat{\boldsymbol{V}}, \widehat{\boldsymbol{V}}, \widehat{\boldsymbol{V}})$ to obtain unit vectors $\{\widehat{\boldsymbol{s}}_j\}_{j=1}^K \in \mathbb{R}^K$. Since $\overline{\boldsymbol{u}}^*$ lies in the subspace $\boldsymbol{V}$, we have $\boldsymbol{V} \boldsymbol{V}^T \overline{\boldsymbol{u}}_j^* = \overline{\boldsymbol{u}}_j^*$. Then, $\widehat{\boldsymbol{V}} \widehat{\boldsymbol{s}}_j$ is an estimate of $\overline{\boldsymbol{u}}_j^*$. After we obtain the estimated augmented weights $\widehat{\boldsymbol{u}}_j^*$, the estimated weights can be generated through $\widehat{\boldsymbol{w}}_{j,\Omega_j}^* = \mathcal{F}_j^\dagger(\widehat{\boldsymbol{u}}_j^*)$, where $\mathcal{F}_j^\dagger$ is the pseudo inverse of $\mathcal{F}_j$. The initialization process is summarized in Subroutine 1.

## C   Proof of Theorem 1

The main idea in proving Theorem 1 is to use triangle inequality as shown in (33) by bounding the second order derivative of the population risk function and the distance between the empirical risk and population risk functions. Lemma 3 provides the lower and upper bound for the population risk function, while Lemma 4 provides the error bound between the second order derivation of empirical risk and population risk functions.

**Lemma 2** (Weyl's inequality, [7]). *Suppose $\boldsymbol{B} = \boldsymbol{A} + \boldsymbol{E}$ be a matrix with dimension $m \times m$. Let $\lambda_i(\boldsymbol{B})$ and $\lambda_i(\boldsymbol{A})$ be the $i$-th largest eigenvalues of $\boldsymbol{B}$ and $\boldsymbol{A}$, respectively. Then, we have*

$$|\lambda_i(\boldsymbol{B}) - \lambda_i(\boldsymbol{A})| \leq \|\boldsymbol{E}\|_2, \quad \forall i \in [m]. \qquad (30)$$

**Lemma 3.** *Let $f$ be the population risk function in (14). Assume $\boldsymbol{W}$ satisfies (6), then the second-order derivative of $f$ over $\widetilde{\boldsymbol{w}}$ is bounded as*

$$\frac{(1-\varepsilon_0)\rho}{11\kappa^2 \gamma K^2} \boldsymbol{I} \leq \nabla^2 f(\widetilde{\boldsymbol{w}}) \leq \frac{7}{K} \boldsymbol{I}, \qquad (31)$$

*where $\widetilde{\boldsymbol{w}}$ only contains the elements of $\boldsymbol{W}$ with respect to the indices of non-pruned weights.*

**Lemma 4.** *Let $\hat{f}_{\mathcal{D}}$ and $f$ be the empirical and population risk function in (12) and (14), respectively, then the second-order derivative of $\hat{f}_{\mathcal{D}}$ is close to its expectation $f$ with an upper bound as:*

$$\|\nabla^2 \hat{f}_{\mathcal{D}} - \nabla^2 f\|_2 \lesssim \frac{1}{K^2} \sum_{k=1}^K \sum_{j=1}^K (1 + \delta_{j,k}) \sqrt{\frac{(r_j + r_k) \log q}{N}} \qquad (32)$$

*with probability at least $1 - q^{-r_{\min}}$.*

*Proof of Theorem 1* . Let $\hat{\lambda}_{\max}$ and $\hat{\lambda}_{\min}$ denote the largest and smallest eigenvalues of $\nabla^2 \hat{f}_{\mathcal{D}}$, respectively. Also, Let $\lambda_{\max}$ and $\lambda_{\min}$ denote the largest and smallest eigenvalues of $\nabla^2 f_{\mathcal{D}}$, respectively.

Then, from Lemma 2, we have

$$\hat{\lambda}_{\max} \leq \lambda_{\max} + \|\nabla^2 \hat{f}_{\mathcal{D}} - \nabla^2 f\|_2 \tag{33}$$

and

$$\hat{\lambda}_{\min} \geq \lambda_{\min} - \|\nabla^2 \hat{f}_{\mathcal{D}} - \nabla^2 f\|_2. \tag{34}$$

When the sample complexity satisfies $N \gtrsim \varepsilon_1^{-2} \rho^{-2} \kappa^4 \gamma^2 K^4 \left[ \frac{1}{K^2} \sum_{k=1}^{K} \sum_{j=1}^{K} (1 + \delta_{j,k}) \sqrt{r_j + r_k} \right]^2 \log q$, then from Lemma 4, we have

$$\|\nabla^2 \hat{f}_{\mathcal{D}} - \nabla^2 f\|_2 \leq \frac{\varepsilon_1 \rho}{11 \kappa^2 \gamma K^2}. \tag{35}$$

Then, from (33), (34) and (35), we have

$$\hat{\lambda}_{\max} \leq \frac{8}{K}, \tag{36}$$

and

$$\hat{\lambda}_{\min} \geq \frac{(1 - \varepsilon_0 - \varepsilon_1)\rho}{11 \kappa^2 \gamma K^2}, \tag{37}$$

which completes the proof. □

## D    Proof of Theorem 2

The major idea in proving Theorem 2 is to first characterize the gradient descent term by intermediate value theorem. Let $\widetilde{\boldsymbol{w}}^{(t)}$ be the vectorized iterate $\boldsymbol{W}^{(t)}$ with respect to the non-pruned weights, then we have

$$\begin{aligned} \nabla \hat{f}_{\Omega_t}(\widetilde{\boldsymbol{w}}^{(t)}) &= f_{\Omega_t}(\widetilde{\boldsymbol{w}}^{(t)}) + \left( \hat{f}_{\Omega_t}(\widetilde{\boldsymbol{w}}^{(t)}) - f_{\Omega_t}(\widetilde{\boldsymbol{w}}^{(t)}) \right) \\ &= \langle \nabla^2 f_{\Omega_t}(\widehat{\boldsymbol{w}}^{(t)}), \widetilde{\boldsymbol{w}}^{(t)} - \widetilde{\boldsymbol{w}}^* \rangle + \left( \hat{f}_{\Omega_t}(\widetilde{\boldsymbol{w}}^{(t)}) - f_{\Omega_t}(\widetilde{\boldsymbol{w}}^{(t)}) \right), \end{aligned} \tag{38}$$

where $\widehat{\boldsymbol{w}}^{(t)}$ lies in the convex hull of $\widetilde{\boldsymbol{w}}^{(t)}$ and $\widetilde{\boldsymbol{w}}^*$. The reason that intermediate value theorem is applied on population risk function instead of empirical risk function is the non-smoothness of the empirical risk functions. Due to the non-smoothness of ReLU activation function at zero point, the empirical risk function is not smooth, either. However, the expectation of the empirical risk function over the Gaussian input $\boldsymbol{x}$ is smooth. Hence, compared with smooth empirical risk function, i.e., neural networks equipped with sigmoid activation function, we have an additional lemma to bound $\nabla \hat{f}_{\mathcal{D}_t}$ to its expectation $\nabla f$, which is summarized in Lemma 5.

The momentum term $\beta(\boldsymbol{W}^{(t)} - \boldsymbol{W}^{(t-1)})$ plays an important role in determining the convergence rate, and the recursive rule is obtained in the following way:

$$\begin{bmatrix} \widetilde{\boldsymbol{w}}^{(t+1)} - \widetilde{\boldsymbol{w}}^* \\ \widetilde{\boldsymbol{w}}^{(t)} - \widetilde{\boldsymbol{w}}^* \end{bmatrix} = \boldsymbol{A}(\beta) \begin{bmatrix} \widetilde{\boldsymbol{w}}^{(t)} - \widetilde{\boldsymbol{w}}^* \\ \widetilde{\boldsymbol{w}}^{(t-1)} - \widetilde{\boldsymbol{w}}^* \end{bmatrix}, \tag{39}$$

where $\boldsymbol{A}(\beta)$ is a matrix with respect to the value of $\beta$ and defined in (44). Then, we know $\widetilde{\boldsymbol{w}}^{(t)}$, which is equivalent to $\boldsymbol{W}^{(t)}$, converges to the ground-truth with a linear rate which is the largest singular value of matrix $\boldsymbol{A}(\beta)$. Recall that AGD reduces to GD with $\beta = 0$, so our analysis applies to GD method as well. We are able to show the convergence rate of AGD is faster than GD by proving the largest singular value of $\boldsymbol{A}(\beta)$ is smaller than $\boldsymbol{A}(0)$ for some $\beta > 0$.

**Lemma 5.** *Let $\hat{f}_{\mathcal{D}}$ and $f$ be the empirical and population risk function in (12) and (14), respectively, then the first-order derivative of $\hat{f}_{\mathcal{D}}$ is close to its expectation $f$ with an upper bound as:*

$$\|\nabla \hat{f}_{\mathcal{D}}(\widetilde{\boldsymbol{w}}) - \nabla f(\widetilde{\boldsymbol{w}})\|_2 \lesssim \frac{1}{K^2} \sum_{k=1}^{K} \sum_{j=1}^{K} (1 + \delta_{j,k}) \sqrt{\frac{r_k \log q}{N}} \|\widetilde{\boldsymbol{w}} - \widetilde{\boldsymbol{w}}^*\|_2 + \frac{1}{K} \sum_{k=1}^{K} \sqrt{\frac{r_k \log q}{N}} \cdot |\xi| \tag{40}$$

*with probability at least $1 - q^{-r_{\min}}$, where $\widetilde{\boldsymbol{w}}$ only contains the elements of $\boldsymbol{W}$ with respect to the indices of non-pruned weights.*

*Proof of Theorem 2.* Since $\|\boldsymbol{W}^{(t)} - \boldsymbol{W}^*\|_F = \|\widetilde{\boldsymbol{w}}^{(t)} - \widetilde{\boldsymbol{w}}^*\|_2$, we can explore the converges of $\{\widetilde{\boldsymbol{w}}^{(t)}\}_{t=1}^T$ instead. Recall that

$$
\begin{aligned}
\widetilde{\boldsymbol{w}}^{(t+1)} =& \widetilde{\boldsymbol{w}}^{(t)} - \eta \nabla \hat{f}_{\mathcal{D}_t}(\widetilde{\boldsymbol{w}}^{(t)}) + \beta(\widetilde{\boldsymbol{w}}^{(t)} - \widetilde{\boldsymbol{w}}^{(t-1)}) \\
=& \widetilde{\boldsymbol{w}}^{(t)} - \eta \nabla f(\widetilde{\boldsymbol{w}}^{(t)}) + \beta\big(\widetilde{\boldsymbol{w}}^{(t)} - \widetilde{\boldsymbol{w}}^{(t-1)}\big) \\
& + \eta\big(\nabla f(\widetilde{\boldsymbol{w}}^{(t)}) - \nabla \hat{f}_{\mathcal{D}_t}(\widetilde{\boldsymbol{w}}^{(t)})\big).
\end{aligned}
\tag{41}
$$

Since $\nabla^2 f$ is a smooth function, by the intermediate value theorem, we have

$$
\begin{aligned}
\widetilde{\boldsymbol{w}}^{(t+1)} =& \widetilde{\boldsymbol{w}}^{(t)} - \eta \nabla^2 f(\widehat{\boldsymbol{w}}^{(t)})(\widetilde{\boldsymbol{w}}^{(t)} - \widetilde{\boldsymbol{w}}^*) + \beta(\widetilde{\boldsymbol{w}}^{(t)} - \widetilde{\boldsymbol{w}}^{(t-1)}) \\
& + \eta\big(\nabla f(\widetilde{\boldsymbol{w}}^{(t)}) - \nabla \hat{f}_{\mathcal{D}_t}(\widetilde{\boldsymbol{w}}^{(t)})\big),
\end{aligned}
\tag{42}
$$

where $\widehat{\boldsymbol{w}}^{(t)}$ lies in the convex hull of $\widetilde{\boldsymbol{w}}^{(t)}$ and $\widetilde{\boldsymbol{w}}^*$.
Next, we have

$$
\begin{aligned}
\begin{bmatrix} \widetilde{\boldsymbol{w}}^{(t+1)} - \widetilde{\boldsymbol{w}}^* \\ \widetilde{\boldsymbol{w}}^{(t)} - \widetilde{\boldsymbol{w}}^* \end{bmatrix} =& \begin{bmatrix} \boldsymbol{I} - \eta \nabla^2 f(\widehat{\boldsymbol{w}}^{(t)}) + \beta \boldsymbol{I} & \beta \boldsymbol{I} \\ \boldsymbol{I} & 0 \end{bmatrix} \begin{bmatrix} \widetilde{\boldsymbol{w}}^{(t)} - \widetilde{\boldsymbol{w}}^* \\ \widetilde{\boldsymbol{w}}^{(t-1)} - \widetilde{\boldsymbol{w}}^* \end{bmatrix} \\
& + \eta \begin{bmatrix} \nabla f(\widetilde{\boldsymbol{w}}^{(t)}) - \nabla \hat{f}_{\mathcal{D}_t}(\widetilde{\boldsymbol{w}}^{(t)}) \\ 0 \end{bmatrix}
\end{aligned}
\tag{43}
$$

Let

$$
\boldsymbol{A}(\beta) = \begin{bmatrix} \boldsymbol{I} - \eta \nabla^2 f(\widehat{\boldsymbol{w}}^{(t)}) + \beta \boldsymbol{I} & \beta \boldsymbol{I} \\ \boldsymbol{I} & 0 \end{bmatrix},
\tag{44}
$$

so we have

$$
\left\| \begin{bmatrix} \widetilde{\boldsymbol{w}}^{(t+1)} - \widetilde{\boldsymbol{w}}^* \\ \widetilde{\boldsymbol{w}}^{(t)} - \widetilde{\boldsymbol{w}}^* \end{bmatrix} \right\|_2 = \|\boldsymbol{A}(\beta)\|_2 \left\| \begin{bmatrix} \widetilde{\boldsymbol{w}}^{(t)} - \widetilde{\boldsymbol{w}}^* \\ \widetilde{\boldsymbol{w}}^{(t-1)} - \widetilde{\boldsymbol{w}}^* \end{bmatrix} \right\|_2 + \eta \left\| \begin{bmatrix} \nabla f(\widetilde{\boldsymbol{w}}^{(t)}) - \nabla \hat{f}_{\mathcal{D}_t}(\widetilde{\boldsymbol{w}}^{(t)}) \\ 0 \end{bmatrix} \right\|_2.
\tag{45}
$$

From Lemma 5, we know that

$$
\begin{aligned}
\eta \left\| \nabla f(\widetilde{\boldsymbol{w}}^{(t)}) - \nabla \hat{f}_{\mathcal{D}_t}(\widetilde{\boldsymbol{w}}^{(t)}) \right\|_2 \leq & \frac{C_5 \eta}{K^2} \sum_{k=1}^K \sum_{j=1}^K (1 + \delta_{j,k}) \sqrt{\frac{r_k \log q}{N_t}} \|\widetilde{\boldsymbol{w}} - \widetilde{\boldsymbol{w}}^*\|_2 \\
& + \frac{C_5 \eta}{K} \sum_{k=1}^K \sqrt{\frac{r_k \log q}{N_t}} \cdot |\xi|
\end{aligned}
\tag{46}
$$

for some constant $C_5 > 0$. Then, we have

$$
\begin{aligned}
\|\widetilde{\boldsymbol{w}}^{(t+1)} - \widetilde{\boldsymbol{w}}^*\|_2 \leq & \left( \|\boldsymbol{A}(\beta)\|_2 + \frac{C_5 \eta}{K^2} \sum_{k=1}^K \sum_{j=1}^K (1 + \delta_{j,k}) \sqrt{\frac{r_k \log q}{N_t}} \right) \|\widetilde{\boldsymbol{w}}^{(t)} - \widetilde{\boldsymbol{w}}^*\|_2 \\
& + \frac{C_5 \eta}{K} \sum_{k=1}^K \sqrt{\frac{r_k \log q}{N_t}} \cdot |\xi| \\
:=& \nu(\beta) \|\widetilde{\boldsymbol{w}}^{(t)} - \widetilde{\boldsymbol{w}}^*\|_2 + \frac{C_5 \eta}{K} \sum_{k=1}^K \sqrt{\frac{r_k \log q}{N_t}} \cdot |\xi|.
\end{aligned}
\tag{47}
$$

Let $\nabla^2 f(\widehat{\boldsymbol{w}}^{(t)}) = \boldsymbol{S} \boldsymbol{\Lambda} \boldsymbol{S}^T$ be the eigendecomposition of $\nabla^2 f(\widehat{\boldsymbol{w}}^{(t)})$. Then, we define

$$
\boldsymbol{A}(\beta) := \begin{bmatrix} \boldsymbol{S}^T & 0 \\ 0 & \boldsymbol{S}^T \end{bmatrix} \boldsymbol{A}(\beta) \begin{bmatrix} \boldsymbol{S} & 0 \\ 0 & \boldsymbol{S} \end{bmatrix} = \begin{bmatrix} \boldsymbol{I} - \eta \boldsymbol{\Lambda} + \beta \boldsymbol{I} & \beta \boldsymbol{I} \\ \boldsymbol{I} & 0 \end{bmatrix}
\tag{48}
$$

Since $\begin{bmatrix} \boldsymbol{S} & 0 \\ 0 & \boldsymbol{S} \end{bmatrix} \begin{bmatrix} \boldsymbol{S}^T & 0 \\ 0 & \boldsymbol{S}^T \end{bmatrix} = \begin{bmatrix} \boldsymbol{I} & 0 \\ 0 & \boldsymbol{I} \end{bmatrix}$, we know $\boldsymbol{A}(\beta)$ and $\boldsymbol{A}(\beta)$ share the same eigenvalues.
Let $\lambda_i$ be the $i$-th eigenvalue of $\nabla^2 f(\widehat{\boldsymbol{w}}^{(t)})$, then the corresponding $i$-th eigenvalue of (48), denoted by $\delta_i(\beta)$, satisfies

$$
\nu_i^2 - (1 - \eta \lambda_i + \beta)\delta_i + \beta = 0.
\tag{49}
$$

Then, we have

$$\delta_i(\beta) = \frac{(1 - \eta\lambda_i + \beta) + \sqrt{(1 - \eta\lambda_i + \beta)^2 - 4\beta}}{2}, \tag{50}$$

and

$$|\delta_i(\beta)| = \begin{cases} \sqrt{\beta}, & \text{if} \quad \beta \geq \left(1 - \sqrt{\eta\lambda_i}\right)^2, \\ \frac{1}{2}\left|(1 - \eta\lambda_i + \beta) + \sqrt{(1 - \eta\lambda_i + \beta)^2 - 4\beta}\right|, \text{otherwise.} \end{cases} \tag{51}$$

Note that the other root of (49) is abandoned because the root in (50) is always larger than or at least equal to the other root with $|1 - \eta\lambda_i| < 1$. By simple calculation, we have

$$\delta_i(0) > \delta_i(\beta), \quad \text{for} \quad \forall\beta \in \left(0, (1 - \eta\lambda_i)^2\right), \tag{52}$$

and specifically, $\delta_i$ achieves the minimum $\delta_i^* = |1 - \sqrt{\eta\lambda_i}|$ when $\beta = \left(1 - \sqrt{\eta\lambda_i}\right)^2$.

Let us first assume $\widetilde{\boldsymbol{w}}^{(t)}$ satisfies (6), then from Lemma 3, we know that

$$0 < \frac{(1 - \varepsilon_0)}{11\kappa^2\gamma K^2} \leq \lambda_i \leq \frac{7}{K}$$

provided that $N_t \gtrsim \varepsilon_0^{-2}\rho^{-1}\kappa^2\gamma K^3\left[\frac{1}{K^2}\sum_j\sum_k(1 + \delta_{j,k})\sqrt{r_k + r_j}\right]^2\log q$. Let $\gamma_1 = \frac{\rho(1-\varepsilon_0)}{11\kappa^2\gamma K^2}$ and $\gamma_2 = \frac{7}{K}$. If we choose $\beta$ such that

$$\beta^* = \max\left\{(1 - \sqrt{\eta\gamma_1})^2, (1 - \sqrt{\eta\gamma_2})^2\right\}, \tag{53}$$

then we have $\beta \geq (1 - \sqrt{\eta\lambda_i})^2$ for any $i$ and $\delta_i = \max\left\{|1 - \sqrt{\eta\gamma_1}|, |1 - \sqrt{\eta\gamma_2}|\right\}$ for any $i$.

Let $\eta = \frac{1}{2\gamma_2}$, then $\beta^*$ equals to $\left(1 - \sqrt{\frac{\gamma_1}{2\gamma_2}}\right)^2$. Then, for any $\varepsilon_0 \in (0, \frac{1}{2})$ we have

$$\begin{aligned} \|\boldsymbol{A}(\beta^*)\|_2 = \max_i \delta_i(\beta^*) = 1 - \sqrt{\frac{\gamma_1}{2\gamma_2}} &= 1 - \sqrt{\frac{1 - \varepsilon_0}{154\rho^{-1}\kappa^2\gamma K}} \\ &\leq 1 - \frac{1 - 3/4 \cdot \varepsilon_0}{\sqrt{154\rho^{-1}\kappa^2\gamma K}}. \end{aligned} \tag{54}$$

Then, let

$$\frac{C_5\eta}{K^2}\sum_{k=1}^K\sum_{j=1}^K(1 + \delta_{j,k})\sqrt{\frac{r_k\log q}{N_t}} \leq \frac{\varepsilon_0}{4\sqrt{154\rho^{-1}\kappa^2\gamma K}}, \tag{55}$$

we need $N_t \gtrsim \varepsilon_0^{-2}\rho^{-1}\kappa^2\gamma K^3\left[\frac{1}{K^2}\sum_j\sum_k(1 + \delta_{j,k})\sqrt{r_k}\right]^2\log q$.

Combine (54) and (55), we have

$$\nu(\beta^*) \leq 1 - \frac{1 - \varepsilon_0}{\sqrt{154\rho^{-1}\kappa^2\gamma K}}. \tag{56}$$

While let $\beta = 0$, we have

$$\nu(0) \geq \|\boldsymbol{A}(0)\|_2 = 1 - \frac{1 - \varepsilon_0}{154\rho^{-1}\kappa^2\gamma K} \tag{57}$$

and

$$\nu(0) \leq 1 - \frac{1 - 2\varepsilon_0}{154\rho^{-1}\kappa^2\gamma K} \tag{58}$$

if $N_t \gtrsim \varepsilon_0^{-2}\rho^{-1}\kappa^2\gamma K^4\left[\frac{1}{K^2}\sum_j\sum_k(1 + \delta_{j,k})\sqrt{r_k + r_j}\right]^2\log q$.

In conclusion, with $\eta = \frac{1}{2\gamma_2}$ and $\beta = \left(1 - \frac{\gamma_1}{2\gamma_2}\right)^2$, we have

$$\|\widetilde{\boldsymbol{w}}^{(t+1)} - \widetilde{\boldsymbol{w}}^*\|_2 \leq \left(1 - \frac{1 - \varepsilon_0}{\sqrt{154\kappa^2\gamma K}}\right)\|\widetilde{\boldsymbol{w}}^{(t)} - \widetilde{\boldsymbol{w}}^*\|_2 + \frac{C\eta}{K}\sum_{k=1}^K\sqrt{\frac{r_k\log q}{N_t}}|\xi|. \tag{59}$$

if $\widetilde{\boldsymbol{w}}^{(t+1)}$ satisfies (6) and $N_t \gtrsim \varepsilon_0^{-2}\rho^{-1}\kappa^2\gamma K^4\left[\frac{1}{K^2}\sum_j\sum_k(1 + \delta_{j,k})\sqrt{r_k + r_j}\right]^2\log q$.

Then, we can start mathematical induction of (59) over $t$.

**Base case**: (6) holds for $\widetilde{\boldsymbol{w}}^{(0)}$ naturally from the assumption in Theorem 2. Since (6) holds and the number of samples exceeds the required bound in (59), we have (59) holds for $t = 0$.

**Induction step**: Assume (59) holds for $t$, to make sure the mathematical induction of (59) holds, we need $\widetilde{\boldsymbol{w}}^{(t+1)}$ satisfies (6). That is

$$\sum_{k=1}^{K} \frac{\eta}{K} \sqrt{\frac{r_k \log q}{N_t}} \lesssim \frac{1 - \varepsilon_0}{\sqrt{132 \kappa^2 \gamma K}} \cdot \frac{\varepsilon_0 \sigma_K}{44 \kappa^2 \gamma K^2}. \tag{60}$$

Hence, we need

$$N_t \gtrsim \varepsilon_0^{-2} \kappa^8 \gamma^3 K^6 \Big( \frac{1}{K} \sum_k \sqrt{r_k} \Big)^2 \log q. \tag{61}$$

In addition, with (6) and (59) hold for all $t \leq T$, the following equation

$$\left\| \begin{bmatrix} \widetilde{\boldsymbol{w}}^{(t+1)} - \widetilde{\boldsymbol{w}}^* \\ \widetilde{\boldsymbol{w}}^{(t)} - \widetilde{\boldsymbol{w}}^* \end{bmatrix} \right\|_\infty = \|\boldsymbol{A}(\beta)\|_2 \left\| \begin{bmatrix} \widetilde{\boldsymbol{w}}^{(t)} - \widetilde{\boldsymbol{w}}^* \\ \widetilde{\boldsymbol{w}}^{(t-1)} - \widetilde{\boldsymbol{w}}^* \end{bmatrix} \right\|_\infty + \eta \left\| \begin{bmatrix} \nabla f(\widetilde{\boldsymbol{w}}^{(t)}) - \nabla \hat{f}_{\mathcal{D}_t}(\widetilde{\boldsymbol{w}}^{(t)}) \\ 0 \end{bmatrix} \right\|_\infty \tag{62}$$

holds as well, and $\|\boldsymbol{A}(\beta)\|_2$ is bounded by $\nu(\beta)$. Hence, (59) also holds in infinity norm as

$$\|\widetilde{\boldsymbol{w}}^{(t+1)} - \widetilde{\boldsymbol{w}}^*\|_\infty \leq \Big( 1 - \frac{1 - \varepsilon_0}{\sqrt{154 \kappa^2 \gamma K}} \Big) \|\widetilde{\boldsymbol{w}}^{(t)} - \widetilde{\boldsymbol{w}}^*\|_\infty + 2C\eta \sqrt{\frac{r \log q}{N_t}} |\xi|. \tag{63}$$

In conclusion, when $N_t \gtrsim \varepsilon_0^{-2} \kappa^8 \gamma^3 K^6 \Big( \frac{1}{K^2} \sum_k \sum_j (1 + \delta_{j,k}) \sqrt{r_k + r_j} \Big)^2 \log d$, we know that (59) holds for all $1 \leq t \leq T$ with probability at least $1 - K^2 T \cdot q^{-r_{\min}}$. By simple calculation, we can obtain

$$\|\widetilde{\boldsymbol{w}}^{(T)} - \widetilde{\boldsymbol{w}}^*\|_2 \leq \Big( 1 - \frac{1 - \varepsilon_0}{\sqrt{132 \kappa^2 \gamma K}} \Big)^T \|\widetilde{\boldsymbol{w}}^{(0)} - \widetilde{\boldsymbol{w}}^*\|_2 + \frac{C}{K} \sum_{k=1}^{K} \sqrt{\frac{\kappa^2 \gamma K^2 r_k \log q}{N_t}} \cdot |\xi|. \tag{64}$$

for some constant $C > 0$.

$\square$

# E   Obtaining a proper learner network via magnitude pruning

In this section, we show that how one can combine Algorithm 1 and magnitude pruning to find a proper learner network such that $r_j \geq r_j^*$ and $\Omega_j \supseteq \Omega_j^*$ from a fully-connected network under some assumptions. Suppose the number of samples is at least $\Omega\big(K^6 d \log q \log(1/\varepsilon)\big)$, we train directly on the fully-connected dense network using Algorithm 1. The number of iteration in line 2 of Algorithm 1 is set as $T_1 = \Theta\big(\log(2\widehat{W}_{\max}/\widehat{W}_{\min})\big)$, where $\widehat{W}_{\min}$ and $\widehat{W}_{\max}$ denote the smallest and largest value of $\boldsymbol{W}^*$, respectively. From (63), after $T_1$ iterations, the returned model, denote by $\boldsymbol{W}^{(T_1)}$, is close to the ground-truth $\boldsymbol{W}^*$. Specifically, if $\boldsymbol{W}_{i,j}^* \neq 0$ and $\boldsymbol{W}_{i',j'}^* = 0$, then $\boldsymbol{W}_{i,j}^{(T_1)} > \boldsymbol{W}_{i',j'}^{(T_1)}$ for any $i, j, i', j'$. Then we sort the weights based on their absolute values and prune them sequentially starting from the least absolute value. As long as the ratio of pruned weights is at most $\Big( 1 - \frac{\sum_j r_j}{Kd} \Big)$, all the weights are removed correctly, leading to a proper learner network. In fact, if we remove exactly $1 - \frac{\sum_j r_j}{Kd}$ fraction of weights, the pruned network has the same architecture as the oracle network.

Specifically, suppose $\boldsymbol{M}^{(t)}$ to denote the mask matrix by truncating the smallest $\Big( 1 - \frac{\sum_j r_j}{Kd} \Big)$ fraction of entries in iterate $\boldsymbol{W}^{(t)}$. Let $\boldsymbol{M}^*$ denote the ground-truth mask matrix for the oracle network, the following corollary holds from Theorem 2.

**Corollary 1.** *Suppose the noise $|\xi| \leq \widehat{W}^*_{\min}$ and the number of samples satisfies $N = \Omega\big(K^6 d \log q \log(1/\varepsilon)\big)$. Let $\{\boldsymbol{W}^{(t_1)}\}_{t_1=1}^{T_1}$ be the iterates generated from Algorithm 1 by setting $r = d$. Then, for any $T_1 \geq \log(\widehat{W}^*_{\max}/\widehat{W}^*_{\min})$, we have*

$$\boldsymbol{M}^{(T_1)} = \boldsymbol{M}^*. \tag{65}$$

*Proof of Corollary 1.* If we train on the dense network, from (63), we know that

$$\|\boldsymbol{W}^{(t+1)} - \boldsymbol{W}^*\|_\infty \leq \left(1 - \frac{1-\varepsilon_0}{\sqrt{154\kappa^2\gamma K}}\right)\|\boldsymbol{W}^{(t)} - \boldsymbol{W}^*\|_\infty + 2C\eta\sqrt{\frac{d\log q}{N_t}}|\xi|. \tag{66}$$

Hence, we have

$$\|\boldsymbol{W}^{(T_1)} - \boldsymbol{W}^*\|_\infty \leq \left(1 - \frac{1-\varepsilon_0}{\sqrt{154\kappa^2\gamma K}}\right)^{T_1}\|\boldsymbol{W}^{(0)} - \boldsymbol{W}^*\|_\infty + 2C\eta\sqrt{\frac{d\log q}{N_t}}|\xi|. \tag{67}$$

With $T_1 \geq \log(2\widehat{W}^*_{\max}/\widehat{W}^*_{\min})$, we have

$$\left(1 - \frac{1-\varepsilon_0}{\sqrt{154\kappa^2\gamma K}}\right)^{T_1}\|\boldsymbol{W}^{(0)} - \boldsymbol{W}^*\|_\infty \leq \frac{1}{4}\widehat{W}^*_{\min} \cdot \frac{\|\boldsymbol{W}^{(0)} - \boldsymbol{W}^*\|_\infty}{\|\boldsymbol{W}^*\|_\infty} \leq \frac{1}{4}\widehat{W}^*_{\min}. \tag{68}$$

Since $N = \Omega\big(K^6 d \log q \log(1/\varepsilon)\big)$ and $|\xi| \leq \widehat{W}^*_{\min}$, we have

$$2C\eta\sqrt{\frac{d\log q}{N_t}}|\xi| \leq \frac{1}{4}\widehat{W}^*_{\min}. \tag{69}$$

From (68) and (69), we know that

$$\|\boldsymbol{W}^{(T_1)} - \boldsymbol{W}^*\|_\infty \leq \frac{1}{2}\widehat{W}^*_{\min}. \tag{70}$$

Therefore, for any entry in $W^{(T_1)}_{i,j}$, if the corresponding entry in augmented ground-truth weights $\boldsymbol{W}^*$ is zero, we have

$$|W^{(T_1)}_{i,j}| \leq \frac{1}{2}\widehat{W}^*_{\min}; \tag{71}$$

if the corresponding entry in $\boldsymbol{W}^*$ is non-zero, we have

$$|W^{(T_1)}_{i,j}| \geq |\widehat{W}^*_{i,j}| - \frac{1}{2}\widehat{W}^*_{\min} \geq \frac{1}{2}\widehat{W}^*_{\min}. \tag{72}$$

As we know that there are only $\sum_j r_j/(Kd)$ fraction of non-zero weights in the ground-truth model, $\boldsymbol{M}^{(T_1)} = \boldsymbol{M}^*$ holds. $\qquad\square$

## F   Proof of Lemma 1

Instead of providing the proof for Lemma 1, we turn to prove a more general bound for the performance of tensor initialization method as shown in Lemma 6. One can easily verify that Lemma 1 holds naturally from Lemma 6.

Recall that in Appendix B, the estimation of $\boldsymbol{w}^*_{j,\Omega_j}$ are converted into estimate the augmented vector $\boldsymbol{u}^*_j$. Further, the estimation of $\boldsymbol{u}^*_j$ are divided into estimating three parts: (1) the estimation of the magnitude of $\boldsymbol{u}^*_j$, which is denoted as $\widehat{\alpha}_j$; (2) the estimation of the subspace of $\boldsymbol{u}^*_j$, which is denoted as $\widehat{\boldsymbol{V}}$; (3) the estimation of the representation of $\boldsymbol{u}^*_j$ on subspace $\boldsymbol{V}$, which is denoted as $\hat{\boldsymbol{s}}_j$. Hence, the major idea of proving Lemma 6 is to characterize the difference of these three estimations to its ground-truth, which are summarized in Lemmas 7, 8 and 9, respectively.

**Lemma 6.** *Assume the noise level $|\xi| \leq K\sigma_1$ and the number of samples $N \gtrsim \kappa^8 K^5 r_{\max} \log^6 q$ with some large constant q, the tensor initialization method in Subroutine 1 outputs $\boldsymbol{W}^{(0)}$ such that*

$$\|\boldsymbol{W}^{(0)} - \boldsymbol{W}^*\|_2 \lesssim \kappa^6 \sqrt{\frac{Kr_{\max}\log q}{N}}(\sigma_1 + |\xi|) \tag{73}$$

*with probability at least $1 - q^{-r_{\max}}$.*

## F.1 Proof of Lemma 6

**Lemma 7.** *Suppose $M_2$ is defined as in (22) and $\widehat{M}_2$ is the estimation of $M_2$ by samples $\mathcal{D} = \{(\boldsymbol{x}_n, y_n)\}_{n=1}^N$. Then, with probability $1 - q^{-r_{\max}}$, we have*

$$\|\widehat{M}_2 - M_2\| \lesssim \sqrt{\frac{r_{\max} \log q}{N}}(\sigma_1 + |\xi|), \tag{74}$$

*provided that $N \gtrsim r_{\max} \log^4 q$.*

**Lemma 8.** *Let $\widehat{V}$ be generated by step 4 in Subroutine 1. Suppose $M_3(\widehat{V}, \widehat{V}, \widehat{V})$ is defined as in (29) and $\widehat{M}_3(\widehat{V}, \widehat{V}, \widehat{V})$ is the estimation of $M_3(\widehat{V}, \widehat{V}, \widehat{V})$ by samples $\mathcal{D} = \{(\boldsymbol{x}_n, y_n)\}_{n=1}^N$. Further, we assume $V \in \mathbb{R}^{r \times K}$ is an orthogonal basis of $\{\boldsymbol{u}_j^*\}_{j=1}^K$ and satisfies $\|VV^T - \widehat{V}\widehat{V}^T\| \leq 1/4$. Then, provided that $N \gtrsim K^5 \log^6 d$, with probability at least $1 - q^{-r_{\max}}$, we have*

$$\|\widehat{M}_3(\widehat{V}, \widehat{V}, \widehat{V}) - M_3(\widehat{V}, \widehat{V}, \widehat{V})\| \lesssim \sqrt{\frac{\log q}{N}}(\sigma_1 + |\xi|). \tag{75}$$

**Lemma 9.** *Suppose $M_1$ is defined as in (21) and $\widehat{M}_1$ is the estimation of $M_1$ by samples $\mathcal{D} = \{(\boldsymbol{x}_n, y_n)\}_{n=1}^N$. Then, with probability $1 - q^{-r_{\max}}$, we have*

$$\|\widehat{M}_1 - M_1\| \lesssim \sqrt{\frac{r_{\max} \log q}{N}}(\sigma_1 + |\xi|) \tag{76}$$

*provided that $N \gtrsim r_{\max} \log^4 d$.*

**Lemma 10** ([53], Theorem 1.6). *Consider a finite sequence $\{\boldsymbol{Z}_k\}$ of independent, random matrices with dimensions $d_1 \times d_2$. Assume that such random matrix satisfies*

$$\mathbb{E}(\boldsymbol{Z}_k) = 0 \quad and \quad \|\boldsymbol{Z}_k\| \leq R \quad almost \ surely.$$

*Define*

$$\delta^2 := \max\left\{\left\|\sum_k \mathbb{E}(\boldsymbol{Z}_k \boldsymbol{Z}_k^*)\right\|, \left\|\sum_k \mathbb{E}(\boldsymbol{Z}_k^* \boldsymbol{Z}_k)\right\|\right\}.$$

*Then for all $t \geq 0$, we have*

$$Prob\left\{\left\|\sum_k \boldsymbol{Z}_k\right\| \geq t\right\} \leq (d_1 + d_2)\exp\left(\frac{-t^2/2}{\delta^2 + Rt/3}\right).$$

**Lemma 11** ([69], Lemma E.6). *Let $V \in \mathbb{R}^{r \times K}$ be an orthogonal basis of $\widetilde{\boldsymbol{w}}^*$ and $\widehat{V}$ be generated by step 4 in Subroutine 1. Assume $\|\widehat{M}_2 - M_2\|_2 \leq \sigma_K(M_2)/10$. Then, we have*

$$\|VV^T - \widehat{V}\widehat{V}^T\|_2 \leq \frac{\|M_2 - \widehat{M}_2\|}{\sigma_K(M_2)}. \tag{77}$$

**Lemma 12** ([69], Lemmas E.13 and E.14). *Let $V \in \mathbb{R}^{r \times K}$ be an orthogonal basis of $\widetilde{\boldsymbol{w}}^*$ and $\widehat{V}$ be generated by step 4 in Subroutine 1. Assume $M_1$ can be written in the form of (25) with some constant $\psi_1$, and let $\widehat{M}_1$ be the estimation of $M_1$ by samples $\mathcal{D} = \{\boldsymbol{x}_n, y_n\}_{n=1}^N$. Let $\widehat{\boldsymbol{\alpha}}$ be the optimal solutions of (28) with $\widehat{\widetilde{\boldsymbol{u}}}_j^* = \widehat{V}\widehat{\boldsymbol{s}}_j$. Then, for each $j \in \{1, 2, \cdots, K\}$, if*

$$T_1 := \|VV^T - \widehat{V}\widehat{V}^T\|_2 \leq \frac{1}{\kappa^2\sqrt{K}},$$

$$T_2 := \|\widehat{\boldsymbol{u}}_j^* - \widehat{V}^T\widehat{\boldsymbol{s}}_j\|_2 \leq \frac{1}{\kappa^2\sqrt{K}}, \tag{78}$$

$$T_3 := \|\widehat{M}_1 - M_1\|_2 \leq \frac{1}{4}\|M_1\|_2,$$

*then we have*

$$\left|\alpha_j^* - \widehat{\alpha}_j\right| \leq \left(\kappa^4 K^{\frac{3}{2}}(T_1 + T_2) + \kappa^2 K^{\frac{1}{2}}T_3\right)|\alpha_j^*|, \tag{79}$$

*where $\alpha_j^* = \|\boldsymbol{u}_j^*\|_2$.*

*Proof of Lemma 1.* By simple calculation, we have

$$
\begin{aligned}
&\|\boldsymbol{u}_j^* - |\widehat{\alpha}_j|\widehat{\boldsymbol{V}}\widehat{\boldsymbol{s}}_j\|_2 \\
\leq& \left\|\boldsymbol{u}_j^* - \|\boldsymbol{u}_j^*\|_2\widehat{\boldsymbol{V}}\widehat{\boldsymbol{s}}_j + \|\boldsymbol{u}_j^*\|_2\widehat{\boldsymbol{V}}\widehat{\boldsymbol{s}}_j - |\widehat{\alpha}_j|\widehat{\boldsymbol{V}}\widehat{\boldsymbol{s}}_j\right\|_2 \\
\leq& \left\|\boldsymbol{u}_j^* - \|\boldsymbol{u}_j^*\|_2\widehat{\boldsymbol{V}}\widehat{\boldsymbol{s}}_j\right\|_2 + \left\|\|\boldsymbol{u}_j^*\|_2\widehat{\boldsymbol{V}}\widehat{\boldsymbol{s}}_j - |\widehat{\alpha}_j|\widehat{\boldsymbol{V}}\widehat{\boldsymbol{s}}_j\right\|_2 \\
\leq& \|\boldsymbol{u}_j^*\|_2\|\overline{\boldsymbol{u}}_j^* - \widehat{\boldsymbol{V}}\widehat{\boldsymbol{s}}_j\|_2 + \left|\|\boldsymbol{u}_j^*\|_2 - |\widehat{\alpha}_j|\right|\|\widehat{\boldsymbol{V}}\widehat{\boldsymbol{s}}_j\|_2 \\
\leq& \sigma_1\big(\|\overline{\boldsymbol{u}}_j^* - \widehat{\boldsymbol{V}}\widehat{\boldsymbol{V}}^T\overline{\boldsymbol{u}}_j^*\|_2 + \|\widehat{\boldsymbol{V}}^T\overline{\boldsymbol{u}}_j^* - \widehat{\boldsymbol{s}}_j\|_2\big) + \left|\|\boldsymbol{u}_j^*\|_2 - |\widehat{\alpha}_j|\right| \\
:=& \sigma_1\big(I_1 + I_2\big) + I_3.
\end{aligned}
\tag{80}
$$

From Lemma 11, we have

$$
I_1 = \|\overline{\boldsymbol{u}}_j^* - \widehat{\boldsymbol{V}}\widehat{\boldsymbol{V}}^T\overline{\boldsymbol{u}}_j^*\|_2 \leq \|\boldsymbol{V}\boldsymbol{V}^T - \widehat{\boldsymbol{V}}\widehat{\boldsymbol{V}}^T\|_2 \leq \frac{\|\widehat{\boldsymbol{M}}_2 - \boldsymbol{M}_2\|_2}{\sigma_K(\boldsymbol{M}_2)},
\tag{81}
$$

where the last inequality comes from Lemma 7. Then, from (26), we know that

$$
\sigma_K(\boldsymbol{M}_2) \lesssim \min_{1\leq j\leq K}\|\boldsymbol{u}_j^*\|_2 = \min_{1\leq j\leq K}\|\widetilde{\boldsymbol{w}}_{j,\Omega_j}^*\|_2 \lesssim \sigma_K.
\tag{82}
$$

From Theorem 3 in [31], we have

$$
I_2 = \|\widehat{\boldsymbol{V}}^T\overline{\boldsymbol{u}}_j^* - \widehat{\boldsymbol{s}}_j\|_2 \lesssim \frac{\kappa}{\sigma_K}\|\widehat{\boldsymbol{M}}_3(\widehat{\boldsymbol{V}},\widehat{\boldsymbol{V}},\widehat{\boldsymbol{V}}) - \boldsymbol{M}_3(\widehat{\boldsymbol{V}},\widehat{\boldsymbol{V}},\widehat{\boldsymbol{V}})\|_2.
\tag{83}
$$

To guarantee the condition (78) in Lemma 12 hold, according to Lemmas 7 and 8, we need $N \gtrsim \kappa^3 K r_{\max}\log q$. Then, from Lemma 12, we have

$$
I_3 = \left(\kappa^4 K^{3/2}(I_1 + I_2) + \kappa^2 K^{1/2}\|\widehat{\boldsymbol{M}}_1 - \boldsymbol{M}_1\|\right)\sigma_1.
\tag{84}
$$

When $r_{\max} \gg K$, according to Lemmas 7, 8 and 9, we have

$$
\left\|\boldsymbol{u}_j^* - |\widehat{\alpha}_j|\widehat{\boldsymbol{V}}\widehat{\boldsymbol{s}}_j\right\|_2 \lesssim \kappa^6\sqrt{\frac{r_{\max}\log q}{N}}(\sigma_1 + |\xi|)
\tag{85}
$$

provided that $N \gtrsim K^3 r_{\max}\log^4 d$.

In conclusion, we have

$$
\begin{aligned}
\|\boldsymbol{W}^{(0)} - \boldsymbol{W}^*\|_F = \|\widetilde{\boldsymbol{w}}^* - \widetilde{\boldsymbol{w}}^{(0)}\|_2 \leq& \sqrt{K}\cdot\left\|\boldsymbol{w}_{j,\Omega_j}^* - \boldsymbol{w}_{j,\Omega_j}^{(0)}\right\|_2 \\
=& \sqrt{K}\cdot\left\|\mathcal{F}_j^\dagger(\boldsymbol{u}_j^* - \widehat{\boldsymbol{u}}_j^*)\right\|_2 \\
\leq& \sqrt{K}\cdot\left\|\boldsymbol{u}_j^* - \widehat{\boldsymbol{u}}_j^*\right\|_2 \\
=& \sqrt{K}\cdot\left\|\boldsymbol{u}_j^* - |\widehat{\alpha}_j|\widehat{\boldsymbol{V}}\widehat{\boldsymbol{s}}_j\right\|_2 \\
\lesssim& \kappa^6\sqrt{\frac{Kr_{\max}\log q}{N}}(\sigma_1 + |\xi|).
\end{aligned}
\tag{86}
$$

$\square$

# G    Additional proof of the lemmas in Appendix C

## G.1    Proof of Lemma 3

The eigenvalues of $\nabla^2 f$ at any fixed point $\widetilde{\boldsymbol{w}}$ is bounded through the ones at the ground truth $\widetilde{\boldsymbol{w}}^*$ by using Lemma 2. The eigenvalues of $\nabla^2 f$ at ground truth $\widetilde{\boldsymbol{w}}^*$ is bounded in (89) and (90).

**Lemma 13.** *Let $f$ be the population risk function in* (14) *and $\widetilde{\boldsymbol{w}}$ satisfy* (6), *then we have*

$$
\|\nabla^2 f(\widetilde{\boldsymbol{w}}) - \nabla^2 f(\widetilde{\boldsymbol{w}}^*)\|_2 \leq \frac{4\|\widetilde{\boldsymbol{w}}^* - \widetilde{\boldsymbol{w}}\|_2}{\sigma_K}.
\tag{87}
$$

*Proof of Lemma 3.* Let $\lambda_{\max}(\widetilde{w})$ and $\lambda_{\min}(\widetilde{w})$ denote the largest and smallest eigenvalues of $\nabla^2 f_{\mathcal{D}}$ at point $\widetilde{w}$, respectively. Then, from Lemma 2, we have

$$
\begin{aligned}
\lambda_{\max}(\widetilde{w}) &\leq \lambda_{\max}(\widetilde{w}^*) + \|\nabla^2 f(\widetilde{w}) - \nabla^2 f(\widetilde{w}^*)\|_2, \\
\text{and} \quad \lambda_{\min}(\widetilde{w}) &\geq \lambda_{\min}(\widetilde{w}^*) - \|\nabla^2 f(\widetilde{w}) - \nabla^2 f(\widetilde{w}^*)\|_2.
\end{aligned}
\tag{88}
$$

Next, we provide the the lower bound of Hessian of population function at ground truth $\widetilde{w}^*$. Then, we have

$$
\begin{aligned}
\min_{\|\boldsymbol{\alpha}\|_2=1} \boldsymbol{\alpha}^T \nabla^2 f(\widetilde{w}^*)\boldsymbol{\alpha} &= \frac{1}{K^2} \min_{\|\boldsymbol{\alpha}\|_2=1} \mathbb{E}_{\boldsymbol{x}}\Big( \sum_{j=1}^K \boldsymbol{\alpha}_j^T \boldsymbol{x}_{\Omega_j} \phi'(\boldsymbol{w}_{j,\Omega_j}^{*T} \boldsymbol{x}_{\Omega_j})\Big)^2 \\
&= \frac{1}{K^2} \min_{\|\widetilde{\boldsymbol{\alpha}}\|_2=1,\, \mathrm{supp}(\widetilde{\boldsymbol{\alpha}}_j)=\mathrm{supp}(\boldsymbol{w}_j^*)} \mathbb{E}_{\boldsymbol{x}}\Big( \sum_{j=1}^K \widetilde{\boldsymbol{\alpha}}_j^T \boldsymbol{x} \phi'(\boldsymbol{w}_j^{*T} \boldsymbol{x})\Big)^2 \\
&\geq \frac{1}{K^2} \min_{\|\widetilde{\boldsymbol{\alpha}}\|_2=1} \mathbb{E}_{\boldsymbol{x}}\Big( \sum_{j=1}^K \widetilde{\boldsymbol{\alpha}}_j^T \boldsymbol{x} \phi'(\boldsymbol{w}_j^{*T} \boldsymbol{x})\Big)^2 \\
&\geq \frac{\rho}{11\kappa^2 \lambda K^2},
\end{aligned}
\tag{89}
$$

where $\widetilde{\boldsymbol{\alpha}} \in \mathbb{R}^{Kd}$ with $\widetilde{\boldsymbol{\alpha}}_j \in \mathbb{R}^d$, and the last inequality comes from Lemma D.6 [69].

Next, the upper bound of Hessian of population function at ground truth $\widetilde{w}^*$ can be bounded in the following way. For any $\boldsymbol{\alpha}$, we have

$$
\begin{aligned}
\boldsymbol{\alpha}^T \nabla^2 f(\widetilde{w}^*)\boldsymbol{\alpha} &= \frac{1}{K^2} \mathbb{E}_{\boldsymbol{x}}\Big( \sum_{j=1}^K \boldsymbol{\alpha}_j^T \boldsymbol{x}_{\Omega_j} \phi'(\boldsymbol{w}_{j,\Omega_j}^{*T} \boldsymbol{x}_{\Omega_j})\Big)^2 \\
&\leq \frac{2}{K^2} \cdot \mathbb{E}_{\boldsymbol{x}} \sum_{j=1}^K \Big( \boldsymbol{\alpha}_j^T \boldsymbol{x}_{\Omega_j} \phi'(\boldsymbol{w}_{j,\Omega_j}^{*T} \boldsymbol{x}_{\Omega_j})\Big)^2 \\
&= \frac{2}{K^2} \sum_{j=1}^K \mathbb{E}_{\boldsymbol{x}} \Big( \boldsymbol{\alpha}_j^T \boldsymbol{x}_{\Omega_j} \phi'(\boldsymbol{w}_{j,\Omega_j}^{*T} \boldsymbol{x}_{\Omega_j})\Big)^2 \\
&\leq \frac{2}{K^2} \sum_{j=1}^K \Big( \mathbb{E}_{\boldsymbol{x}}(\boldsymbol{\alpha}_j^T \boldsymbol{x}_{\Omega_j})^4 \mathbb{E}_{\boldsymbol{x}}|\phi'|^4 \Big)^{\frac{1}{2}} \\
&\leq \frac{2}{K^2} \cdot K \cdot 3 = \frac{6}{K}.
\end{aligned}
\tag{90}
$$

Then, from Lemma 13, when $\widetilde{w}$ satisfies (6), we have that

$$
\|\nabla^2 f(\widetilde{w}) - \nabla^2 f(\widetilde{w}^*)\|_2 \leq \frac{\varepsilon_0 \rho}{11\kappa^2 \gamma}.
\tag{91}
$$

Hence, from (88) and (91), we have that

$$
\frac{(1-\varepsilon_0)\rho}{11\kappa^2 \gamma K^2} \boldsymbol{I} \preceq \nabla^2 f(\widetilde{w}) \preceq \frac{7}{K} \boldsymbol{I}.
\tag{92}
$$

$\square$

## G.2 Proof of Lemma 4

We first show that the second order derivative of $\hat{f}_{\mathcal{D}}$ is a sum of several random sub-exponential variables as shown in (101) and (102). Then, by concentration theory, i.e., Chernoff bound, we can show that the error bound of $\nabla^2 \hat{f}_{\mathcal{D}}$ to its expectation.

**Definition 1** (Definition 5.7, [54]). *A random variable $X$ is called a sub-Gaussian random variable if it satisfies*

$$(\mathbb{E}|X|^p)^{1/p} \leq c_1\sqrt{p} \tag{93}$$

*for all $p \geq 1$ and some constant $c_1 > 0$. In addition, we have*

$$\mathbb{E}e^{s(X-\mathbb{E}X)} \leq e^{c_2\|X\|_{\psi_2}^2 s^2} \tag{94}$$

*for all $s \in \mathbb{R}$ and some constant $c_2 > 0$, where $\|X\|_{\phi_2}$ is the sub-Gaussian norm of $X$ defined as $\|X\|_{\psi_2} = \sup_{p\geq 1} p^{-1/2}(\mathbb{E}|X|^p)^{1/p}$.*

*Moreover, a random vector $\boldsymbol{X} \in \mathbb{R}^d$ belongs to the sub-Gaussian distribution if one-dimensional marginal $\boldsymbol{\alpha}^T\boldsymbol{X}$ is sub-Gaussian for any $\boldsymbol{\alpha} \in \mathbb{R}^d$, and the sub-Gaussian norm of $\boldsymbol{X}$ is defined as $\|\boldsymbol{X}\|_{\psi_2} = \sup_{\|\boldsymbol{\alpha}\|_2=1} \|\boldsymbol{\alpha}^T\boldsymbol{X}\|_{\psi_2}$.*

**Definition 2** (Definition 5.13, [54]). *A random variable $X$ is called a sub-exponential random variable if it satisfies*

$$(\mathbb{E}|X|^p)^{1/p} \leq c_3 p \tag{95}$$

*for all $p \geq 1$ and some constant $c_3 > 0$. In addition, we have*

$$\mathbb{E}e^{s(X-\mathbb{E}X)} \leq e^{c_4\|X\|_{\psi_1}^2 s^2} \tag{96}$$

*for $s \leq 1/\|X\|_{\psi_1}$ and some constant $c_4 > 0$, where $\|X\|_{\psi_1}$ is the sub-exponential norm of $X$ defined as $\|X\|_{\psi_1} = \sup_{p\geq 1} p^{-1}(\mathbb{E}|X|^p)^{1/p}$.*

**Lemma 14** (Lemma 5.2, [54]). *Let $\mathcal{B}(0,1) \in \{\boldsymbol{\alpha} | \|\boldsymbol{\alpha}\|_2 = 1, \boldsymbol{\alpha} \in \mathbb{R}^d\}$ denote a unit ball in $\mathbb{R}^d$. Then, a subset $\mathcal{S}_\xi$ is called a $\xi$-net of $\mathcal{B}(0,1)$ if every point $\boldsymbol{z} \in \mathcal{B}(0,1)$ can be approximated to within $\xi$ by some point $\boldsymbol{\alpha} \in \mathcal{B}(0,1)$, i.e. $\|\boldsymbol{z} - \boldsymbol{\alpha}\|_2 \leq \xi$. Then the minimal cardinality of a $\xi$-net $\mathcal{S}_\xi$ satisfies*

$$|\mathcal{S}_\xi| \leq (1 + 2/\xi)^d. \tag{97}$$

**Lemma 15** (Lemma 5.3, [54]). *Let $\boldsymbol{A}$ be an $d_1 \times d_2$ matrix, and let $\mathcal{S}_\xi(d)$ be a $\xi$-net of $\mathcal{B}(0,1)$ in $\mathbb{R}^d$ for some $\xi \in (0,1)$. Then*

$$\|\boldsymbol{A}\|_2 \leq (1-\xi)^{-1} \max_{\boldsymbol{\alpha}_1 \in \mathcal{S}_\xi(d_1), \boldsymbol{\alpha}_2 \in \mathcal{S}_\xi(d_2)} |\boldsymbol{\alpha}_1^T \boldsymbol{A} \boldsymbol{\alpha}_2|. \tag{98}$$

*Proof of Lemma 4.* Recall the definition of $f$ and $\hat{f}$ in (14) and (12), we have

$$
\begin{aligned}
&\frac{\partial^2 f}{\partial \boldsymbol{w}_{j_1,\Omega_{j_1}} \partial \boldsymbol{w}_{j_2,\Omega_{j_2}}} - \frac{\partial^2 \hat{f}_{\mathcal{D}}}{\partial \boldsymbol{w}_{j_1,\Omega_{j_1}} \partial \boldsymbol{w}_{j_2,\Omega_{j_2}}} \\
&= \mathbb{E}_{\boldsymbol{x}}\Big[\phi'(\boldsymbol{w}_{j_1,\Omega_{j_1}}^T \boldsymbol{x}_{\Omega_{j_1}})\phi'(\boldsymbol{w}_{j_2,\Omega_{j_2}}^T \boldsymbol{x}_{\Omega_{j_2}})\boldsymbol{x}_{\Omega_{j_1}} \boldsymbol{x}_{\Omega_{j_2}}^T \\
&\quad - \frac{1}{N}\sum_{n=1}^N \phi(\boldsymbol{w}_{j_1,\Omega_{j_1}}^T \boldsymbol{x}_{n,\Omega_{j_1}})\phi'(\boldsymbol{w}_{j_2,\Omega_{j_2}}^T \boldsymbol{x}_{n,\Omega_{j_2}})\boldsymbol{x}_{n,\Omega_{j_1}} \boldsymbol{x}_{n,\Omega_{j_2}}^T\Big].
\end{aligned} \tag{99}
$$

For any $\boldsymbol{\alpha}$, we have

$$
\begin{aligned}
&\|\nabla^2 f - \nabla^2 \hat{f}_{\mathcal{D}}\|_2 \\
&= \max_{\|\boldsymbol{\alpha}\|_2=1} \Big|\boldsymbol{\alpha}^T(\nabla^2 f - \nabla^2 \hat{f}_{\mathcal{D}})\boldsymbol{\alpha}\Big| \\
&= \sum_{j_1=1}^K \sum_{j_2=1}^K \max_{\|\boldsymbol{\alpha}\|_2=1} \left|\boldsymbol{\alpha}_{j_1}^T \Big(\frac{\partial^2 f}{\partial \boldsymbol{w}_{j_1,\Omega_{j_1}} \partial \boldsymbol{w}_{j_2,\Omega_{j_2}}} - \frac{\partial^2 \hat{f}_{\mathcal{D}}}{\partial \boldsymbol{w}_{j_1,\Omega_{j_1}} \partial \boldsymbol{w}_{j_2,\Omega_{j_2}}}\Big)\boldsymbol{\alpha}_{j_2}\right| \\
&= \frac{1}{K^2}\sum_{j_1=1}^K \sum_{j_2=1}^K \max_{\|\boldsymbol{\alpha}\|_2=1} \mathbb{E}_{\boldsymbol{x}}\Big[\phi'(\boldsymbol{w}_{j_1,\Omega_{j_1}}^T \boldsymbol{x}_{\Omega j_1})\phi'(\boldsymbol{w}_{j_2,\Omega_{j_2}}^T \boldsymbol{x}_{\Omega j_2})\boldsymbol{\alpha}_{j_1}^T \boldsymbol{x}_{\Omega j_1} \boldsymbol{\alpha}_{j_2}^T \boldsymbol{x}_{\Omega j_2} \\
&\quad - \frac{1}{N}\sum_{n=1}^N \phi'(\boldsymbol{w}_{j_1,\Omega_{j_1}}^T \boldsymbol{x}_{n,\Omega j_1})\phi'(\boldsymbol{w}_{j_2,\Omega_{j_2}}^T \boldsymbol{x}_{n,\Omega j_2})\boldsymbol{\alpha}_{j_1}^T \boldsymbol{x}_{n,\Omega j_1} \boldsymbol{\alpha}_{j_2}^T \boldsymbol{x}_{n,\Omega j_2}\Big].
\end{aligned} \tag{100}
$$

Then, define $Z_n(j_1, j_2) = \phi(\boldsymbol{w}_{j_1,\Omega_{j_1}}^T \boldsymbol{x}_{n,\Omega j_1})\phi'(\boldsymbol{w}_{j_2,\Omega_{j_2}}^T \boldsymbol{x}_{n,\Omega j_2})\boldsymbol{\alpha}_{j_1}^T \boldsymbol{x}_{n,\Omega j_1} \boldsymbol{\alpha}_{j_2}^T \boldsymbol{x}_{n,\Omega j_2}$, and we say $Z$ belongs to sub-Exponential distribution by Definition 2. If $|\Omega_{j_1} \cap \Omega_{j_2}| \neq \emptyset$, namely, $\Omega_{j_1}$ and $\Omega_{j_2}$ are not disjointed, we have

$$
\begin{aligned}
\left(\mathbb{E}|Z_n|^p\right)^{1/p} &\leq \left(\mathbb{E}\left|\left(\boldsymbol{\alpha}_{j_1}^T \boldsymbol{x}_{n,\Omega j_1}\right) \cdot \left(\boldsymbol{\alpha}_{j_2}^T \boldsymbol{x}_{n,\Omega j_2}\right)\right|^p\right)^{1/p} \\
&\leq \left(\mathbb{E}\left|\left(\boldsymbol{\alpha}_{j_1}^T \boldsymbol{x}_{n,\Omega j_1}\right)\right|^{2p}\right)^{1/(2p)} \cdot \left(\mathbb{E}\left|\left(\boldsymbol{\alpha}_{j_2}^T \boldsymbol{x}_{n,\Omega j_2}\right)\right|^{2p}\right)^{1/(2p)} \\
&\leq C_{\boldsymbol{x}} \cdot \sqrt{2p} \cdot C_{\boldsymbol{x}} \sqrt{2p} \\
&= 2C_{\boldsymbol{x}}^2 \cdot p.
\end{aligned}
\tag{101}
$$

While if $|\Omega_{j_1} \cap \Omega_{j_2}| = \emptyset$, namely, $\Omega_{j_1}$ and $\Omega_{j_2}$ are disjointed, we have

$$
\begin{aligned}
\left(\mathbb{E}|Z_n|^p\right)^{1/p} &\leq \left(\mathbb{E}\left|\left(\boldsymbol{\alpha}_{j_1}^T \boldsymbol{x}_{n,\Omega j_1}\right) \cdot \left(\boldsymbol{\alpha}_{j_2}^T \boldsymbol{x}_{n,\Omega j_2}\right)\right|^p\right)^{1/p} \\
&= \left(\mathbb{E}\left|\left(\boldsymbol{\alpha}_{j_1}^T \boldsymbol{x}_{n,\Omega j_1}\right)\right|^p\right)^{1/(p)} \cdot \left(\mathbb{E}\left|\left(\boldsymbol{\alpha}_{j_2}^T \boldsymbol{x}_{n,\Omega j_2}\right)\right|^p\right)^{1/(p)} \\
&\leq C_{\boldsymbol{x}} \cdot \sqrt{p} \cdot C_{\boldsymbol{x}} \sqrt{p} \\
&= C_{\boldsymbol{x}}^2 \cdot p.
\end{aligned}
\tag{102}
$$

Then, we have

$$
\mathbb{E}_{Z_n} e^{s(Z_n - \mathbb{E}Z_n)} \leq e^{-C\|Z_n\|_{\psi_1}^2 s^2}
\tag{103}
$$

for some constant $C > 0$ and any $s \in \mathbb{R}$. From Chernoff bound, we have

$$
\text{Prob}\left\{\left|\frac{1}{N}\sum_{n=1}^N (Z_n - \mathbb{E}Z_n)\right| < t\right\} \leq 1 - \frac{e^{-C\|Z_n\|_{\psi_1}^2 \cdot Ns^2}}{e^{Nst}}.
\tag{104}
$$

Let us select $t = \|Z_n\|_{\psi_1}\sqrt{\frac{(r_{j_1}+r_{j_2})\log q}{N}}$ and $s = \frac{\sqrt{2}}{C\|z_n\|_{\psi_1}^2} \cdot t$, then we have

$$
\left|\frac{1}{N}\sum_{n=1}^N \left(Z_n(j_1, j_2) - \mathbb{E}Z_n(j_1, j_2)\right)\right| \leq \|Z_n\|_{\psi_1}\sqrt{\frac{(r_{j_1}+r_{j_2})\log q}{N}}
\tag{105}
$$

with probability at least $1 - q^{-(r_{j_1}+r_{j_2})}$.

Hence, from Lemma 15, we have

$$
\begin{aligned}
\max_{\|\boldsymbol{\alpha}_{j_1}\|_2 \leq 1, \|\boldsymbol{\alpha}_{j_2}\|_2 \leq 1} &\left|\boldsymbol{\alpha}_{j_1}^T\left(\frac{\partial^2 f}{\partial \boldsymbol{w}_{j_1,\Omega_{j_1}}\partial \boldsymbol{w}_{j_2,\Omega_{j_2}}} - \frac{\partial^2 \hat{f}_{\mathcal{D}}}{\partial \boldsymbol{w}_{j_1,\Omega_{j_1}}\partial \boldsymbol{w}_{j_2,\Omega_{j_2}}}\right)\boldsymbol{\alpha}_{j_2}\right| \\
&\leq 2\left|\frac{1}{N}\sum_{n=1}^N (Z_n - \mathbb{E}Z_n)\right|
\end{aligned}
\tag{106}
$$

with probability at least $1 - \left(|\mathcal{S}_{\frac{1}{2}}(r_{j_1})| \cdot |\mathcal{S}_{\frac{1}{2}}(r_{j_2})|\right) \cdot q^{-(r_{j_1}+r_{j_2})}$, where $\mathcal{S}_{\frac{1}{2}}(r_{j_1})$ and $\mathcal{S}_{\frac{1}{2}}(r_{j_2})$ are the covering sets defined in Lemma 14. From Lemma 14, we know that $|\mathcal{S}_{\frac{1}{2}}(r_{j_1})| \cdot |\mathcal{S}_{\frac{1}{2}}(r_{j_2})| \leq 5^{(r_{j_1}+r_{j_2})}$. As long as $q$ is a constant that is larger than 5, (106) holds with the probability at least $1 - \left(\frac{q}{5}\right)^{-(r_{j_1}+r_{j_2})}$. For notation simplification, we use probability $1 - q^{-(r_{j_1}+r_{j_2})}$ instead.

From (101) and (102), we know that

$$
\|Z_n(j_1, j_2)\|_{\psi_1} \leq \begin{cases} 2C_{\boldsymbol{x}}^2, & \text{if } \Omega_{j_1} \text{ and } \Omega_{j_2} \text{ are joint sets} \\ C_{\boldsymbol{x}}^2, & \text{if } \Omega_{j_1} \text{ and } \Omega_{j_2} \text{ are disjoint sets} \end{cases} .
\tag{107}
$$

Hence, we have

$$
\begin{aligned}
&\|\nabla^2 f(\widetilde{\boldsymbol{w}}) - \nabla^2 \hat{f}_\Omega(\widetilde{\boldsymbol{w}})\|_2 \\
&\leq \sum_{j_1=1}^{K} \sum_{j_2=1}^{K} \max_{\|\boldsymbol{\alpha}\|_2=1} \left| \boldsymbol{\alpha}_{j_1}^T \left( \frac{\partial^2 f}{\partial \boldsymbol{w}_{j_1,\Omega_{j_1}} \partial \boldsymbol{w}_{j_2,\Omega_{j_2}}} - \frac{\partial^2 \hat{f}_{\mathcal{D}}}{\partial \boldsymbol{w}_{j_1,\Omega_{j_1}} \partial \boldsymbol{w}_{j_2,\Omega_{j_2}}} \right) \boldsymbol{\alpha}_{j_2} \right| \\
&\leq \frac{2}{K^2} \sum_{j_1=1}^{K} \sum_{j_2=1}^{K} \max_{\|\boldsymbol{\alpha}\|_2=1} \left| \frac{1}{N} \sum_{n=1}^{N} \left( Z_n(j_1, j_2) - \mathbb{E} Z_n(j_1, j_2) \right) \right| \\
&\lesssim \frac{1}{K^2} \sum_{j_1=1}^{K} \sum_{j_2=1}^{K} \sqrt{\frac{(1+\delta_{j_1,j_2})^2 (r_{j_1} + r_{j_2}) \log q}{N}}
\end{aligned}
\tag{108}
$$

with probability at least $1 - q^{-r_{\min}}$, where $\delta_{j_1,j_2}$ equals to 0 if $\Omega_{j_1}$ and $\Omega_{j_2}$ are disjoint and 1 otherwise. $\qquad\square$

## H    Proof of Lemma 5

*Proof of Lemma 5.* The first-order derivative of the empirical risk function is written as

$$
\begin{aligned}
\frac{\partial \hat{f}_{\mathcal{D}}}{\partial \boldsymbol{w}_{k,\Omega_k}} =& \frac{1}{K \cdot N} \sum_{n=1}^{N} \left( y_n - \frac{1}{K} \sum_{j=1}^{K} \phi(\boldsymbol{w}_{j,\Omega_j}^T \boldsymbol{x}_{n,\Omega_j}) \right) \boldsymbol{x}_{n,\Omega_k} \phi'(\boldsymbol{w}_{k,\Omega_k}^T \boldsymbol{x}_{n,\Omega_k}) \\
=& \frac{1}{K^2 \cdot N} \sum_{n=1}^{N} \sum_{j=1}^{K} \left( \phi(\boldsymbol{w}_{j,\Omega_j}^{*T} \boldsymbol{x}_{n,\Omega_j}) - \phi(\boldsymbol{w}_{j,\Omega_j}^T \boldsymbol{x}_{n,\Omega_j}) \right) \boldsymbol{x}_{n,\Omega_k} \phi'(\boldsymbol{w}_{k,\Omega_k}^T \boldsymbol{x}_{n,\Omega_k}) \\
& + \frac{1}{K \cdot N} \sum_{j=1}^{K} \xi_n \boldsymbol{x}_{n,\Omega_k} \phi'(\boldsymbol{w}_{k,\Omega_k}^T \boldsymbol{x}_{n,\Omega_k})
\end{aligned}
\tag{109}
$$

Define $\boldsymbol{z}_n(j,k) = \left( \phi(\boldsymbol{w}_{j,\Omega_j}^{*T} \boldsymbol{x}_{n,\Omega_j}) - \phi(\boldsymbol{w}_{j,\Omega_j}^T \boldsymbol{x}_{n,\Omega_j}) \right) \phi'(\boldsymbol{w}_{k,\Omega_k}^T \boldsymbol{x}_{n,\Omega_k}) \boldsymbol{x}_{n,\Omega_k}$. Then, for any $\boldsymbol{\alpha}_k \in \mathbb{R}^r$, we have

$$
\begin{aligned}
& p^{-1} \left( \mathbb{E}_{\boldsymbol{x}} \left| \boldsymbol{\alpha}_k^T \boldsymbol{z}_n \right|^p \right)^{\frac{1}{p}} \\
=& p^{-1} \left( \mathbb{E}_{\boldsymbol{x}} \left| (\boldsymbol{\alpha}_k^T \boldsymbol{x}_{n,\Omega_k}) \left( \phi(\boldsymbol{w}_{j,\Omega_j}^{*T} \boldsymbol{x}_{n,\Omega_j}) - \phi(\boldsymbol{w}_{j,\Omega_j}^T \boldsymbol{x}_{n,\Omega_j}) \right) \phi'(\boldsymbol{w}_{k,\Omega_k}^T \boldsymbol{x}_{n,\Omega_k}) \right|^p \right)^{\frac{1}{p}} \\
\leq& p^{-1} \left( \mathbb{E}_{\boldsymbol{x}} \left| (\boldsymbol{\alpha}_k^T \boldsymbol{x}_{n,\Omega_k}) \left( \phi(\boldsymbol{w}_{j,\Omega_j}^{*T} \boldsymbol{x}_{n,\Omega_j}) - \phi(\boldsymbol{w}_{j,\Omega_j}^T \boldsymbol{x}_{n,\Omega_j}) \right) \right|^p \right)^{\frac{1}{p}}.
\end{aligned}
\tag{110}
$$

If $\Omega_j$ and $\Omega_k$ are joint, then

$$
\begin{aligned}
& p^{-1} \left( \mathbb{E}_{\boldsymbol{x}} \left| \boldsymbol{\alpha}_k^T \boldsymbol{z}_n \right|^p \right)^{\frac{1}{p}} \\
\leq& p^{-1} \left( \mathbb{E}_{\boldsymbol{x}} | \boldsymbol{\alpha}_k^T \boldsymbol{x}_{n,\Omega_k} |^{2p} \right)^{\frac{1}{2p}} \cdot \left( \mathbb{E}_{\boldsymbol{x}} \left| \phi(\boldsymbol{w}_{j,\Omega_j}^{*T} \boldsymbol{x}_{n,\Omega_j}) - \phi(\boldsymbol{w}_{j,\Omega_j}^T \boldsymbol{x}_{n,\Omega_j}) \right|^{2p} \right)^{\frac{1}{2p}} \\
\leq& p^{-1} \left( \mathbb{E}_{\boldsymbol{x}} | \boldsymbol{\alpha}_k^T \boldsymbol{x}_{n,\Omega_k} |^{2p} \right)^{\frac{1}{2p}} \cdot \left( \mathbb{E}_{\boldsymbol{x}} \left| (\boldsymbol{w}_{j,\Omega_j}^* - \boldsymbol{w}_{j,\Omega_j})^T \boldsymbol{x}_{n,\Omega_j} \right|^{2p} \right)^{\frac{1}{2p}} \\
\leq& 2 \| \boldsymbol{w}_{j,\Omega_j}^* - \boldsymbol{w}_{j,\Omega_j} \|_2 \leq 2 \| \widetilde{\boldsymbol{w}}^* - \widetilde{\boldsymbol{w}} \|_2.
\end{aligned}
\tag{111}
$$

If $\Omega_j$ and $\Omega_k$ are disjoint, then

$$
\begin{aligned}
& p^{-1} \left( \mathbb{E}_{\boldsymbol{x}} \left| \boldsymbol{\alpha}_k^T \boldsymbol{z}_n \right|^p \right)^{\frac{1}{p}} \\
\leq& p^{-1} \left( \mathbb{E}_{\boldsymbol{x}} | \boldsymbol{\alpha}_j^T \boldsymbol{x}_{n,\Omega_j} |^p \right)^{\frac{1}{p}} \cdot \left( \mathbb{E}_{\boldsymbol{x}} \left| \phi(\boldsymbol{w}_{j,\Omega_j}^{*T} \boldsymbol{x}_{n,\Omega_j}) - \phi(\boldsymbol{w}_{j,\Omega_j}^T \boldsymbol{x}_{n,\Omega_j}) \right|^p \right)^{\frac{1}{p}} \\
\leq& p^{-1} \left( \mathbb{E}_{\boldsymbol{x}} | \boldsymbol{\alpha}_j^T \boldsymbol{x}_{n,\Omega_j} |^p \right)^{\frac{1}{p}} \cdot \left( \mathbb{E}_{\boldsymbol{x}} \left| (\boldsymbol{w}_{j,\Omega_j}^* - \boldsymbol{w}_{j,\Omega_j})^T \boldsymbol{x}_{n,\Omega_j} \right|^p \right)^{\frac{1}{p}} \\
\leq& \| \boldsymbol{w}_{j,\Omega_j}^* - \boldsymbol{w}_{j,\Omega_j} \|_2 \leq \| \widetilde{\boldsymbol{w}}^* - \widetilde{\boldsymbol{w}} \|_2.
\end{aligned}
\tag{112}
$$

Following similar steps in (104), by Chernoff bound, we have

$$\Big\|\frac{1}{N}\sum_{n=1}^{N}(\boldsymbol{z}_n - \mathbb{E}_{\boldsymbol{x}}\boldsymbol{z}_n)\Big\|_2 \lesssim \|\boldsymbol{z}_n(j,k)\|_{\psi_1}\sqrt{\frac{r_j\log q}{N}}\cdot\|\boldsymbol{w}_{j,\Omega_j}^* - \boldsymbol{w}_{j,\Omega_j}\|_2 \tag{113}$$

with probability at least $1 - q^{-r_j}$, where

$$\|\boldsymbol{z}_n(j,k)\|_{\psi_1} = \begin{cases} 2\|\widetilde{\boldsymbol{w}} - \widetilde{\boldsymbol{w}}^*\|_2, & \text{if } \Omega_k \text{ and } \Omega_j \text{ are joint,} \\ \|\widetilde{\boldsymbol{w}} - \widetilde{\boldsymbol{w}}^*\|_2, & \text{if } \Omega_k \text{ and } \Omega_j \text{ are disjoint} \end{cases} \tag{114}$$

That is $\|\boldsymbol{z}_n(j,k)\|_{\psi_1} = (1 + \delta_{j,k})\|\widetilde{\boldsymbol{w}} - \widetilde{\boldsymbol{w}}^*\|_2$. Also, we know that $\boldsymbol{x}_{n,\Omega_k}\phi'(\boldsymbol{w}_{k,\Omega_k}^T\boldsymbol{x}_{n,\Omega_k})$ belongs to sub-Gaussian distribution as well. Then, by Chernoff bound, we have

$$\Big\|\frac{1}{N}\sum_{n=1}^{N}\xi_n\boldsymbol{x}_{n,\Omega_k}\phi'(\boldsymbol{w}_{k,\Omega_k}^T\boldsymbol{x}_{n,\Omega_k})\Big\|_2 \lesssim |\xi|\cdot\Big\|\frac{1}{N}\sum_{n=1}^{N}\boldsymbol{x}_{n,\Omega_k}\phi'(\boldsymbol{w}_{k,\Omega_k}^T\boldsymbol{x}_{n,\Omega_k})\Big\|_2$$
$$\lesssim |\xi|\cdot\sqrt{\frac{r_k\log q}{N}} \tag{115}$$

with probability at least $q^{-r_k}$.

In conclusion, we have

$$\begin{aligned}
\|\nabla\hat{f}_{\mathcal{D}} - \nabla f\|_2 &\leq \sum_{k=1}^{K}\Big\|\frac{\partial\hat{f}_{\mathcal{D}}}{\partial\boldsymbol{w}_k} - \frac{\partial f}{\partial\boldsymbol{w}_k}\Big\|_2 \\
&\leq \sum_{k=1}^{K}\frac{1}{K^2}\sum_{j=1}^{K}\Big\|\frac{1}{N}\sum_{n=1}^{N}(\boldsymbol{z}_n(j,k) - \mathbb{E}_{\boldsymbol{x}}\boldsymbol{z}_n(j,k))\Big\|_2 \\
&\quad + \sum_{k=1}^{K}\frac{1}{K}\Big\|\frac{1}{N}\sum_{n=1}^{N}\xi_n\boldsymbol{x}_{n,\Omega_k}\phi'(\boldsymbol{w}_{k,\Omega_k}^T\boldsymbol{x}_{n,\Omega_k})\Big\|_2 \\
&\lesssim \frac{1}{K^2}\sum_{k=1}^{K}\sum_{j=1}^{K}\sqrt{\frac{(1+\delta_{j,k})^2 r_k\log q}{N}}\|\widetilde{\boldsymbol{w}}^* - \widetilde{\boldsymbol{w}}\|_2 + \frac{1}{K}\sum_{k=1}^{K}\sqrt{\frac{r_k\log q}{N}}\cdot|\xi|.
\end{aligned} \tag{116}$$

$\square$

# I   Proof of Lemma 13

*Proof of Lemma 13.* Recall the definition of population risk function, we have

$$\frac{\partial^2 f(\boldsymbol{w}^*)}{\partial\boldsymbol{w}_{j_1,\Omega_{j_1}}\partial\boldsymbol{w}_{j_2,\Omega_{j_2}}} = \frac{1}{K^2}\mathbb{E}_{\boldsymbol{x}}\phi'(\boldsymbol{w}_{j_1,\Omega_{j_1}}^{*T}\boldsymbol{x}_{\Omega j_1})\phi'(\boldsymbol{w}_{j_2,\Omega_{j_2}}^{*T}\boldsymbol{x}_{\Omega j_2})\boldsymbol{x}_{\Omega j_1}\boldsymbol{x}_{\Omega j_2}^T \tag{117}$$

and

$$\frac{\partial^2 f(\boldsymbol{w})}{\partial\boldsymbol{w}_{j_1,\Omega_{j_1}}\partial\boldsymbol{w}_{j_2,\Omega_{j_2}}} = \frac{1}{K^2}\mathbb{E}_{\boldsymbol{x}}\phi'(\boldsymbol{w}_{j_1,\Omega_{j_1}}^{T}\boldsymbol{x}_{\Omega j_1})\phi'(\boldsymbol{w}_{j_2,\Omega_{j_2}}^{T}\boldsymbol{x}_{\Omega j_2})\boldsymbol{x}_{\Omega j_1}\boldsymbol{x}_{\Omega j_2}^T \tag{118}$$

Then, we have

$$
\frac{\partial^2 f(\boldsymbol{w}^*)}{\partial \boldsymbol{w}_{j_1,\Omega_{j_1}} \partial \boldsymbol{w}_{j_2,\Omega_{j_2}}} - \frac{\partial^2 f(\boldsymbol{w})}{\partial \boldsymbol{w}_{j_1,\Omega_{j_1}} \partial \boldsymbol{w}_{j_2,\Omega_{j_2}}}
$$

$$
= \frac{1}{K^2} \mathbb{E}_{\boldsymbol{x}} \Big[ \phi'(\boldsymbol{w}^{*T}_{j_1,\Omega_{j_1}} \boldsymbol{x}_{\Omega j_1}) \phi'(\boldsymbol{w}^{*T}_{j_2,\Omega_{j_2}} \boldsymbol{x}_{\Omega j_2}) - \phi'(\boldsymbol{w}^{T}_{j_1,\Omega_{j_1}} \boldsymbol{x}_{\Omega j_1}) \phi'(\boldsymbol{w}^{T}_{j_2,\Omega_{j_2}} \boldsymbol{x}_{\Omega j_2}) \Big] \boldsymbol{x}_{\Omega j_1} \boldsymbol{x}^T_{\Omega j_2}
$$

$$
= \frac{1}{K^2} \mathbb{E}_{\boldsymbol{x}} \Big[ \phi'(\boldsymbol{w}^{*T}_{j_1,\Omega_{j_1}} \boldsymbol{x}_{\Omega j_1}) \big( \phi'(\boldsymbol{w}^{*T}_{j_2,\Omega_{j_2}} \boldsymbol{x}_{\Omega j_2}) - \phi'(\boldsymbol{w}^{T}_{j_2,\Omega_{j_2}} \boldsymbol{x}_{\Omega j_2}) \big)
$$

$$
+ \phi'(\boldsymbol{w}^{T}_{j_2,\Omega_{j_2}} \boldsymbol{x}_{\Omega j_2}) \big( \phi'(\boldsymbol{w}^{*T}_{j_1,\Omega_{j_1}} \boldsymbol{x}_{\Omega j_1}) - \phi'(\boldsymbol{w}^{T}_{j_1,\Omega_{j_1}} \boldsymbol{x}_{\Omega j_1}) \big) \Big] \boldsymbol{x}_{\Omega j_1} \boldsymbol{x}^T_{\Omega j_2}
$$

$$
= \frac{1}{K^2} \Big[ \mathbb{E}_{\boldsymbol{x}} \phi'(\boldsymbol{w}^{*T}_{j_1,\Omega_{j_1}} \boldsymbol{x}_{\Omega j_1}) \big( \phi'(\boldsymbol{w}^{*T}_{j_2,\Omega_{j_2}} \boldsymbol{x}_{\Omega j_2}) - \phi'(\boldsymbol{w}^{T}_{j_2,\Omega_{j_2}} \boldsymbol{x}_{\Omega j_2}) \big) \boldsymbol{x}_{\Omega j_1} \boldsymbol{x}^T_{\Omega j_2}
$$

$$
+ \mathbb{E}_{\boldsymbol{x}} \phi'(\boldsymbol{w}^{T}_{j_2,\Omega_{j_2}} \boldsymbol{x}_{\Omega j_2}) \big( \phi'(\boldsymbol{w}^{*T}_{j_1,\Omega_{j_1}} \boldsymbol{x}_{\Omega j_1}) - \phi'(\boldsymbol{w}^{T}_{j_1,\Omega_{j_1}} \boldsymbol{x}_{\Omega j_1}) \big) \boldsymbol{x}_{\Omega j_1} \boldsymbol{x}^T_{\Omega j_2} \Big]
$$

$$
:= \frac{1}{K^2} (\boldsymbol{I}_1 + \boldsymbol{I}_2). \tag{119}
$$

For any $\boldsymbol{\alpha}_{j_1} \in \mathbb{R}^{r_{j_1}}$ and $\boldsymbol{\alpha}_{j_2} \in \mathbb{R}^{r_{j_2}}$, we have

$$
\max_{\|\boldsymbol{\alpha}_{j_1}\|_2, \|\boldsymbol{\alpha}_{j_2}\|_2 = 1} \boldsymbol{\alpha}^T_{j_1} \boldsymbol{I}_1 \boldsymbol{\alpha}_{j_2}
$$

$$
= \max_{\|\boldsymbol{\alpha}_{j_1}\|_2, \|\boldsymbol{\alpha}_{j_2}\|_2 = 1} \mathbb{E}_{\boldsymbol{x}} \phi'(\boldsymbol{w}^{*T}_{j_1,\Omega_{j_1}} \boldsymbol{x}_{\Omega j_1}) \big( \phi'(\boldsymbol{w}^{*T}_{j_2,\Omega_{j_2}} \boldsymbol{x}_{\Omega j_2}) - \phi'(\boldsymbol{w}^{T}_{j_2,\Omega_{j_2}} \boldsymbol{x}_{\Omega j_2}) \big)
$$

$$
\cdot (\boldsymbol{\alpha}^T_{j_1} \boldsymbol{x}_{\Omega j_1}) \cdot (\boldsymbol{\alpha}^T_{j_2} \boldsymbol{x}_{\Omega j_2}) \tag{120}
$$

$$
\leq \max_{\|\boldsymbol{a}\|_2 = 1} \mathbb{E}_{\boldsymbol{x}} \phi'(\boldsymbol{w}^{*T}_{j_2} \boldsymbol{x}) \big( \phi'(\boldsymbol{w}^{*T}_{j_2} \boldsymbol{x}) - \phi'(\boldsymbol{w}^{T}_{j_2} \boldsymbol{x}) \big) \cdot (\boldsymbol{a}^T \boldsymbol{x})^2,
$$

where $\boldsymbol{a} \in \mathbb{R}^d$. Let $I = \phi'(\boldsymbol{w}^{*T}_{j_1} \boldsymbol{x}) \big( \phi'(\boldsymbol{w}^{*T}_{j_2} \boldsymbol{x}) - \phi'(\boldsymbol{w}^{T}_{j_2} \boldsymbol{x}) \big) \cdot (\boldsymbol{a}^T \boldsymbol{x})^2$. It is easy to verify there exists a basis such that $\mathcal{B} = \{\boldsymbol{a}, \boldsymbol{b}, \boldsymbol{c}, \boldsymbol{a}^\perp_4, \cdots, \boldsymbol{a}^\perp_d\}$ with $\{\boldsymbol{a}, \boldsymbol{b}, \boldsymbol{c}\}$ spans a subspace that contains $\boldsymbol{a}, \boldsymbol{w}_{j_2}$ and $\boldsymbol{w}^*_{j_2}$. Then, for any $\boldsymbol{x}$, we have a unique $\boldsymbol{z} = [z_1 \quad z_2 \quad \cdots \quad z_d]^T$ such that

$$
\boldsymbol{x} = z_1 \boldsymbol{a} + z_2 \boldsymbol{b} + z_3 \boldsymbol{c} + \cdots + z_d \boldsymbol{a}^\perp_d.
$$

Also, since $\boldsymbol{x} \sim \mathcal{N}(\boldsymbol{0}, \boldsymbol{I}_d)$, we have $\boldsymbol{z} \sim \mathcal{N}(\boldsymbol{0}, \boldsymbol{I}_d)$. Then, we have

$$
I = \mathbb{E}_{z_1, z_2, z_3} |\phi'(\boldsymbol{w}^{T}_{j_2} \boldsymbol{x}) - \phi'(\boldsymbol{w}^{*T}_{j_2} \boldsymbol{x})| \cdot |\boldsymbol{a}^T \boldsymbol{x}|^2
$$

$$
= \int |\phi'(\boldsymbol{w}^{T}_{j_2} \boldsymbol{x}) - \phi'(\boldsymbol{w}^{*T}_{j_2} \boldsymbol{x})| \cdot |\boldsymbol{a}^T \boldsymbol{x}|^2 \cdot f_Z(z_1, z_2, z_3) dz_1 dz_2 dz_3,
$$

where $\boldsymbol{x} = z_1 \boldsymbol{a} + z_2 \boldsymbol{b} + z_3 \boldsymbol{c}$ and $f_Z(z_1, z_2, z_3)$ is probability density function of $(z_1, z_2, z_3)$. Next, we consider spherical coordinates with $z_1 = R\cos\phi_1, z_2 = R\sin\phi_1\sin\phi_2, z_3 = z_2 = R\sin\phi_1\cos\phi_2$. Hence,

$$
I = \int |\phi'(\boldsymbol{w}^{T}_{j_2} \boldsymbol{x}) - \phi'(\boldsymbol{w}^{*T}_{j_2} \boldsymbol{x})| \cdot |r\cos\phi_1|^2 \cdot \cdot f_Z(R, \phi_1, \phi_2) R^2 \sin\phi_1 dR d\phi_1 d\phi_2. \tag{121}
$$

It is easy to verify that $\phi'(\boldsymbol{w}^{T}_{j_2} \boldsymbol{x})$ only depends on the direction of $\boldsymbol{x}$ and

$$
f_Z(R, \phi_1, \phi_2) = \frac{1}{(2\pi)^{\frac{3}{2}}} e^{\frac{x_1^2 + x_2^2 + x_3^2}{2}} = \frac{1}{(2\pi)^{\frac{3}{2}}} e^{\frac{R^2}{2}}
$$

only depends on $R$. Then, we have

$$
\begin{aligned}
&I(i_2, j_2)\\
&= \int |\phi'\big(\boldsymbol{w}_{j_2}^T(\boldsymbol{x}/R)\big) - \phi'\big(\boldsymbol{w}_{j_2}^{*T}(\boldsymbol{x}/R)\big)| \cdot |R\cos\phi_1|^2 \cdot f_Z(R)R^2 \sin\phi_1 dR d\phi_1 d\phi_2\\
&= \int_0^\infty r^4 f_z(R)dR \int_0^\pi \int_0^{2\pi} |\cos\phi_1|^2 \cdot \sin\phi_1 \cdot |\phi'\big(\boldsymbol{w}_{j_2}^T(\boldsymbol{x}/R)\big) - \phi'\big(\boldsymbol{w}_{j_2}^{*T}(\boldsymbol{x}/R)\big)| d\phi_1 d\phi_2\\
&\leq \sqrt{\frac{8}{\pi}} \int_0^\infty R^2 f_z(R)dR \int_0^\pi \int_0^{2\pi} \sin\phi_1 \cdot |\phi'\big(\boldsymbol{w}_{j_2}^T(\boldsymbol{x}/R)\big) - \phi'\big(\boldsymbol{w}_{j_2}^{*T}(\boldsymbol{x}/R)\big)| d\phi_1 d\phi_2\\
&= \sqrt{\frac{8}{\pi}} \mathbb{E}_{z_1,z_2,z_3} |\phi'\big(\boldsymbol{w}_{j_2}^T\boldsymbol{x}\big) - \phi'\big(\boldsymbol{w}_{j_2}^{*T}\boldsymbol{x}\big)|\\
&= \sqrt{\frac{8}{\pi}} \mathbb{E}_{\boldsymbol{x}} |\phi'\big(\boldsymbol{w}_{j_2}^T\boldsymbol{x}\big) - \phi'\big(\boldsymbol{w}_{j_2}^{*T}\boldsymbol{x}\big)|.
\end{aligned}
\tag{122}
$$

Define a set $\mathcal{A}_1 = \{\boldsymbol{x} | (\boldsymbol{w}_{j_2}^{*T}\boldsymbol{x})(\boldsymbol{w}_{j_2}^T\boldsymbol{x}) < 0\}$. If $\boldsymbol{x} \in \mathcal{A}_1$, then $\boldsymbol{w}_{j_2}^{*T}\boldsymbol{x}$ and $\boldsymbol{w}_{j_2}^T\boldsymbol{x}$ have different signs, which means the value of $\phi'(\boldsymbol{w}_{j_2}^T\boldsymbol{x})$ and $\phi'(\boldsymbol{w}_{j_2}^{*T}\boldsymbol{x})$ are different. This is equivalent to say that

$$
|\phi'(\boldsymbol{w}_{j_2}^T\boldsymbol{x}) - \phi'(\boldsymbol{w}_{j_2}^{*T}\boldsymbol{x})| = \begin{cases} 1, & \text{if } \boldsymbol{x} \in \mathcal{A}_1 \\ 0, & \text{if } \boldsymbol{x} \in \mathcal{A}_1^c \end{cases}.
\tag{123}
$$

Moreover, if $\boldsymbol{x} \in \mathcal{A}_1$, then we have

$$
|\boldsymbol{w}_{j_2}^{*T}\boldsymbol{x}| \leq |\boldsymbol{w}_{j_2}^{*T}\boldsymbol{x} - \boldsymbol{w}_{j_2}^T\boldsymbol{x}| \leq \|\boldsymbol{w}_{j_2}^* - \boldsymbol{w}_{j_2}\| \cdot \|\boldsymbol{x}\|.
\tag{124}
$$

Define a set $\mathcal{A}_2$ such that

$$
\mathcal{A}_2 = \left\{ \boldsymbol{x} \Big| \frac{|\boldsymbol{w}_{j_2}^{*T}\boldsymbol{x}|}{\|\boldsymbol{w}_{j_2}^*\|\|\boldsymbol{x}\|} \leq \frac{\|\boldsymbol{w}_{j_2}^* - \boldsymbol{w}_{j_2}\|}{\|\boldsymbol{w}_{j_2}^*\|} \right\} = \left\{ \theta_{\boldsymbol{x},\boldsymbol{w}_{j_2}^*} \Big| |\cos\theta_{\boldsymbol{x},\boldsymbol{w}_{j_2}^*}| \leq \frac{\|\boldsymbol{w}_{j_2}^* - \boldsymbol{w}_{j_2}\|}{\|\boldsymbol{w}_{j_2}^*\|} \right\}.
\tag{125}
$$

Hence, we have that

$$
\begin{aligned}
\mathbb{E}_{\boldsymbol{x}} |\phi'(\boldsymbol{w}_{j_2}^T\boldsymbol{x}) - \phi'(\boldsymbol{w}_{j_2}^{*T}\boldsymbol{x})|^2 &= \mathbb{E}_{\boldsymbol{x}} |\phi'(\boldsymbol{w}_{j_2}^T\boldsymbol{x}) - \phi'(\boldsymbol{w}_{j_2}^{*T}\boldsymbol{x})|\\
&= \text{Prob}(\boldsymbol{x} \in \mathcal{A}_1)\\
&\leq \text{Prob}(\boldsymbol{x} \in \mathcal{A}_2).
\end{aligned}
\tag{126}
$$

Since $\boldsymbol{x} \sim \mathcal{N}(\boldsymbol{0}, \boldsymbol{I})$, $\theta_{\boldsymbol{x},\boldsymbol{w}_{j_2}^*}$ belongs to the uniform distribution on $[-\pi, \pi]$, we have

$$
\begin{aligned}
\text{Prob}(\boldsymbol{x} \in \mathcal{A}_2) = \frac{\pi - \arccos\frac{\|\boldsymbol{w}_{j_2}^* - \boldsymbol{w}_{j_2}\|}{\|\boldsymbol{w}_{j_2}^*\|}}{\pi} &\leq \frac{1}{\pi}\tan(\pi - \arccos\frac{\|\boldsymbol{w}_{j_2}^* - \boldsymbol{w}_{j_2}\|}{\|\boldsymbol{w}_{j_2}^*\|})\\
&= \frac{1}{\pi}\cot(\arccos\frac{\|\boldsymbol{w}_{j_2}^* - \boldsymbol{w}_{j_2}\|}{\|\boldsymbol{w}_{j_2}^*\|})\\
&\leq \frac{2}{\pi}\frac{\|\boldsymbol{w}_{j_2}^* - \boldsymbol{w}_{j_2}\|}{\|\boldsymbol{w}_{j_2}^*\|}\\
&= \frac{2}{\pi}\frac{\|\boldsymbol{w}_{j_2,\Omega_{j_2}}^* - \boldsymbol{w}_{j_2,\Omega_{j_2}}\|}{\|\boldsymbol{w}_{j_2,\Omega_{j_2}}^*\|}\\
&\leq \frac{2}{\pi}\frac{\|\widetilde{\boldsymbol{w}}^* - \widetilde{\boldsymbol{w}}\|}{\sigma_K}.
\end{aligned}
\tag{127}
$$

Hence, (122) and (127) suggest that

$$
I \leq \frac{6}{\pi}\frac{\|\widetilde{\boldsymbol{w}}^* - \widetilde{\boldsymbol{w}}\|}{\sigma_K}.
\tag{128}
$$

The same bound that shown in (128) holds for $\boldsymbol{I}_2$ as well.

Therefore, we have

$$
\begin{aligned}
&\|\nabla^2 f(\widetilde{\boldsymbol{w}}) - \nabla^2 f(\widetilde{\boldsymbol{w}}^*)\|_2 \\
&\leq \sum_{j_1=1}^K \sum_{j_2=1}^K \left\| \frac{\partial^2 f(\widetilde{\boldsymbol{w}}^*)}{\partial \boldsymbol{w}_{j_1,\Omega_{j_1}} \partial \boldsymbol{w}_{j_2,\Omega_{j_2}}} - \frac{\partial^2 f(\widetilde{\boldsymbol{w}})}{\partial \boldsymbol{w}_{j_1,\Omega_{j_1}} \partial \boldsymbol{w}_{j_2,\Omega_{j_2}}} \right\|_2 \\
&\leq \|\boldsymbol{I}_1 + \boldsymbol{I}_2\|_2 \leq \|\boldsymbol{I}_1\|_2 + \|\boldsymbol{I}_2\|_2 \\
&\leq \frac{12}{\pi} \frac{\|\widetilde{\boldsymbol{w}}^* - \widetilde{\boldsymbol{w}}\|_2}{\sigma_K}
\end{aligned}
\tag{129}
$$

$\square$

# J  Additional proofs of lemmas in Appendix F

## J.1  Error bound for the second-order moment

*Proof of Lemma 7.* For $\widehat{\boldsymbol{M}}_2 - \boldsymbol{M}_2$, we have

$$
\begin{aligned}
&\widehat{\boldsymbol{M}}_2 - \boldsymbol{M}_2 \\
&= \frac{1}{N} \sum_{n=1}^N y_n (\widetilde{\boldsymbol{x}}_n \otimes \widetilde{\boldsymbol{x}}_n - \mathbb{E}\widetilde{\boldsymbol{x}}_n \widetilde{\boldsymbol{x}}_n^T) - \mathbb{E}_{\boldsymbol{x}}\, y(\widetilde{\boldsymbol{x}} \otimes \widetilde{\boldsymbol{x}} - \mathbb{E}\widetilde{\boldsymbol{x}}\widetilde{\boldsymbol{x}}^T) \\
&= \frac{1}{N} \sum_{n=1}^N \Big( \frac{1}{K} \sum_{j=1}^K \phi(\boldsymbol{u}_j^{*T} \widetilde{\boldsymbol{x}}_{n,\widetilde{\Omega}_j} + \xi_n)(\widetilde{\boldsymbol{x}}_n \otimes \widetilde{\boldsymbol{x}}_n - \mathbb{E}\widetilde{\boldsymbol{x}}_n \widetilde{\boldsymbol{x}}_n^T) \Big) \\
&\quad - \mathbb{E}_{\boldsymbol{x}} \frac{1}{K} \sum_{j=1}^K \phi(\boldsymbol{u}_j^{*T} \widetilde{\boldsymbol{x}}_{\widetilde{\Omega}_j})(\widetilde{\boldsymbol{x}} \otimes \widetilde{\boldsymbol{x}} - \mathbb{E}\widetilde{\boldsymbol{x}}\widetilde{\boldsymbol{x}}^T) \\
&= \frac{1}{K \cdot N} \sum_{n=1}^N \sum_{j=1}^K \Big( \phi(\boldsymbol{u}_j^{*T} \widetilde{\boldsymbol{x}}_{n,\widetilde{\Omega}_j})(\widetilde{\boldsymbol{x}}_n \otimes \widetilde{\boldsymbol{x}}_n - \mathbb{E}\widetilde{\boldsymbol{x}}_n \widetilde{\boldsymbol{x}}_n^T) - \mathbb{E}_{\boldsymbol{x}}\, \phi(\boldsymbol{u}_j^{*T} \widetilde{\boldsymbol{x}}_{\widetilde{\Omega}_j})(\widetilde{\boldsymbol{x}} \otimes \widetilde{\boldsymbol{x}} - \mathbb{E}\widetilde{\boldsymbol{x}}\widetilde{\boldsymbol{x}}^T) \Big) \\
&\quad + \frac{1}{N} \sum_{n=1}^N \xi_n (\widetilde{\boldsymbol{x}}_n \otimes \widetilde{\boldsymbol{x}}_n - \mathbb{E}\widetilde{\boldsymbol{x}}_n \widetilde{\boldsymbol{x}}_n^T)
\end{aligned}
\tag{130}
$$

Following the notations in Lemma E.2 of [40], we denote

$$
\boldsymbol{B}_2(\boldsymbol{x}_n) := \frac{1}{K} \sum_{j=1}^K \phi(\boldsymbol{u}_j^{*T} \widetilde{\boldsymbol{x}}_{n,\widetilde{\Omega}_j})(\widetilde{\boldsymbol{x}}_n \otimes \widetilde{\boldsymbol{x}}_n - \mathbb{E}\widetilde{\boldsymbol{x}}_n \widetilde{\boldsymbol{x}}_n^T).
\tag{131}
$$

Following the similar calculations of (I) - (III) in Lemma E.2 [40], we know that

$$
\begin{aligned}
\|\boldsymbol{B}_2(\boldsymbol{x})\|_2 &\lesssim \sigma_1 r_{\max} \log^{\frac{3}{2}} q, \\
\|\mathbb{E}_{\boldsymbol{x}} \boldsymbol{B}_2(\boldsymbol{x})\|_2 &\lesssim \sigma_1, \\
\|\mathbb{E}_{\boldsymbol{x}} \boldsymbol{B}_2^2(\boldsymbol{x})\|_2 &\lesssim \frac{1}{K} \sigma_1^2 r_{\max}
\end{aligned}
\tag{132}
$$

hold with probability at least $1 - q^{-r_{\max}}$.

Define $\boldsymbol{Z}_{2,n} = \frac{1}{N}\big(\boldsymbol{B}_2(\boldsymbol{x}_n) - \mathbb{E}_{\boldsymbol{x}} \boldsymbol{B}_2(\boldsymbol{x})\big)$ for $\boldsymbol{x}_n$ with $n \in [N]$, and it is obvious $\boldsymbol{Z}_{2,n}$ is zero mean. Also, we have

$$
R_2 = \|\boldsymbol{Z}_{2,n}\|_2 \leq \frac{1}{N}\big(\|\boldsymbol{B}_2(\boldsymbol{x}_n)\|_2 + \|\mathbb{E}_{\boldsymbol{x}} \boldsymbol{B}_2(\boldsymbol{x})\|_2\big) \lesssim \frac{1}{K} \sum_{j=1}^K N^{-1} \sigma_1 r_k \log^{\frac{3}{2}} q,
\tag{133}
$$

and

$$\delta_2^2 = \Big\| \sum_{n=1}^{N} \mathbb{E} \mathbf{Z}_{2,n}^2 \Big\|_2^2 \leq \Big\| \sum_{n=1}^{N} \frac{1}{N^2} \Big( \mathbb{E} \mathbf{B}_2^2(\mathbf{x}_n) - \big( \mathbb{E} \mathbf{B}_2(\mathbf{x}_n) \big)^2 \Big) \Big\|_2 \tag{134}$$

$$\leq \frac{1}{N} \Big( \| \mathbb{E} \mathbf{B}_2^2(\mathbf{x}_n) \|_2 + \| \mathbb{E} \mathbf{B}_2(\mathbf{x}_n) \|_2^2 \Big)$$

$$\lesssim N^{-1} \sigma_1^2 r_{\max}.$$

Next, let $t = \Theta(\sigma_1 \sqrt{\frac{r_{\max} \log q}{N}})$. To make sure $\delta_2^2 \geq R_2 t/3$, we need $N \gtrsim r_{\max} \log^4 q$. Then, by Lemma 10, we have

$$\text{Prob}\Big\{ \Big\| \sum_n \mathbf{Z}_{2,n} \Big\|_2 \geq t \Big\} \leq 2r \exp \Big( \frac{-t^2/2}{\delta_2^2 + R_2 t/3} \Big) \leq 2r \exp \Big( \frac{-t^2}{4\delta_2^2} \Big). \tag{135}$$

That is

$$\Big\| \sum_{n=1}^{N} \mathbf{Z}_{2,n} \Big\|_2 \lesssim \sigma_1 \sqrt{\frac{r_{\max} \log q}{N}} \tag{136}$$

with probability at least $1 - q^{-r_{\max}}$. Because $\widetilde{\mathbf{x}}_n$ belongs to the sub-Gaussian distribution, we know that

$$\Big\| \frac{1}{N} \sum_{n=1}^{N} (\widetilde{\mathbf{x}}_n \otimes \widetilde{\mathbf{x}}_n - \mathbb{E} \widetilde{\mathbf{x}}_n \widetilde{\mathbf{x}}_n^T) \Big\|_2 \lesssim \sqrt{\frac{r_{\max} \log q}{N}} \tag{137}$$

with probability at least $1 - q^{-r_{\max}}$.

In conclusion, we have

$$\| \widehat{\mathbf{M}}_2 - \mathbf{M}_2 \| \frac{1}{K} \sum_{k=1}^{K} \lesssim (\sigma_1 + |\xi|) \sqrt{\frac{r_{\max} \log q}{N}} \tag{138}$$

with probability at least $1 - q^{-r_{\max}}$ provided that $N \gtrsim r_{\max} \log^4 q$. $\qquad\square$

## J.2   Error bound for the third-order moment

*Proof of Lemma 8.* For $\widehat{\mathbf{M}}_3(\widehat{\mathbf{V}}, \widehat{\mathbf{V}}, \widehat{\mathbf{V}}) - \mathbf{M}_3(\widehat{\mathbf{V}}, \widehat{\mathbf{V}}, \widehat{\mathbf{V}})$, we have

$$\widehat{\mathbf{M}}_3(\widehat{\mathbf{V}}, \widehat{\mathbf{V}}, \widehat{\mathbf{V}}) - \mathbf{M}_3(\widehat{\mathbf{V}}, \widehat{\mathbf{V}}, \widehat{\mathbf{V}})$$

$$= \frac{1}{N} \sum_{n=1}^{N} y_n \big[ (\widehat{\mathbf{V}}^T \widetilde{\mathbf{x}}_n)^{\otimes 3} - (\widehat{\mathbf{V}}^T \widetilde{\mathbf{x}}_n) \otimes (\mathbb{E}(\widehat{\mathbf{V}}^T \widetilde{\mathbf{x}}_n)(\widehat{\mathbf{V}}^T \widetilde{\mathbf{x}}_n)^T) \big]$$

$$\quad - \mathbb{E}_{\mathbf{x}} y \big[ (\widehat{\mathbf{V}}^T \widetilde{\mathbf{x}})^{\otimes 3} - (\widehat{\mathbf{V}}^T \mathbf{x}^T) \otimes \mathbb{E}(\widehat{\mathbf{V}}^T \widetilde{\mathbf{x}}_n)(\widehat{\mathbf{V}}^T \widetilde{\mathbf{x}}_n)^T \big]$$

$$= \frac{1}{N} \sum_{n=1}^{N} \Big( \frac{1}{K} \sum_{j=1}^{K} \phi(\mathbf{u}_j^{*T} \widetilde{\mathbf{x}}_{n,\widetilde{\Omega}_j}) + \xi_n \Big) \cdot \big[ (\widehat{\mathbf{V}}^T \widetilde{\mathbf{x}}_n)^{\otimes 3} - (\widehat{\mathbf{V}}^T \widetilde{\mathbf{x}}_n) \otimes (\mathbb{E}(\widehat{\mathbf{V}}^T \widetilde{\mathbf{x}}_n)(\widehat{\mathbf{V}}^T \widetilde{\mathbf{x}}_n)^T)) \big]$$

$$\quad - \mathbb{E}_{\mathbf{x}} \frac{1}{K} \sum_{j=1}^{K} \phi(\mathbf{u}_j^{*T} \widetilde{\mathbf{x}}_{\widetilde{\Omega}_j}) \big[ (\widehat{\mathbf{V}}^T \widetilde{\mathbf{x}})^{\otimes 3} - (\widehat{\mathbf{V}}^T \widetilde{\mathbf{x}}) \otimes (\mathbb{E}(\widehat{\mathbf{V}}^T \widetilde{\mathbf{x}})(\widehat{\mathbf{V}}^T \widetilde{\mathbf{x}})^T)) \big] \tag{139}$$

$$= \frac{1}{K \cdot N} \sum_{n=1}^{N} \sum_{j=1}^{K} \Big[ \phi(\mathbf{u}_j^{*T} \widetilde{\mathbf{x}}_{n,\widetilde{\Omega}_j}) \cdot \big[ (\widehat{\mathbf{V}}^T \widetilde{\mathbf{x}}_n)^{\otimes 3} - (\widehat{\mathbf{V}}^T \widetilde{\mathbf{x}}_n) \otimes (\mathbb{E}(\widehat{\mathbf{V}}^T \widetilde{\mathbf{x}}_n)(\widehat{\mathbf{V}}^T \widetilde{\mathbf{x}}_n)^T)) \big]$$

$$\quad - \mathbb{E}_{\mathbf{x}} \phi(\mathbf{u}_j^{*T} \widetilde{\mathbf{x}}_{\widetilde{\Omega}_j}) \big[ (\widehat{\mathbf{V}}^T \widetilde{\mathbf{x}})^{\otimes 3} - (\widehat{\mathbf{V}}^T \widetilde{\mathbf{x}}) \otimes (\mathbb{E}(\widehat{\mathbf{V}}^T \widetilde{\mathbf{x}})(\widehat{\mathbf{V}}^T \widetilde{\mathbf{x}})^T)) \big] \Big]$$

$$\quad + \frac{1}{N} \sum_{n=1}^{N} \xi_n \big[ (\widehat{\mathbf{V}}^T \widetilde{\mathbf{x}})^{\otimes 3} - (\widehat{\mathbf{V}}^T \widetilde{\mathbf{x}}) \otimes (\mathbb{E}(\widehat{\mathbf{V}}^T \widetilde{\mathbf{x}})(\widehat{\mathbf{V}}^T \widetilde{\mathbf{x}})^T)) \big]$$

Following the notations in Lemma E.8 of [40], we define

$$\boldsymbol{T}(\boldsymbol{x}) := \frac{1}{K} \sum_{j=1}^{K} \phi(\boldsymbol{u}_j^{*T} \widetilde{\boldsymbol{x}}_{n,\widetilde{\Omega}_j}) \cdot \left[ (\widehat{\boldsymbol{V}}^T \boldsymbol{x}_n)^{\otimes 3} - (\widehat{\boldsymbol{V}}^T \boldsymbol{x}_n) \otimes (\mathbb{E}(\widehat{\boldsymbol{V}}^T \boldsymbol{x}_n)(\widehat{\boldsymbol{V}}^T \boldsymbol{x}_n)^T) \right]. \quad (140)$$

Then, $\boldsymbol{B}_3(\boldsymbol{x}) \in \mathbb{R}^{K \times K^2}$ is defined as flattening the tensor $\boldsymbol{T}(\boldsymbol{x})$ along the first dimension. Hence, we have

$$\|\boldsymbol{B}_3(\boldsymbol{x})\|_2 \lesssim \max_j |\boldsymbol{u}_j^* \widetilde{\boldsymbol{x}}_{\widetilde{\Omega}_j}| \cdot \left( \|\widehat{\boldsymbol{V}}^T \boldsymbol{x}_n\|_2^3 + 3K \|\widehat{\boldsymbol{V}}^T \boldsymbol{x}_n\|_2 \right)$$
$$\lesssim \sigma_1 K^{\frac{3}{2}} \log^{\frac{5}{2}} q \quad (141)$$

with probability at least $1 - q^{-K}$.

Following the similar calculations of (II) and (III) in Lemma E.8 of [40], we know that

$$\|\mathbb{E}_{\boldsymbol{x}} \boldsymbol{B}_3(\boldsymbol{x})\|_2 \lesssim \sigma_1,$$
$$\max \left\{ \left\| \mathbb{E}_{\boldsymbol{x}} [\boldsymbol{B}_3(\boldsymbol{x})^T \boldsymbol{B}_3(\boldsymbol{x})] \right\|_2, \left\| \mathbb{E}_{\boldsymbol{x}} [\boldsymbol{B}_3(\boldsymbol{x})^T \boldsymbol{B}_3(\boldsymbol{x})] \right\|_2 \right\} \lesssim K^2 \sigma_1^2. \quad (142)$$

Define $\boldsymbol{Z}_{3,n} = \frac{1}{N} (\boldsymbol{B}_3(\boldsymbol{x}_n) - \mathbb{E}_{\boldsymbol{x}} \boldsymbol{B}_3(\boldsymbol{x}))$ for $(\boldsymbol{x}_n, y_n) \in \mathcal{D}$, and it is obvious $\boldsymbol{Z}_{3,n}$ is zero mean. Also, we have

$$R_3 = \|\boldsymbol{Z}_{3,n}\|_2 \leq \frac{1}{N} \left( \|\boldsymbol{B}_3(\boldsymbol{x}_n)\|_2 + \|\mathbb{E}_{\boldsymbol{x}} \boldsymbol{B}_3(\boldsymbol{x})\|_2 \right)$$
$$\lesssim N^{-1} \sigma_1 K^{\frac{3}{2}} \log^{\frac{5}{2}} q, \quad (143)$$

and

$$\delta_3^2 = \left\{ \left\| \sum_{n=1}^{N} \mathbb{E} \boldsymbol{Z}_{3,n} \boldsymbol{Z}_{3,n}^T \right\|_2, \left\| \sum_{n=1}^{N} \mathbb{E} \boldsymbol{Z}_{3,n} \boldsymbol{Z}_{3,n}^T \right\|_2 \right\} \leq \frac{1}{N} \left( \|\mathbb{E} \boldsymbol{B}_3^2(\boldsymbol{x}_n)\|_2 + \|\mathbb{E} \boldsymbol{B}_3(\boldsymbol{x}_n)\|_2^2 \right) \quad (144)$$
$$\lesssim N^{-1} K^2 \sigma_1^2.$$

Similar to (135), by applying Lemma 10, we have

$$\left\| \sum_{n=1}^{N} \boldsymbol{Z}_{3,n} \right\|_2 \lesssim \sigma_1 \sqrt{\frac{\log q}{N}} \quad (145)$$

with probability at least $1 - q^{-K}$ provided that $N \gtrsim K^5 \log^6 q$.

Similar to (141), we define $\boldsymbol{B}$ by flattening the tensor $\sum_{n=1}^{N} \left[ (\widehat{\boldsymbol{V}}^T \widetilde{\boldsymbol{x}})^{\otimes 3} - (\widehat{\boldsymbol{V}}^T \widetilde{\boldsymbol{x}}) \otimes (\mathbb{E}(\widehat{\boldsymbol{V}}^T \widetilde{\boldsymbol{x}})(\widehat{\boldsymbol{V}}^T \widetilde{\boldsymbol{x}})^T) \right]$ along the first dimension. Then, we know that

$$\|\boldsymbol{B}\|_2 \leq \left\| \sum_{n=1}^{N} \widehat{\boldsymbol{V}}^T \widetilde{\boldsymbol{x}}_n \right\|_2^3 + 3K \left\| \sum_{n=1}^{N} \widehat{\boldsymbol{V}}^T \widetilde{\boldsymbol{x}}_n \right\|_2 \lesssim \left( \frac{K^{-4} \log q}{N} \right)^{\frac{3}{2}} + 3K \left( \frac{K^{-4} \log q}{N} \right)^{\frac{1}{2}}$$
$$\lesssim \left( \frac{\log q}{N} \right)^{\frac{1}{2}} + \left( \frac{\log q}{N} \right)^{\frac{1}{2}} \quad (146)$$
$$\lesssim \sqrt{\frac{\log q}{N}},$$

provided that $N \gtrsim K^5 \log q$.

In conclusion, we have

$$\left\| \widehat{\boldsymbol{M}}_3(\widehat{\boldsymbol{V}}, \widehat{\boldsymbol{V}}, \widehat{\boldsymbol{V}}) - \boldsymbol{M}_3(\widehat{\boldsymbol{V}}, \widehat{\boldsymbol{V}}, \widehat{\boldsymbol{V}}) \right\| \lesssim (\sigma_1 + |\xi|) \sqrt{\frac{\log q}{N}} \quad (147)$$

with probability at least $1 - q^{-K}$ provided that $N \gtrsim K^3 \log^6 q$.

$\square$

## J.3 Error bound for the first-order moment

*Proof of Lemma 9.* For $\widehat{\boldsymbol{M}}_1 - \boldsymbol{M}_1$, we have

$$
\begin{aligned}
\widehat{\boldsymbol{M}}_1 - \boldsymbol{M}_1 =& \frac{1}{N} \sum_{n=1}^{N} y_n \widetilde{\boldsymbol{x}}_n - \mathbb{E}_{\boldsymbol{x}} \, y \widetilde{\boldsymbol{x}} \\
=& \frac{1}{N} \sum_{n=1}^{N} \Big( \frac{1}{K} \sum_{j=1}^{K} \phi(\boldsymbol{u}_j^{*T} \widetilde{\boldsymbol{x}}_{n,\widetilde{\Omega}_j}) + \xi_n \Big) \widetilde{\boldsymbol{x}}_n - \mathbb{E}_{\boldsymbol{x}} \sum_{j=1}^{K} \frac{1}{K} \phi(\boldsymbol{u}_j^{*T} \widetilde{\boldsymbol{x}}_{\widetilde{\Omega}_j}) \widetilde{\boldsymbol{x}} \qquad (148) \\
=& \frac{1}{K \cdot N} \sum_{j=1}^{K} \sum_{n=1}^{N} \Big( \phi(\boldsymbol{u}_j^{*T} \widetilde{\boldsymbol{x}}_{n,\widetilde{\Omega}_j}) \widetilde{\boldsymbol{x}}_n - \mathbb{E}_{\boldsymbol{x}} \, \phi(\boldsymbol{u}_j^{*T} \widetilde{\boldsymbol{x}}_{\widetilde{\Omega}_j}) \widetilde{\boldsymbol{x}} \Big) + \frac{1}{N} \sum_{n=1}^{N} \xi_n \cdot \widetilde{\boldsymbol{x}}_n.
\end{aligned}
$$

Define $\boldsymbol{B}_1(\boldsymbol{x}) := \frac{1}{K} \sum_{j=1}^{K} \phi(\boldsymbol{u}_j^{*T} \widetilde{\boldsymbol{x}}_{n,\widetilde{\Omega}_j}) \widetilde{\boldsymbol{x}}_n$, then we have

$$
\begin{aligned}
\|\boldsymbol{B}_1(\boldsymbol{x})\|_2 &\lesssim \frac{1}{K} \sum_{k=1}^{K} \sigma_1 r_k \log^{\frac{3}{2}} q; \\
\|\mathbb{E}_{\boldsymbol{x}} \boldsymbol{B}_1(\boldsymbol{x})\|_2 &\lesssim \sigma_1; \\
\Big\{ \big\| \boldsymbol{E}_{\boldsymbol{x}} [\boldsymbol{B}_1(\boldsymbol{x}) \boldsymbol{B}_1(\boldsymbol{x})^T] \big\|_2, \big\| \boldsymbol{E}_{\boldsymbol{x}} [\boldsymbol{B}_{1,j}(\boldsymbol{x})^T \boldsymbol{B}_1(\boldsymbol{x})] \big\|_2 \Big\} &\lesssim \sigma_1^2.
\end{aligned} \qquad (149)
$$

Next, define $\boldsymbol{Z}_{1,n} = \frac{1}{N} \big( \boldsymbol{B}_{1,j}(\boldsymbol{x}_n) - \mathbb{E}_{\boldsymbol{x}} \boldsymbol{B}_2(\boldsymbol{x}) \big)$ for $(\boldsymbol{x}_n, y_n) \in \mathcal{D}$, by calculation, we can obtain

$$
R_1 = \|\boldsymbol{Z}_{1,n}\|_2 \lesssim N^{-1} \sigma_1 r_{\max} \log^{\frac{3}{2}} q, \qquad (150)
$$

and

$$
\delta_1^2 = \max \Big\{ \Big\| \sum_{n=1}^{N} \mathbb{E} \boldsymbol{Z}_{1,n} \boldsymbol{Z}_{1,n}^T \Big\|_2^2, \Big| \sum_{n=1}^{N} \boldsymbol{Z}_{1,n}^T \boldsymbol{Z}_{1,n} \Big| \Big\} \lesssim N^{-1} \sigma_1^2 r_{\max}. \qquad (151)
$$

By applying Lemma 10, we have

$$
\Big\| \sum_{n=1}^{N} \boldsymbol{Z}_{1,n} \Big\|_2 \lesssim \sigma_1 \sqrt{\frac{r_{\max} \log q}{N}} \qquad (152)
$$

with probability at least $1 - q^{-r_{\max}}$ provided that $N \gtrsim r_{\max} \log^4 q$. Since $\boldsymbol{x} \in \mathbb{R}^r$ belongs to the Gaussian distribution, we have

$$
\Big\| \frac{1}{N} \sum_{n=1}^{N} \widetilde{\boldsymbol{x}}_n \Big\|_2 \lesssim \sqrt{\frac{r_{\max} \log q}{N}} \qquad (153)
$$

with probability at least $1 - q^{-r_{\max}}$.

In conclusion, we have

$$
\|\widehat{\boldsymbol{M}}_1 - \boldsymbol{M}_1\| \lesssim (\sigma_1 + |\xi|) \sqrt{\frac{r_{\max} \log q}{N}} \qquad (154)
$$

with probability at least $1 - q^{-r_{\max}}$, provided that $N \gtrsim r_{\max} \log^4 q$. $\qquad \square$