# OpenReview forum: "Why Lottery Ticket Wins? A Theoretical Perspective of Sample Complexity on Sparse Neural Networks"
_NeurIPS.cc/2021/Conference — NeurIPS 2021 Poster_

### Official Review · Reviewer_1uan · 2021-07-15

**Rating:** 7
**Confidence:** 3

**Summary:**

This paper presents a theoretical analysis of the advantages of learning pruned neural networks . This analysis considers a teacher-student setup with a finite number of training samples. The theoretical results presented in the paper show that pruned networks can have multiple advantages, such as faster training convergence, a lower number of samples required for successful convergence and an enlarged convex region.

**Main Review:**

Pros:

This paper tries to give some theoretical insights that explain the performance of pruned neural networks. I agree with the authors that this problem has been studied mostly empirically and a deeper theoretical analysis can provide a better understanding of the behaviour of pruned neural networks. For this reason, I believe that this paper can be a valid contribution in this direction.

Cons:

 I think that the analysis presented in the paper has some limitations: i) as highlighted by the authors in Sec. 3.4, this analysis considers only one hidden layer and assumes that the input follows the Gaussian distribution; ii) the authors assume that the whole training data is used at each iteration (instead of using minibatches).
For these reasons, it is not clear to me if the theoretical results presented in the paper can be useful when we consider a real setting. Are there any useful insights that can be used in real applications to, e.g., improve the training of pruned neural networks or find better winning tickets? The experiments on real datasets presented in the paper are quite limited and show only the effect of sample complexity, which is not very informative and does not reflect what happens in real applications since usually training is performed using minibatches and does not use the whole training set at each iteration.

Minor comments:

Many details of the theoretical analysis are discussed only in the appendix. Some of them, such as the tensor initialization method, are fundamental in order to understand the theoretical results presented in the paper. I suggest to try to add them to the main paper. This would increases the paper readability.


**Time Spent Reviewing:**

2

---

> ### Author Response · Authors · 2021-08-07
> **Response to the comments by Reviewer 1uan**
>
> Thanks for your comments.
>
> $\textbf{[Clarification on mini-batch setups]}$ We want to first clarify that the experiments for real datasets and part of the synthetic datasets are performed using mini-batches and do not use the whole training set at each iteration. Also, the proposed algorithm (see in line 180) belongs to mini-batch gradient descent algorithms. In the following paragraphs, we will provide the point-to-point response to the concerns mentioned in comments.
>
> $\textbf{Q1}$: “this analysis considers only one hidden layer and assumes that the input follows the Gaussian distribution.”
>
> $\textbf{A1}$: We admit that there are still gaps between theoretical analyses and numerical results in assumptions. However, due to the complexity of neural networks, it is currently impossible to theoretically characterize the behavior of a neural network (NN) without any assumption or regularization.
>
> (Gaussian input distribution.) Gaussian distribution is currently a common assumption for theoretical works especially under the setup of a finite number of samples [8, 16, 46, 63, 67]. This assumption is motivated by the data whitening and batch normalization techniques that are commonly used in practice to improve the learning performance. The  learning  method  converges  faster  if  the  inputs  are whitened  to  be  the  standard  Gaussian [L1].  Batch  normalization [L2]  modifies  the  mean  and  variance  in  each  layer  and  is  a popular practical method to achieve fast and stable convergence. For example, the python code “transforms.Normalize” is a standard data processing method and also used in our simulation experiments when loading the datasets like MINST, Cifar-10, Cifar-100 and SVHN. Moreover, analyzing the standard Gaussian distribution also provides an insight into understanding rotation-invariant distributions. On the other hand, there exist major issues in developing the theoretical explanations for general distributions. First, the landscape of the population risk function depends on the distribution, and completely new tools need to be developed to analyze the landscape for general input distributions. Second, our objective is to reduce both training error and testing error, and that requires the empirical risk function to be a good approximation of the population risk function, at least in the neighborhood of $W^*$. Otherwise, a model with a small training error may have a large test error. The Gaussian assumption helps to capture the distance of these two risk functions, and new concentration bounds are needed for general distributions.
>
> Thus, the standard Gaussian assumption is well-acceptable in the current theoretical analyses of neural networks, but we will be investigating the extension to other distributions.
>
> [L1] Y. LeCun, L. Bottou, G. B. Orr, and K.-R. Müller, “Efficient backprop,” in Neural Networks: Tricks of the Trade, Springer, 1998, pp. 9–50.
>
> [L2] S. Loffe and C. Szegedy, “Batch normalization: Accelerating deep network training by reducing internal covariate shift,” in International Conference on Machine Learning, 2015, pp. 448–456.
>
>
> (One-hidden-layer neural network.) The setup of the one-hidden-layer neural network enables us to analyze the convergence of the trained model to the optimal one with both zero training error and zero testing error in noiseless cases. Almost all the existing works that achieve both zero training and testing errors are limited to one-hidden-layer neural networks. To the best of our knowledge, only one paper can achieve this on two-hidden-layer fully-connected neural networks (https://arxiv.org/abs/1811.04918) under the setup of overparameterization, but the convergence rate is proved as sub-linear. Beyond two hidden layers, all the existing theoretical analyses can either achieve zero training error or develop bounds of the generalization gap (difference between the training error and the testing error), but no existing works can achieve both. Therefore, in terms of achieving both zero training and zero testing error, we believe the one-hidden-layer neural network is still the state-of-the-art setup.
>
> Nevertheless, to diminish the limitation of the one-hidden-layer case, the conclusions derived from the one-hidden-layer case are verified for multi-layer cases through numerical results as shown in Figures 11 & 12, which are conducted on Lenet-5 and Resnet-50, respectively. One can observe that the results for real data and multi-layer neural networks coincide with our findings for one-hidden-layer cases as well. For example, a properly pruned network (i.e., winning ticket) helps reduce the sample complexity required to reach the same test accuracy as the original dense model.
>
>
> $\textbf{Q2}$: “The authors assume that the whole training data is used at each iteration (instead of using minibatches). For these reasons, it is not clear to me if the theoretical results presented in the paper can be useful when we consider a real setting. Are there any useful insights that can be used in real applications to, e.g., improve the training of pruned neural networks or find better winning tickets? The experiments on real datasets presented in the paper are quite limited and show only the effect of sample complexity, which is not very informative and does not reflect what happens in real applications since usually training is performed using minibatches and does not use the whole training set at each iteration.”
>
> $\textbf{A2}$: Sorry for the possible confusion caused by the statement “we use the whole training data instead of a fresh subset in each iteration” in line 264. We implemented the full gradient descent only in the experiments in section 4.1 to speed up the convergence due to the simple setup with a relatively small number of synthetic training data. All the experiments on the practical datasets in Section 4.3 and the synthetic datasets in Section 4.2 are conducted using mini-batch methods. In fact, the algorithm analyzed in this paper (see in line 180) is a mini-batch stochastic gradient descent method. If the reviewer is interested, we can replace all the experiments in section 4.1 with a mini-batch method as well.
>
> $\textbf{Q3}$: “Many details of the theoretical analysis are discussed only in the appendix. Some of them, such as the tensor initialization method, are fundamental in order to understand the theoretical results presented in the paper. I suggest trying to add them to the main paper. This would increase the paper readability.”
>
> $\textbf{A3}$: Thanks for your suggestion. We will further polish our paper and include part of the critical analysis and the sketch of proofs in the revised paper.

---

> > ### Comment · Reviewer_1uan · 2021-08-20
> > **Response to rebuttal**
> >
> > Thank you very much for the response to my comments. In particular, the clarification about the use of mini-batches makes the analysis more valuable.
> >
> > I agree that the assumptions made in the theoretical analysis (gaussianity of input data and one-hidden-layer neural network) are common in neural network analysis and removing them might be unfeasible.
> >
> > For these reasons, I have increased my score to 7 (accept).

---

> > > ### Author Response · Authors · 2021-08-20
> > > **Thank you**
> > >
> > > Thank you very much for raising the score. We are very glad to learn that our responses have successfully made clarification about your confusions, and we will revise the paper accordingly to avoid the confusions as well.

---

### Official Review · Reviewer_Udgq · 2021-07-16

**Rating:** 8
**Confidence:** 3

**Summary:**

This work takes a theoretical view of LTH, leveraging the geometric structure of the objective function to analyze the generalization error of a pruned network trained in a teacher-student fashion. In particular, they prove that, as a model is pruned, the desirable (convex) region with high-guaranteed generalization performance enlarges, providing explanation for improved performance of winning tickets. Then, the sample complexity of training the pruned network to achieve zero generalization error is analyzed, finding that the number of samples required is proportional to the number of un-pruned weights. Interestingly, the work finds that pruned models enjoy faster convergence to high performance, providing another possible explanation of why winning tickets outperform dense networks.


**Limitations And Societal Impact:**

The authors discuss limitations of their work explicitly, see above review for more details. There is no negative societal impact of this work.

**Main Review:**

I begin my review by emphasizing that I am completely open to author feedback upon my review. My final score will be mostly based upon discussion with authors in regard to my points below.

General opinion:
The theoretical contributions of this work are very extensive/impressive. All theoretical results are extensively studied in simulated experiments, and experiments on real datasets are also provided. Although the problem setup within the work is somewhat peculiar (I have not seen such a student-teacher setup used within theoretical LTH analysis), I find the work to be well-written, interesting, and useful.

Pros:
- The analysis of the sample complexity required to train the pruned model to convergence seems novel and interesting. It is especially interesting that this sample complexity is found to be superior to the dense model from which the pruned model is derived.
- The incorporation of momentum into the gradient descent algorithm for pruning is interesting. The fact that acceleration is also achieved in theory is well-done.
- Analysis is also performed for the generalization performance of pruned networks, which is not as common in theoretical work for LTH. (The only others I know of are https://arxiv.org/abs/1802.05296 or https://arxiv.org/abs/1804.05345, which are both referenced).
- Theoretical analysis within this work, although based on arguments in [63], seems to be novel and requires numerous technical developments in comparison to previous work.
- Experiments on real datasets are included.

Cons:
- Analysis is limited to gaussian input data. I believe this is the main/most unrealistic limitation of the theoretical analysis.
- Theoretical analysis is only done one one-hidden-layer networks (though this is not truly a con, as this is true of much theoretical work for LTH and neural networks in general).

Questions:
- I do not fully understand why both teacher and student networks are pruned within the setup. Additionally, this teacher-student setup is not fully reflective of standard methodologies for LTH, so I am a bit curious about how this setup was derived (I assume this setup was adopted from [63], http://proceedings.mlr.press/v70/zhong17a.html?).
- The paper claims that the convex regions for pruned networks are enlarged. This means that pruned networks have a larger region in which linear convergence is achieved (if I am not mistaken). Although this may mean the pruned network converges faster due to the larger region of convexity, this does not say anything about the quality of the solution, correct? It seems like this analysis cannot be used as evidence for winning tickets having superior performance (though superiority is shown in other areas with sample complexity/generalization bounds).
- Is there any previous work that studies whether analysis with gaussian input data is reflective of practical behavior? I am unsure whether such an assumption is very limiting (possibly this determination can be made by discussing differences between synthetic/real experiments within the experimental section of this work).

Minor Comments:
- There are a lot of other papers that analyze over-parameterized neural networks that could be added to the related work (though maybe they are not as relevant to this work, I am not sure). For example, https://arxiv.org/abs/1902.04674
- I would recommend that the numerical experiments devote more space/attention to experiments on real datasets. Right now, it seems to be denominated by synthetic datasets, though I understand this is in an attempt to numerically verify aspects of the theory.


**Time Spent Reviewing:**

1.5

---

> ### Author Response · Authors · 2021-08-07
> **Point-to-point response to the concerns mentioned in Cons & Questions**
>
> Thanks for your comments, and we will provide the point-to-point response to the concerns mentioned in comments above.
>
> $\textbf{Q1}$: “Analysis is limited to Gaussian input data. I believe this is the main/most unrealistic limitation of theoretical analysis. Theoretical analysis is only done one one-hidden-layer networks (though this is not truly a con, as this is true of much theoretical work for LTH and neural networks in general).” & “Is there any previous work that studies whether analysis with gaussian input data is reflective of practical behavior? I am unsure whether such an assumption is very limiting (possibly this determination can be made by discussing differences between synthetic/real experiments within the experimental section of this work).”
>
> $\textbf{A1}$: We appreciate your understanding that there are still gaps between theoretical analyses and numerical results in assumptions, especially for pruned network training. In fact, due to the complexity of neural networks, it is really difficult to theoretically characterize the behavior of a neural network (NN) without any assumption or regularization.
>
> (Gaussian input distribution.) Gaussian distribution is currently a common assumption for theoretical works especially under the setup of a finite number of samples [8, 16, 46, 63, 67]. This assumption is motivated by the data whitening and batch normalization techniques that are commonly used in practice to improve learning performance. The learning method converges faster if  the inputs  are whitened to  be  the standard  Gaussian [L1].  Batch normalization [L2]  modifies  the mean and  variance  in  each layer  and  is  a popular practical method to achieve fast and stable convergence. For example, the python code “transforms. Normalize” is a standard data processing method and also used in our simulation experiments when loading the datasets like MINST, Cifar-10, Cifar-100 and SVHN. Moreover, analyzing the standard Gaussian distribution also provides an insight into understanding rotation-invariant distributions. On the other hand, there exist major issues in developing the theoretical explanations for general distributions. First, the landscape of the population risk function depends on the distribution, and completely new tools need to be developed to analyze the landscape for general input distributions. Second, our objective is to reduce both training error and testing error, and that requires the empirical risk function to be a good approximation of the population risk function, at least in the neighborhood of $W^*$. Otherwise, a model with a small training error may have a large test error. The Gaussian assumption helps to capture the distance of these two risk functions, and new concentration bounds are needed for general distributions.
>
> Thus, the standard Gaussian assumption is well-acceptable in the current theoretical analyses of neural networks, but we will be investigating the extension to other distributions.
>
> [L1] Y. LeCun, L. Bottou, G. B. Orr, and K.-R. Müller, “Efficient backprop,” in Neural Networks: Tricks of the Trade, Springer, 1998, pp. 9–50.
>
> [L2] S. Loffe and C. Szegedy, “Batch normalization: Accelerating deep network training by reducing internal covariate shift,” in International Conference on Machine Learning, 2015, pp. 448–456.
>
>
> (One-hidden-layer neural network.) The setup of the one-hidden-layer neural network enables us to analyze the convergence of the trained model to the optimal one with both zero training error and zero testing error in noiseless cases. Almost all the existing works that achieve both zero training and testing errors are limited to one-hidden-layer neural networks. To the best of our knowledge, only one paper can achieve this on two-hidden-layer fully-connected neural networks (https://arxiv.org/abs/1811.04918) under the setup of overparameterization, but the convergence rate is proved as sub-linear. Beyond two hidden layers, all the existing theoretical analyses can either achieve zero training error or develop bounds of the generalization gap (difference between the training error and the testing error), but no existing works can achieve both. Therefore, in terms of achieving both zero training and zero testing error, we believe the one-hidden-layer neural network is still the state-of-the-art setup.
>
> Nevertheless, to diminish the limitation of the one-hidden-layer case, the conclusions derived from the one-hidden-layer case are verified for multi-layer cases through numerical results as shown in Figures 11 & 12, which are conducted on Lenet-5 and Resnet-50, respectively. One can observe that the results for real data and multi-layer neural networks coincide with our findings for one-hidden-layer cases as well. For example, a properly pruned network (i.e., winning ticket) helps reduce the sample complexity required to reach the same test accuracy as the original dense model.
>
> $\textbf{Q2}$: “I do not fully understand why both teacher and student networks are pruned within the setup. Additionally, this teacher-student setup is not fully reflective of standard methodologies for LTH, so I am a bit curious about how this setup was derived (I assume this setup was adopted from [63]?).”
>
> $\textbf{A2}$: Sorry for the confusion caused by our teacher-student setup. The teacher-student setup in this paper does NOT refer to the setup of  knowledge distillation from a teacher model to a student model. We only borrow the terminology "teacher-student’’ to reflect that the training data are generated from an unknown optimal network, and the training is carried out on a student network with the hope of recovering the ground-truth model. Thus, the teacher model represents the oracle (ground-truth) model, and the student model stands for the learned model. The training process is indeed isolated, and no additional information is provided by the teacher model, i.e. no teacher-guided predictions. The input data and corresponding labels can be represented by an oracle model as shown in eqn. (1) in the paper. Given the data generated from the oracle model, the training process is conducted on a pruned neural network via solving eqn. (3) (just like a standard LTH implementation). During the training, there is no other additional information provided by the oracle model. Based on the reviewer’s comment, we plan to replace "teacher-student" with "oracle-leaner" to avoid confusion.
>
> $\textbf{Q3}$: “The paper claims that the convex regions for pruned networks are enlarged. This means that pruned networks have a larger region in which linear convergence is achieved (if I am not mistaken). Although this may mean the pruned network converges faster due to the larger region of convexity, this does not say anything about the quality of the solution, correct? It seems like this analysis cannot be used as evidence for winning tickets having superior performance (though superiority is shown in other areas with sample complexity/generalization bounds)”
>
> $\textbf{A3}$: We appreciate your concerns about Theorem 1, and your understanding is correct if only considering Theorem 1. However, the enlarged convex region still provides some insight into understanding the properties of the winning tickets.
>
> First, as the reviewer said, an enlarged convex region near the ground truth suggests a faster convergence rate to the ground truth in the population risk function. A faster convergence indicates that with the same number of iterations, the model trained on a pruned network can be closer to the ground truth than the model trained on the dense network in the noisy setting, which is also reflected in Theorem 2.
>
> Second, we want to share some insights into understanding the concept of “linearly connected region” proposed in [20]. As Figure 1 shown in [20], the authors claim that the learning is stable if the linear interpolation of the learned models with SGD noises still remains similar in performance. As suggested by Figures 2 & 5 in [20], even with the same initialization, the original dense network is unstable except for Lenet (MNIST), but the winning ticket is stable in all cases. The intuition is that the learned model within the convex region is stable. Therefore, we believe the concept of a “linearly connected region” is connected with the convex region. Intuitively, we conjecture that the winning ticket shows a better performance in the stability analysis because it has a larger convex region. Or in the other words, a larger convex region indicates that the learning is more likely to be stable in the linearly connected region.
>
> Third, the superiority of the winning ticket in the sample complexity indicates a superior performance. That is because given the same number of training samples, learning a pruned network converges faster than learning a dense network, as shown in Remark 2.1. That means with the same number of iterations, the model returned when learning on a pruned network is closer to $W^*$, and that leads to better testing performance. Moreover, in the noisy setting, the difference of the returned model to $W^*$ is also smaller when training on a pruned network, as shown in eqn. (10).

---

> > ### Comment · Reviewer_Udgq · 2021-08-20
> > **Response to rebuttal part two**
> >
> > I find the authors' responses to be very useful.
> >
> > Thank you for the clarification in Q2, which allows me to better understand the practical methodology that is being analyzed theoretically. Specifically, it seems that the authors study a normal pruning setup, which (in my opinion) actually makes the analysis more valuable.
> >
> > I believe asking to remove the Gaussian assumption is unreasonable given the discussion above, and it seem this setup is in line with current theoretical results on neural networks.
> >
> > In certain cases, analysis of multi-layer networks has been made possible in theory (both in general NN analysis and theoretical LTH papers). See these papers for some inspiration: https://arxiv.org/abs/2010.15969 and https://arxiv.org/abs/2003.05508. However, given the current state of theory for neural networks, I do not think extending to multiple layers is a reason for rejection (i.e., it is seldom done in many papers that still meet the requirements for neurIPS).
> >
> > I would ask that the authors: i) include a discussion in the paper of why the gaussian assumption is valid (i.e., similar to the rebuttal above but obviously more brief), ii) try to incorporate your answer to Q2 above within the methodology of the paper (i.e., it would be nice to avoid confusion on the pruning setup because of the ``teacher-student'' description), and iii) incorporate your answer to Q3 within the discussion of the paper -- I believe the insight you provide within the Q3 portion of the rebuttal is valuable and should be included/emphasized.
> >
> > Based on my comments above, I raise my score one point.

---

> > > ### Author Response · Authors · 2021-08-20
> > > **Thank you**
> > >
> > > Thank you very much for raising the score. We are very glad to learn that our responses have successfully made further clarification and provided useful insights. We will follow your suggestions to revise the paper.

---

> ### Author Response · Authors · 2021-08-07
> **Point-to-point response to the concerns mentioned in Minor comments**
>
> $\textbf{Q4}$: “There are a lot of other papers that analyze over-parameterized neural networks that could be added to the related work (though maybe they are not as relevant to this work, I am not sure). For example, https://arxiv.org/abs/1902.04674”
>
> $\textbf{A4}$: Thanks for bringing the relative paper to us, and we believe it is a good and relevant paper to this work.
>
> $\textbf{Q5}$: “I would recommend that the numerical experiments devote more space/attention to experiments on real datasets. Right now, it seems to be denominated by synthetic datasets, though I understand this is in an attempt to numerically verify aspects of the theory.”
>
> $\textbf{A5}$: First, we appreciate your understanding that the synthetic data experiments are conducted to numerically verify aspects of the theory. Still, we provided experimental results on the Cifar-10 dataset using SOTA LTH algorithms IMP as shown in Figures 12. These real-world datasets and neural network models also justified our theoretical findings. One the other hand, there are many works [9, 10, 19, 49] that have explored and verified the superior performance of pruned networks, which already verified the faster convergence rate and better generalization error of the winning tickets. We believe it is not necessary to repeat the experiments, especially given the page limit. Nevertheless, if the reviewer has any suggestions on the experiments that can increase the credibility of this paper, please let us know as well.

---

> > ### Comment · Reviewer_Udgq · 2021-08-20
> > **Response to rebuttal part one**
> >
> > Thank you for this response. After giving some more thought I agree with the authors' position on Q5. Replicating experiments to show that winning tickets generalize well is pointless. This would only be needed if a new pruning methodology is proposed, while this paper studies the theoretical underpinnings of widely-studied methods in practice (this became more clear to me after reading the response to Q2 below). I would ask that the authors please include a note similar to the response to Q5 above within their paper, as to avoid criticism regarding the synthetic experiments. Possibly the authors could even reference some work with large-scale experiments that match the methodology being studied, just to provide confidence that the methodology being studied is already empirically validated.

---

> > > ### Author Response · Authors · 2021-08-20
> > > **Thank you**
> > >
> > > Thank you very much for the great suggestion for our work. We will include the corresponding discussions within our paper to improve the clarity and validity of our studies, and we are grateful for your comments in helping us strengthen the paper further.

---

### Official Review · Reviewer_b8vh · 2021-07-16

**Rating:** 3
**Confidence:** 4

**Summary:**

The paper analyzes the possibility of recovering the weight of a teacher network using a student network which is as dense as the original one. The authors prove that with enough samples drawn from the teacher network, the recovery can be done on one-hidden-layer neural networks with linear convergence speed. The authors also discuss the correlation between the needed samples and the sparsity of the teacher network.

To sum up, the well-designed structure makes the workﬂow clear and easy to follow, but the further analysis and discussion are expected to clarify some contributions in the techniques as well as in the evaluation section.

**Ethical Concerns:**

No explicit ethical concerns

**Limitations And Societal Impact:**

The paper provided some interesting theoretical insights in the training process of pruned neural networks. However, the analysis is based on impractical assumptions while the derived bound is not tight enough, such that the proposed algorithm is difficult to work on real-scenario neural networks.

**Main Review:**

Strength
1. The paper analyzes the influence of the number of remaining parameters in pruned subnetworks and derives the required number of samples for successful convergence.

2. The authors conduct the theoretical analysis and justification for improved generalization error of winning ticket in the LTH.

3. The extensive evaluation on both synthetic and real datasets demonstrates the effectiveness of the proposed approach for neural network pruning.

Weakness
1. The strong assumption that one-hidden-layer neural networks and the input data must follow the Gaussian distribution seriously limit the applicability of the proposed method.

2. The authors claims to provide the theoretical analysis and justification of the LTH. The paper learns pruned neural networks by using the teacher-student setup. However, this is inconsistent with the Lottery Ticket Hypothesis setting that the pruned subnetwork should be "trained in isolation".

3. In the main theorems (Theorems 1 and 2), the number of samples drawn from the teacher model are required to be proportional to \Omega(K^4) or \Omega(K^6), where K is number of the neurons in the neural network. This may result in a serious efficiency issue, such that the method only work on small-size neural networks. In fact, the paper conduct the experiments over neural networks with merely 5 or 10 neurons.

4. It would be nice to conduct the experiments with the SOTA LTH and neural network pruning algorithms.

-------------------

Post-rebuttal update:

Thanks for the authors' efforts in addressing the raised concerns. I have read the author response and other reviews and keep my score due to the following two reasons.

In the paper, the derived theoretical justification or explanation over LTH or network pruning is based on a strong assumption of the one-hidden-layer neural networks and the input data with Gaussian distribution. Typically, LTH or network pruning methods don't have such strong assumption. If the authors claim the theoretical justification over LTH or network pruning is the main contribution of this paper, then the authors need to extend the proof to general neural networks without this restriction, although this extension is non-trivial. Otherwise, it is hard to say that the paper theoretically justifies the LTH or network pruning. In fact, real-world neural networks are often not satisfied with this assumption, such as Lenet-5 and Resnet-50 used in the experiments in the paper.

Local convexity (radius of convex region) and convergence are two main theoretical results in the paper. They are validated on synthetic data with small networks (5 or 10 neurons), but not on real data with large ones (Lenet-5, and Resnet-50).

**Time Spent Reviewing:**

4

---

> ### Author Response · Authors · 2021-08-07
> **Point-to-point response to the concerns mentioned in comments.**
>
> In the following paragraphs, we will provide the point-to-point response to the concerns mentioned in comments.
>
> $\textbf{Q1}$: "The strong assumption that one-hidden-layer neural networks and the input data must follow the Gaussian distribution seriously limit the applicability of the proposed method.”
>
> $\textbf{A1}$: We admit that there are still gaps between theoretical analyses and numerical results in assumptions. However, due to the complexity of neural networks, it is currently impossible to theoretically characterize the behavior of a neural network (NN) without any assumption or regularization.
>
> (Gaussian input distribution.) Gaussian distribution is currently a common assumption for theoretical works especially under the setup of a finite number of samples [8, 16, 46, 63, 67]. This assumption is motivated by the data whitening and batch normalization techniques that are commonly used in practice to improve the learning performance. The  learning  method  converges  faster  if  the  inputs  are whitened  to  be  the  standard  Gaussian [L1].  Batch  normalization [L2]  modifies  the  mean  and  variance  in  each  layer  and  is  a popular practical method to achieve fast and stable convergence. For example, the python code “transforms.Normalize” is a standard data processing method and also used in our simulation experiments when loading the datasets like MINST, Cifar-10, Cifar-100 and SVHN. Moreover, analyzing the standard Gaussian distribution also provides an insight into understanding rotation-invariant distributions. On the other hand, there exist major issues in developing the theoretical explanations for general distributions. First, the landscape of the population risk function depends on the distribution, and completely new tools need to be developed to analyze the landscape for general input distributions. Second, our objective is to reduce both training error and testing error, and that requires the empirical risk function to be a good approximation of the population risk function, at least in the neighborhood of $W^*$. Otherwise, a model with a small training error may have a large test error. The Gaussian assumption helps to capture the distance of these two risk functions, and new concentration bounds are needed for general distributions.
>
> Thus, the standard Gaussian assumption is well-acceptable in the current theoretical analyses of neural networks, but we will be investigating the extension to other distributions.
>
> [L1] Y. LeCun, L. Bottou, G. B. Orr, and K.-R. Müller, “Efficient backprop,” in Neural Networks: Tricks of the Trade, Springer, 1998, pp. 9–50.
>
> [L2] S. Loffe and C. Szegedy, “Batch normalization: Accelerating deep network training by reducing internal covariate shift,” in International Conference on Machine Learning, 2015, pp. 448–456.
>
>
> (One-hidden-layer neural network.) The setup of the one-hidden-layer neural network enables us to analyze the convergence of the trained model to the optimal one with both zero training error and zero testing error in noiseless cases. Almost all the existing works that achieve both zero training and testing errors are limited to one-hidden-layer neural networks. To the best of our knowledge, only one paper can achieve this on two-hidden-layer fully-connected neural networks (https://arxiv.org/abs/1811.04918) under the setup of overparameterization, but the convergence rate is proved as sub-linear. Beyond two-hidden layers, all the existing theoretical analyses can either achieve zero training error or develop bounds of the generalization gap (difference between the training error and the testing error), but no existing theoretical works can guarantee both a small training error and a small testing error simultaneously. Therefore, in terms of achieving both zero training and zero testing error theoretically, we believe the one-hidden-layer neural network is still the state-of-the-art setup.
>
> Nevertheless, to diminish the limitation of the one-hidden-layer case, the conclusions derived from the one-hidden-layer case are verified for multi-layer cases through numerical results as shown in Figures 11 & 12, which are conducted on Lenet-5 and Resnet-50, respectively. One can observe that the results for real data and multi-layer neural networks coincide with our findings for one-hidden-layer cases as well. For example, a properly pruned network (i.e., winning ticket) helps reduce the sample complexity required to reach the same test accuracy as the original dense model.
>
>
> $\textbf{Q2}$: “The authors claim to provide the theoretical analysis and justification of the LTH. The paper learns pruned neural networks by using the teacher-student setup. However, this is inconsistent with the Lottery Ticket Hypothesis setting that the pruned subnetwork should be ‘trained in isolation’.”
>
> $\textbf{A2}$: We apologize for the confusion on the teacher-student setup defined in this paper. As mentioned before,  we are not using teacher-student knowledge distillation in this paper. We only borrow the terminology "teacher-student’’ to reflect that the training data are generated from an unknown optimal network, and the training is carried out on a student network with the hope of recovering the ground-truth model.  Thus, the teacher model represents the oracle (ground-truth) model, and the student model stands for the learned model. The training process is indeed isolated, and no additional information is provided by the teacher model, i.e. no teacher-guided predictions. The input data and corresponding labels can be represented by an oracle model as shown in eqn. (1) in the paper. Given the data generated from the oracle model, the training process is conducted on a pruned neural network via solving eqn. (3) (just like a standard LTH implementation). During the training, there is no other additional information provided by the oracle model.
>
>
> $\textbf{Q3}$: “In the main theorems (Theorems 1 and 2), the number of samples drawn from the teacher model are required to be proportional to \Omega(K^4) or \Omega(K^6), where K is the number of the neurons in the neural network. This may result in a serious efficiency issue, such that the method only works on small-sized neural networks. In fact, the paper conducts the experiments over neural networks with merely 5 or 10 neurons.”
>
> $\textbf{A3}$: First, we respectfully point out that the comment “the paper [conducts] the experiments over neural networks with merely 5 or 10 neurons” is imprecise. We also provided experiments on Lenet-5 and Resnet-50 as in Figures 11 & 12, where Lenet-5 is a 5-layer convolutional neural network with around 60K parameters and Resnet-50 is 50-layer deep with over 23 million trainable parameters.  Second, we need to clarify that we did not claim that the dependence of our sample complexity on $K$ is optimal. Its tightness remains an open question and requires much more research effort. However, this will not prevent us from acquiring useful and technically grounded insights on the merits of winning tickets. For example, we show the dependence of the sample complexity on $d$ and $r$, where $d$ is the input dimension, and $r$ is the weight sparsity. Our empirical results also show that the dependence of  $K^4$ or $K^6$ in the sample complexity bound might just be a sufficient condition, but not a necessary requirement. For example, in our experiments on synthetic data, the number of samples is slightly larger than Kr, less than the predicted value by the bound. Lastly, almost all theoretical studies on neural networks have a gap with empirical studies. For example,  for fully connected neural networks [22, 63], the corresponding works omit the discussion of $K$ and leave it as poly($K$) in the sample complexity bounds. Another example is the widely studied neural tangent kernel (NTK), which requires the number of neurons to be a high polynomial of the number of samples, and that number is the order-of-magnitude larger than the actual number of neurons used in practice. Still, NTK-based analyses provide important insights about learning neural networks. Similarly, we believe our work also provides interesting insights to study the lottery ticket hypothesis.
>
> $\textbf{Q4}$: “It would be nice to conduct the experiments with the SOTA LTH and neural network pruning algorithms.”
>
> $\textbf{A4}$: We would like to respectfully point out that we have already conducted experiments using “SOTA LTH and neural network pruning algorithms,’’ as the reviewer might have missed it. We have considered existing pruned neural network algorithms for numerical experiments to support our theoretical analysis, and the results shown in sections 4.2 and 4.3 are based on GraSP algorithm [49] and IMP algorithm [19], respectively. Additionally, the performance evaluation of the algorithms has been conducted as training Cifar-10 dataset on the Reset-50 network, which is a standard way for LTH algorithms as well in [9, 10, 19, 49]. In addition, if the reviewer has any specific suggestions on the experiments that can increase the credibility of this paper, please let us know.

---

> ### Author Response · Authors · 2021-08-07
> **Clarification of some possible misunderstandings**
>
> Thanks for your comments. We want to first clarify some possible $\textbf{misunderstandings}$ about this paper reflected from the questions above.
>
> 1. Our teacher-student setup does NOT refer to the setup of  knowledge distillation from a teacher model to a student model. If there is a confusion, we sincerely apologize for that. In this paper, we only borrow the terminology "teacher-student’’ to reflect that the training data are generated from an UNKNOWN optimal network (that we call 'teacher’), and the training is carried out on a learner (that we call 'student’) with the hope to recover the optimal model.  Please see lines 127-135 in the paper for our definition. Our purpose is to use this terminology to formalize the theoretical analysis on LTH. There is no joint teacher-student training and our implementation of LTH follows the standard pruning, re-winding, and re-training setting. Based on reviewer’s comment, we plan to replace ‘teacher-student’ with `oracle-leaner’ to avoid the confusion. Therefore,  the pruned network is indeed “trained in isolation”, like the practical LTH setup.
>
> 2. The focus of this paper is to provide new theoretical explanations for the advantages of winning tickets over original dense networks, and a straight-forward application of bounds from the  prior work such as [63] yields a suboptimal bound for the pruned network as appraised by Reviewers HCZd, Udgq, and 1uan. Our purpose is not to develop a new learning method but to study the performance of the most common (mini-batch) stochastic  gradient descent algorithm (see in line 180) for LTH.
>
> 3. We respectfully disagree with the points 3 and 4 in weakness that  “the method only works on small-size neural networks” and “It would be nice to conduct the experiments with the SOTA LTH and neural network pruning algorithms”. In fact, we have provided experiments on some practical neural networks and SOTA LTH algorithms. For example, Figure 12 shows the results for training on Resnet-50, which has over 23 million trainable parameters. Also, the GraSP and IMP algorithms applied in Sections 4.2 & 4.3 are the SOTA pruning algorithms for lottery ticket hypothesis. One can check the corresponding literatures [9, 10, 19, 49] for details.
>
> Therefore, we hope our response clarifies these misunderstandings and the reviewer would consider to re-evaluate our work. To the best of the authors' knowledge, our work makes the first solid step to establish  the theoretical foundations of learning pruned neural networks and proves that training on a pruned network enjoys faster convergence rate and a better generalization error bound than training on a dense network.
>
> We will provide a point-to-point response to specific comments in the next comment

---

> ### Author Response · Authors · 2021-08-24
> **Looking forward to follow-up discussion**
>
> Dear $\textbf{Reviewer b8vh}$,
>
> Thanks again for your initial review comments. Since the deadline of discussion is approaching while we haven’t heard any feedback from you based on our responses, we are writing to express our willingness to provide any additional clarification that you may need.
>
> In our previous responses, we have carefully read your comments and made detailed responses summarized below, including pointing out some misunderstandings that we believe may affect your assessment of the merits of this work:
>
> ($\textbf{Teacher-student setup}$.) Our teacher-student setup does NOT refer to the setup of knowledge distillation from a teacher model to a student model. In this paper, we only borrow the terminology "teacher-student" to reflect that the training data are generated from an UNKNOWN optimal network (that we call ''teacher"), and the training is carried out on a learner (that we call "student") with the hope to recover the optimal model. Therefore, the pruned network is indeed "trained in isolation", like the practical LTH setup. Based on your comments and $\textbf{Reviewer Udgq’s feedback}$, we plan to replace "teacher-student" with "oracle-leaner" to avoid the confusion. As $\textbf{Reviewer Udgq}$ has accepted our clarification and raised the score, we are hoping the new terminology "oracle-leaner" will also help to address your concern about  our problem setup.
>
> ($\textbf{Verification on large size of networks}$.) We believe the initial review comments "the method only works on small-size neural networks" and "It would be nice to conduct the experiments with the SOTA LTH and neural network pruning algorithms"  misinterpreted our results. In this paper, we have provided experiments on practical neural networks and SOTA LTH algorithms. For example, Figure 12 shows the results for training on Resnet-50, which has over 23 million trainable parameters. Also, the GraSP and IMP algorithms applied in Sections 4.2 & 4.3 are the SOTA pruning algorithms for lottery ticket hypothesis. Moreover, we want to clarify that this paper actually studies the theoretical underpinnings of widely used methods in practice instead of proposing a new pruning methodology.  As recognized by $\textbf{Reviewer Udgq’s post-rebuttal feedback}$, "Replicating experiments to show that winning tickets generalize well would only be needed if a new pruning methodology is proposed, while this paper studies the theoretical underpinnings of widely-studied methods in practice". Thus, in this paper, we decided to conduct a comprehensive experiment on synthetic data first and then verify our theoretical findings over typical LTH-used pruning models and datasets. We hope that the existing experiments and our theoretical contributions sufficiently alleviate the reviewer's previous concern. If the reviewer has any specific suggestions on the experiments, please also feel free to let us know.
>
> Thank you for your time!
>
> Best
>
> Authors

---

> ### Author Response · Authors · 2021-09-01
> **Response to the post-rebuttal update**
>
> Thank you for checking our response and the other reviewers' comments. In the meantime, we would like to provide further responses to the two concerns that you raised.
>
> $\textbf{Why assumption is needed: One-hidden-layer.}$
>
> Most existing results in the line of neural network theory are centered on the one-hidden-layer cases since the analyses quickly become intractable when the number of layers increases. However, it does not prevent them from offering many interesting and insightful results. The advantages of winning tickets in convergence speed, enlarged local convex regions, and reduced sample complexities have been observed in practical neural network pruning. Our paper made the first step to justify them in the one-hidden-layer case. As another example,  neural tangent kernel based methods required that over-parametrization is orders-of-magnitude larger than the overparamterization used in practice. However, as it is one of the earliest approaches that provide the theoretical understanding of neural networks, such an impractical assumption does not prevent NTK from becoming one of the most well-known tools in the field of neural network theory.
>
> $\textbf{Why assumption is needed: Gaussian input.}$
>
> When studying stochastic gradient descent algorithms, it is important to utilize the concentration theorem to bound the distance of randomly selected samples to the expectations. As a smooth distribution with a small tail bound, the Gaussian distribution is always a good starting point. Specifically, if the analysis on Gaussian inputs does not show good performance, it is hard to indicate any theoretical guarantee on other distributions or real cases. By contrast, studying shallow neural networks under Gaussian input has not been fully solved yet until now. As proved in a recent work [R1], shallow neural networks with multiple neurons in the hidden layer have spurious local minima almost everywhere even if considering Gaussian input. A parallel work [R2] points out that a wide range of objective functions, including piece-wise continuous functions, are difficult to learn by gradient-based algorithms in general even if the distribution is fixed. All in all, Gaussian input assumption is still an important method in studying the convergence analysis of SGD algorithm in training shallow neural networks with generalization guarantee.
>
> [R1] Safran, Itay, and Ohad Shamir. "Spurious local minima are common in two-layer relu neural networks." International Conference on Machine Learning. PMLR, 2018.
>
> [R2] Shamir, Ohad. "Distribution-specific hardness of learning neural networks." The Journal of Machine Learning Research 19.1 (2018): 1135-1163.
>
> $\textbf{Why the insight is general:}$ $\textbf{Lottery Ticket Hypothesis}$ &  $\textbf{Linear mode connectivity.}$
>
> As we have mentioned in responses and acknowledged by other reviewers, there are many works [9, 10, 19, 49] that have explored and verified the superior performance of pruned networks for real data, which already empirically observed the faster convergence rate and better generalization error of the winning tickets. Our Theorem 2 (convergence analysis) suggests that a proper pruned network provably enjoys a faster convergence rate and smaller error to the ground truth, which is indeed the generalization error, and we believe the insight of our Theorem 2 is general. On the other hand, Theorem 1 (enlarged convex region) also casts an insight in understanding the concept of “linearly connected region” proposed in [20]. Starting from the same initialization, the authors in [20] use the same dataset but with different random noise to train the model, and the learning is stable if all the linear interpolations of two trained models still maintain relatively small training errors. In terms of the landscape, it is easy to verify that the learned model is stable if it lies in a flat-convex region. As suggested by Figures 2 & 5 in [20], even with the same initialization, the original dense network is unstable except for Lenet (MNIST), but the winning ticket is stable in all cases. Therefore, our theorem 1 can be viewed as a possible explanation for the experiment results in [20].
>
> We sincerely understand your concerns about our strong assumptions in terms of “one-hidden-layer” and “Gaussian distribution assumptions”. However, as in the early stage of theoretical analysis of neural networks, especially for the generalization guarantee, it is impossible to make progress without making any assumptions.  As acknowledged by Reviewer b8vh, “given the current state of theory for neural networks, I do not think extending to multiple layers is a reason for rejection (i.e., it is seldom done in many papers that still meet the requirements for neurIPS)”. For instance, the publications [11], [28] and [56] in our reference lists also consider one-layer analysis but have been accepted by  NeurIPS.
>
> Best regards,
>
> Authors

---

### Official Review · Reviewer_HCZd · 2021-07-18

**Rating:** 7
**Confidence:** 4

**Summary:**

The paper gives theoretical explanation for improved generalization error of winning tickets for which only empirical results are known. Their results are based on teacher-student setting in which training samples are assumed to be generated from an unknown teacher network and student network is supposed to learn only from those samples. They give an accelerated gradient descent method for learning the pruned  network and convergence and sample complexity analysis for this algorithm. The empirical risk function is shown to have an enlarged convex region for a pruned network, which justifies the importance of the winning ticket. Learning on pruned network with the AGD algorithm gives a model closer to the teacher network with the same number of iterations, which implies better generalization of the trained pruned network. These findings are validated with experiments on synthetic and real datasets (MNIST, CIFAR-10).


**Limitations And Societal Impact:**

There is a limitations section, which highlights key limitations of this work, like it is based on teacher-student setting and one-hidden layer neural network.

**Main Review:**

There is significant empirical evidence in support of lottery tickets, however theoretical understanding/justification is not very clear in the literature. This work tries to offer theoretical explanation for the winning tickets in the teacher student setup. Most of the results of the paper are inspired by [63], which has similar results for fully-connected one layer neural network. Straight forward application of bounds from this prior work yields suboptimal bounds for pruned network and hence they adapted it to pruned networks setting. In addition to some key differences in analysis like application of novel concentration bounds, non-smooth activation functions, they also construct new tensors for pruned networks. Overall, the paper is well written, looks technically sound. Though the proofs are inspired from prior works, the adaptation to pruned networks setting is also not trivial. In sum, despite limited technical novelty, I think its a good contribution towards theoretical understanding of winning tickets.

**Time Spent Reviewing:**

3

---

> ### Author Response · Authors · 2021-08-07
> **Reply to the comments by Reviewer HCZd**
>
> Thanks for your comments, and we are delighted to hear your positive comments on our contribution to the theoretical explanation of the winning tickets. Following your precise assessment, we also believe that our work makes a solid step to establish the theoretical foundations of learning pruned neural networks and proves that training on a pruned network enjoys a faster convergence rate and a better generalization error bound than training on a dense network. We also want to remark that a straightforward application of bounds from the prior work such as [63] would yield a suboptimal bound, which is a linear function of the feature dimension instead of the weight sparsity, for the pruned network. We believe that our work provides a state-of-the-art bound that is a linear function of the weight sparsity for the pruned networks.

---

### Decision · Program_Chairs · 2021-09-27

**Decision:**

Accept (Poster)

**Comment:**

We thank the authors for this submission. The paper well-motivates the approach. The authors have provided extensive responses to the concerns raised and the AC + reviewers really thank them for their effort. Overall, the new results obtained during the rebuttal definitely improve the quality of the paper. We all believe that the inclusion of these results (summarized) during the rebuttal period is something that does not heavily change the message of this paper.

There was discussion and consensus that this work is interesting. Having in mind issues/concerns raised by the reviewers and the authors (via private communication), the main points of reviewers during further discussion were that this paper deserves publication, given the promised fixes by the authors during the discussion period.